# Prediction of the 3D cancer genome from whole-genome sequencing using InfoHiC

Yeonghun Lee[1], Sung-Hye Park [ID] [2,3] & Hyunju Lee [ID] [1,4 ✉]

## Abstract

The 3D genome prediction in cancer is crucial for uncovering the impact of structural variations (SVs) on tumorigenesis, especially when they are present in noncoding regions. We present InfoHiC, a systemic framework for predicting the 3D cancer genome directly from whole-genome sequencing (WGS). InfoHiC utilizes contig-specific copy number encoding on the SV contig assembly, and performs a contig-to-total Hi-C conversion for the cancer Hi-C prediction from multiple SV contigs. We showed that InfoHiC can predict 3D genome folding from all types of SVs using breast cancer cell line data. We applied it to WGS data of patients with breast cancer and pediatric patients with medulloblastoma, and identified neo topologically associating domains. For breast cancer, we discovered super-enhancer hijacking events associated with oncogenic overexpression and poor survival outcomes. For medulloblastoma, we found SVs in noncoding regions that caused super-enhancer hijacking events of medulloblastoma driver genes (*GFI1*, *GFI1B*, and *PRDM6*). In addition, we provide trained models for cancer Hi-C prediction from WGS at https://github.com/dmcb-gist/InfoHiC, uncovering the impacts of SVs in cancer patients and revealing novel therapeutic targets.

**Keywords** Hi-C Prediction; 3D Genome; Structural Variation; Cancer Genome; Deep Learning
**Subject Categories** Cancer; Chromatin, Transcription & Genomics; Computational Biology

## Introduction

Complex genomic rearrangements have been observed in various cancers (Lee and Lee, 2021; Hadi et al, 2020). High-throughput sequencing has revealed breakpoints of genomic rearrangements or so-called structural variations (SVs) at the single-base level, and an association with tumorigenesis in terms of copy number alterations (CNAs) or gene fusions (Hurles et al, 2008). A recent study revealed complex genomic rearrangements, including homogeneously staining regions (HSRs), double minutes (DMs), and breakage-fusion-bridge cycles, which contributed to massive gene copy number (CN) changes (Lee and Lee, 2021). However, most SV breakpoints have been found in noncoding regions or outside genes, and their effect on tumorigenesis remains unclear.

Recent studies (Akdemir et al, 2020; Helmsauer et al, 2020; Spielmann et al, 2018) using Hi-C data have advanced our understanding of the 3D genome organization of DNA sequences. Topologically associating domains (TADs) (Dixon et al, 2012) are key components of the 3D genome architecture, and DNA elements in the same TAD are more likely to interact with each other. SVs might form neo-TADs and develop novel interactions between gene promoters and enhancers (Spielmann et al, 2018). However, to identify neo-TAD formation from cancer patients, high-resolution Hi-C data are required, which are expensive to obtain. Recently, several deep learning methods (Schwessinger et al, 2020; Fudenberg et al, 2020; Zhou, 2022) have shown an analytical potential of Hi-C prediction; however, they have technical limitations. First, the existing methods were developed to predict Hi-C of noncancerous samples, and applications were restricted to few instances of simple SVs. Second, the existing methods lack an interface from whole-genome sequencing (WGS) to DNA sequences. They require a user-provided DNA sequence, and the user should utilize a third-party tool for SV contig assembly. Thus, they are not solely applicable to WGS datasets. Third, the existing methods used the reference sequence for training, which ignores CNA- and SV-derived Hi-C contacts existing in cancer Hi-C data. Cancer genomes have genomic variations that can change Hi-C contact maps on a genome-wide scale (Akdemir et al, 2020). To leverage accumulated cancer Hi-C data (Kim et al, 2021a) for variant-derived 3D genome prediction, genomic variations existing in cancer genomes should be included in the prediction model. In addition, because there is no validation method for cancer Hi-C prediction, users should manipulate the observed Hi-C contact map (Tan et al, 2023; Wang et al, 2021) to validate prediction results outside of their prediction workflows. Because of such limitations, the 3D cancer genome has been poorly investigated by the existing prediction methods.

In this study, we present a systemic framework named InfoHiC that enables cancer Hi-C prediction directly from WGS. Based on complex genomic rearrangements in different haplotypes, InfoHiC predicts the cancer Hi-C matrix in SV contig and total Hi-C views. We then validated our model using Hi-C data from breast cancer cell lines (T47D, BT474, HCC1954, SKBR3, and MCF7) and showed that

[1]School of Electrical Engineering and Computer Science, Gwangju Institute of Science and Technology, 123 Cheomdangwagi-ro, Buk-gu, Gwangju 61005, Republic of Korea. [2]Department of Pathology, Seoul National University Hospital, Seoul National University College of Medicine, 103 Daehak-ro, Jongno-gu, Seoul 03080, Republic of Korea. [3]Neuroscience Research Institute, Seoul National University College of Medicine, 103 Daehak-ro, Jongno-gu, Seoul 03080, Republic of Korea. [4]AI Graduate School, Gwangju Institute of Science and Technology, 123 Cheomdangwagi-ro, Buk-gu, Gwangju 61005, Republic of Korea. ✉E-mail: hyunjulee@gist.ac.kr

InfoHiC outperforms the reference-based model that does not utilize WGS. We also identified neo-TADs and neo-loops, as well as enhancer hijacking events from contig Hi-C prediction in breast cancer cell lines, which can be validated using the InfoHiC validation scheme. Furthermore, we applied InfoHiC to patients with breast cancer from The Cancer Genome Atlas (TCGA) and identified enhancer hijacking events that resulted in oncogenic overexpression and poor survival outcomes. Finally, we applied InfoHiC to pediatric patients with medulloblastoma, and highlighted its analytical potential for discovering noncoding driver SVs associated with oncogenic expression.

## Results

### InfoHiC predicts cancer Hi-C from WGS

A cancer Hi-C matrix is a mixture of Hi-C matrices from multiple genomic contigs, in which Hi-C reads are observed as a sum in the reference coordinates. In contrast to reference-based models (Schwessinger et al, 2020; Fudenberg et al, 2020; Zhou, 2022), the cancer Hi-C prediction model requires merging of multiple prediction outputs. In addition, contigs have different genomic variations, including single nucleotide polymorphisms (SNPs), SVs, and CNAs; thus, genomic variations must be encoded in a contig matrix. To this end, we developed an architecture called InfoHiC, composed of convolutional neural networks (CNNs), that outputs chromatin interactions of genomic contigs and merges them in the total Hi-C views based on contig-specific copy number (CSCN) encoding, which represents genomic variants in the contig matrix (Fig. 1).

To predict a target pair of the Hi-C contact between genomic bins observed in the reference coordinates, we collected contig DNA sequences in which a contig target pair of genomic bins (40 kb) was matched with the reference target pair in a pre-defined genomic window with a length $l$ of 1 Mb or 2 Mb. For the contig, we used InfoGenomeR (Lee and Lee, 2021), which enables the assembly of

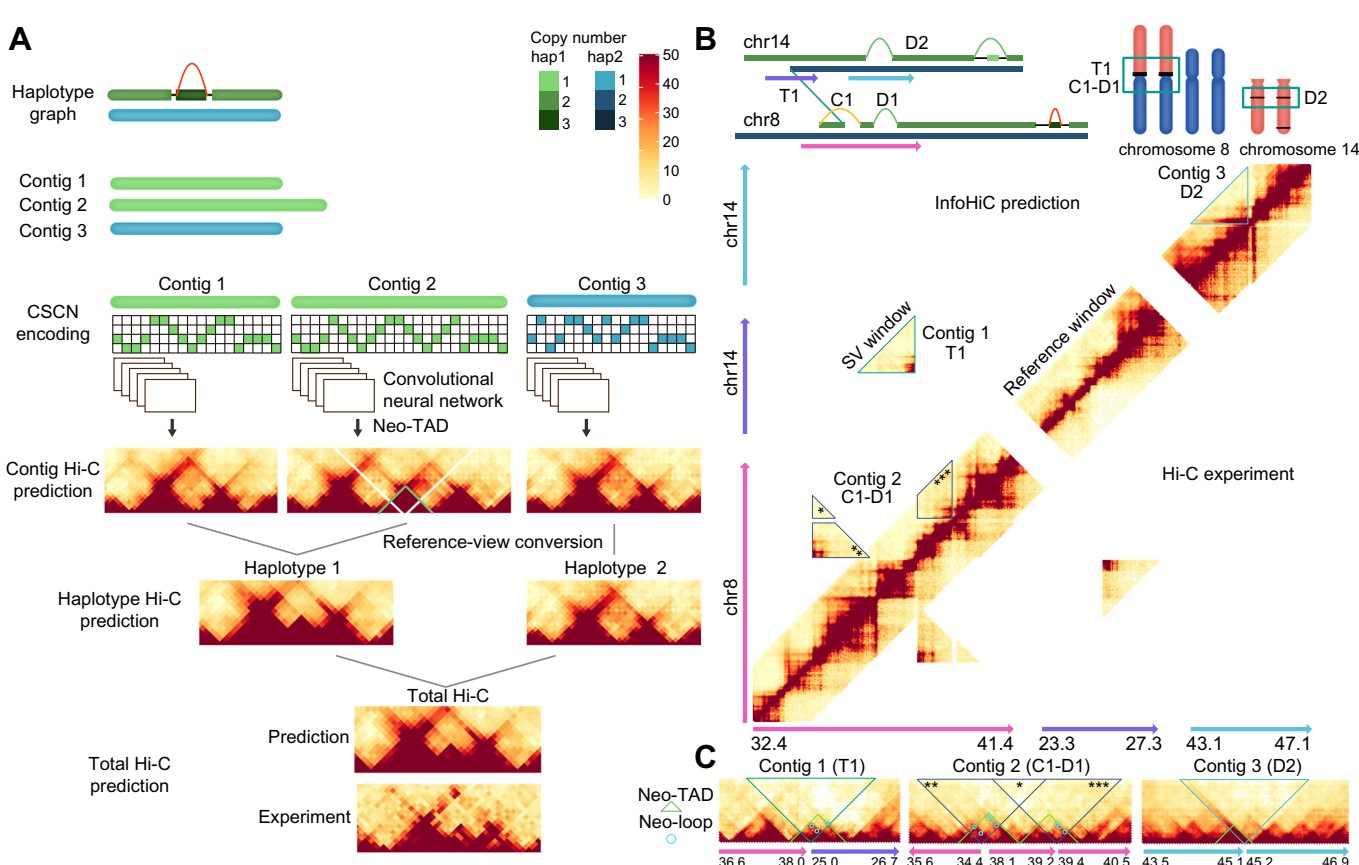

**Figure 1. Schematic diagram of InfoHiC.**

(A) InfoHiC prediction of the total Hi-C matrix from multiple genomic contigs. Copy numbers of each haplotype (haplotype 1 in green and haplotype 2 in blue) and the number of Hi-C reads (red) are represented by color saturation. Genomic contigs were derived from the haplotype breakpoint graph, and the contig-specific copy number matrix was employed to encode the SNP composition and CN of each contig. Contig 2 has tandem duplication, and the CNN predicts a neo-TAD (green) from contig 2. Hi-C predictions from contigs 1, 2, and 3 were successively merged into the haplotype Hi-C and the total Hi-C matrix. (B) InfoHiC prediction of chromosomes 8 and 14 in the T47D cell line. SVs (T1, C1, D1, and D2) are annotated in the haplotype graph and karyotypes with two copies of der(8)t(8;14). The upper diagonal matrix represents the InfoHiC prediction and the lower diagonal matrix represents the Hi-C experiment of T47D. SV windows (SV-induced Hi-C contacts) are outlined by each genomic contig. (C) Contig Hi-C prediction by InfoHiC. Three contig Hi-C matrices were annotated with SVs (T1, C1-D1, and D2), neo-TADs (green), and neo-loops (cyan); reference coordinates are shown below at the mega-base scale. SV windows in C1-D1 are annotated by asterisks (*) to clarify matches in (B, C). Note that the contigs were from two copies of der(8)t(8;14), and the matrices were encoded by two copies, instead of the one-hot encoding. Prediction was performed on the matrices where each base element has a contig-specific copy number value.

haplotype contigs from SVs and SNPs, and measures their CNs using WGS data. Next, based on haplotype contigs, we encoded contig DNA features using a $4 \times l$ matrix, where the row index represents the CSCN of each base sequence (A, C, G, and T). We used a CNN model as a component of the sequence-based prediction for each genomic contig, which has an architecture similar to deepC (Schwessinger et al, 2020), which has been developed for a single reference sequence. Each CNN model predicts a target contact, and target contacts may lie in different distances in the contig coordinates. The predicted contig Hi-C matrices were then transformed into reference coordinate matrices and merged into the haplotype Hi-C and the total Hi-C matrix (Fig. 1A).

In addition, we proposed a validation scheme of 3D genome folding changes predicted from SVs using InfoHiC, which mapped contig Hi-C matrices into the reference coordinates and validated them via a cancer Hi-C experiment (Fig. 1B). Scattered Hi-C contacts resulting from SVs were outlined by each genomic contig, and those from the InfoHiC prediction and the Hi-C experiment were compared in the reference coordinates (T1, C1-D1, and D2 in Fig. 1B). To discover neo-TADs and neo-loops from the SV-induced Hi-C contacts, we applied a contig-specific normalization on the contig Hi-C matrices, where the scattered Hi-C contacts were assembled in contig coordinates (contig 1, 2, and 3 in Fig. 1C). Hi-C normalization methods (Imakaev et al, 2012; Kim and Jung, 2021b) can be applied to the contig Hi-C matrices without the loss of SV-induced Hi-C contacts. Finally, InfoHiC annotated neo-TADs and neo-loops resulting from the SVs on the normalized contig Hi-C matrices (Fig. 1C).

## Comparison with the reference-based CNN model

We compared InfoHiC with the reference-based CNN model, deepC (Schwessinger et al, 2020). The main feature of InfoHiC against deepC is that genomic variants are included in the prediction model by utilizing WGS. In the training phase, we used the same transfer learning approach with deepC, where the first five CNN layers were pretrained to predict chromatin features (Zhou and Troyanskaya, 2015). The layers were initialized with the same weights between InfoHiC and deepC. Then, we used cancer Hi-C data to train the entire CNN models of deepC and InfoHiC with dilated CNN layers added to the first five layers. DeepC used the reference sequence for training, and it applied several normalization steps to raw Hi-C data, which includes (1) the implicit Hi-C normalization (Imakaev et al, 2012), (2) percentile normalization (Schwessinger et al, 2020), and (3) mean filter smoothing (Schwessinger et al, 2020). Because the normalization steps assumed that Hi-C contacts were from the reference sequence where no genomic variation exists, it is not readily applicable to cancer Hi-C data where genomic variations change Hi-C contact maps. Specifically, percentile normalization assigned fixed values (one to ten integers) depending on the percentile of the Hi-C intensity for each genomic distance. However, cancer Hi-C intensities depend on large genomic variants (CNAs and SVs); Hi-C contacts have different intensities even in the same genomic distance. Therefore, we removed the normalization steps, and trained deepC and InfoHiC using raw Hi-C data. InfoHiC assumed that the Hi-C contact could be derived from the haplotypes containing different SNPs and CNAs, and used haplotype sequences that were inferred from WGS for training.

In the testing phase, deepC was not available for SV-derived Hi-C prediction, specifically for large SVs or translocations commonly existing in cancer genomes (Lee and Lee, 2021). Thus, to compare

with InfoHiC, we assumed a common scenario of the deepC usage for SV-derived Hi-C prediction, named deepC-SV, where users would manipulate the reference sequence using variant calls from an SV caller, and then perform Hi-C prediction on the SV-derived sequence. Because the SV caller detects an SV in a local window, the sequence used for deepC prediction contained one SV. In contrast, InfoHiC used SV contigs reconstructed from InfoGenomeR, which could contain multiple SVs, and then performed Hi-C prediction on SV contigs. SV contigs could contain different SNPs and CNAs, which were encoded in the matrices of SV contig sequences. In addition, InfoHiC assumed that the Hi-C contacts can be derived from multiple SV contigs, and summarized prediction results from the multiple SV contigs in the output with the total Hi-C matrix. An illustration of the differences among deepC, deepC-SV, and InfoHiC for cancer Hi-C prediction is shown in Appendix Figs. S1 and S2).

## InfoHiC outperforms the reference-based CNN model

We trained InfoHiC using Hi-C datasets from a breast cancer cell line (T47D) and performed internal validation. To evaluate the prediction performance for regions with and without SVs separately, we first extracted genomic regions without SVs. Then, for these regions, we performed five random splits of 4-Mb reference genomic windows, and 80%, 10%, and 10% of regions were used for the training set, validation set, and test set 1, respectively. The 4-Mb window contained 100 (4 Mb/40 kb) output vectors at a 40-kb resolution. For each output vector, the 1-Mb or 2-Mb genomic sequence centered on the output vector was used for prediction. We named the 1-Mb or 2-Mb model to indicate the 1-Mb or 2-Mb genomic sequence was used for prediction of each output vector, respectively. The remaining genomic regions with SVs, named test set 2, were used for the SV prediction of 4-Mb reference windows (Appendix Fig. S1) and SV windows (novel windows derived from SVs, Appendix Fig. S2). We measured the distance-stratified correlation (DSC) (Schwessinger et al, 2020) and Pearson correlation for reference windows (4-Mb windows that exist in the reference coordinate) of test sets 1 and 2 and measured the Pearson correlation for SV windows of test set 2. The DSC was computed as an average of correlations across all genomic distances, where each correlation value was measured in a specific genomic distance.

In Fig. 2, the internal validation of test set 1 (Fig. 2; Appendix Table S1) showed performance improvement by utilizing WGS for Hi-C training. First, deepC did not have prior knowledge of SV breakpoints. Thus, reference windows around SV regions (test set 2) could not be excluded when training on the reference sequence, presenting a DSC value of 0.607. In contrast, InfoHiC excluded SV regions for training (breakpoint removal), showing performance improvement of 0.622 DSC. In addition, we utilized CSCN information obtained from WGS, and our model showed better performance than training on the reference sequence (CSCN decoding and encoding: 0.647 and 0.668 DSC, respectively). We then extended the 1-Mb prediction window to 2 Mb to identify TADs greater than 1 Mb. We used the CSCN encoding model for 2-Mb prediction, and the 2-Mb model showed the best performance when the 1-Mb model was transferred (InfoHiC transfer, 0.691 DSC).

Next, we validated InfoHiC for test set 2 containing SV-derived Hi-C contacts (Fig. 2; Appendix Table S2), which represent major challenges for cancer Hi-C prediction. Because deepC cannot use

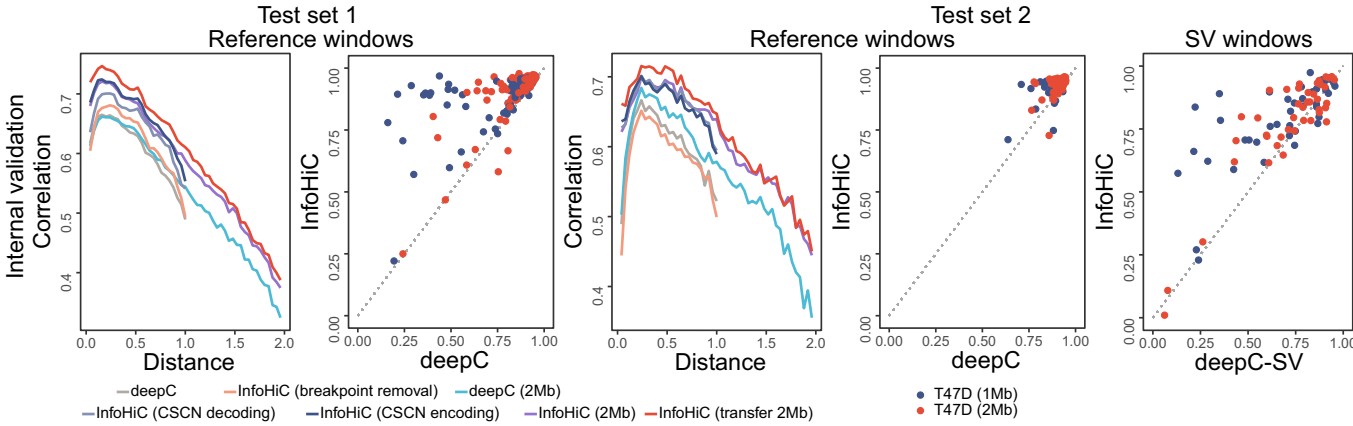

**Figure 2. Performance comparison between InfoHiC and deepC on the T47D cell line.**

InfoHiC and deepC were compared for training usage of the T47D cell line in the internal test sets 1 and 2. Distance-stratified correlations (line) are shown according to the mega-base scale distance range (0–2 Mb), and the Pearson correlation (dot) for each genomic window was compared between deepC or deepC-SV (bottom) and InfoHiC (left).

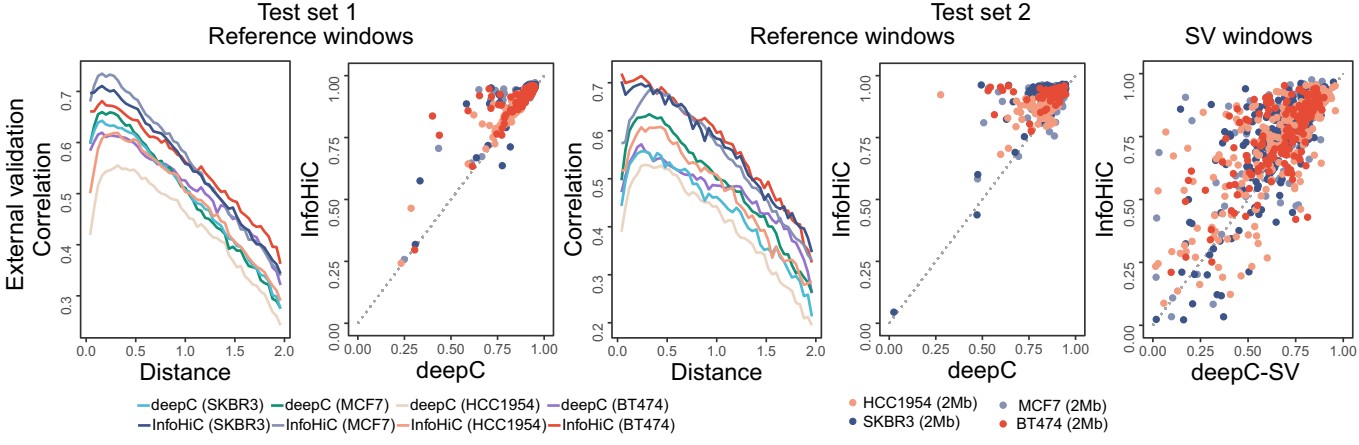

**Figure 3. Performance comparison between InfoHiC and deepC on four breast cancer cell lines (SKBR3, MCF7, HCC1954, and BT474).**

Methods were validated using the external test sets 1 and 2. Distance-stratified correlations (line) are shown according to the mega-base scale distance range (0–2 Mb), and the Pearson correlation (dot) for each genomic window was compared between deepC or deepC-SV (bottom) and InfoHiC (left).

cancer genomes and requires a user-provided input DNA sequence, the model cannot be solely compared to InfoHiC on test 2. Instead, we compared our model to a derivative usage of deepC; we generated a DNA sequence from each SV and provided it to deepC (deepC-SV). In contrast, InfoHiC utilized an SV contig assembly from WGS, rather than using individual SV calls. On test set 2, InfoHiC (1 Mb, 0.765; 2 Mb, 0.754 Pearson's R) showed better performance than deepC-SV (1 Mb, 0.649; 2 Mb, 0.702 Pearson's R) for T47D (SV window in Fig. 2). The result showed that InfoHiC with the WGS interface of the SV contig assembly outperformed the derivative usage of deepC (deepC-SV) for SV Hi-C prediction.

## External test on breast cancer cell lines

Further, we tested the models trained by T47D on four breast cancer cell lines (MCF7, BT474, SKBR3, and HCC1954) as external test sets

(Fig. 3; Appendix Table S2). Because the cell lines have different SVs from T47D, test sets were newly defined on each of these cell lines as follows: Test $set_{ext}$ 1 refers to reference window regions used in test set 1 for T47D after extracting regions with SVs for each cell line and test $set_{ext}$ 2 refers to regions with SVs for each cell line. First, for test $set_{ext}$ 1 without SVs, InfoHiC (2-Mb CSCN encoding model with 1-Mb transfer) outperformed deepC (InfoHiC, average 0.631 DSC; deepC, average 0.572 DSC). The regions did not contain SVs, and testing on the regions showed how well the model was trained by T47D Hi-C data, before applying it to SV regions. In contrast to deepC, InfoHiC utilized WGS for T47D training, resulting in a performance improvement on test $set_{ext}$ 1. Second, for test $set_{ext}$ 2 with SVs, InfoHiC (1 Mb, 0.699; 2 Mb, 0.715 Pearson's R) again outperformed deepC-SV (1 Mb, 0.588; 2 Mb, 0.642 average Pearson's R on SV windows). InfoHiC performed Hi-C prediction on the SV contig assembly from WGS data of the cell lines, which could improve performance on test $set_{ext}$ 2.

## Training on noncancerous cell lines

For some cancer types, high-resolution cancer Hi-C data may not be available. Under such circumstances, InfoHiC can be trained by noncancerous cell lines for cancer Hi-C prediction ("Methods and protocols"). In addition to the T47D cell line, we used breast epithelial cell lines (HMEC and MCF10A) for InfoHiC training. Then, we tested the models on the test set of four breast cancer cell lines (Appendix Fig. S3). The HMEC and MCF10A-trained models showed comparable performance to the T47D-trained model on test set$_{ext}$ 2: on reference windows (HMEC, 0.595; MCF10A, 0.604; T47D, 0.628 average DSC) and on SV windows (HMEC, 0.711; MCF10A, 0.713; T47D, 0.715 average Pearson's R). The T47D-trained model showed slightly better performance, and the following explanations are possible: (1) T47D Hi-C data had better quality, (2) CSCN encoding used for the T47D model showed better performance than CSCN decoding used for the HMEC and MCF10A models, or (3) Hi-C data from the cancer sample was more favorable to cancer Hi-C prediction than Hi-C data from normal tissues. Because InfoHiC is available for both cancer Hi-C data and noncancerous Hi-C data, there are more chances to perform Hi-C prediction on cancer samples; if Hi-C data of a certain type of cancer is not available, it can be trained by Hi-C data from the normal tissue and used for Hi-C prediction on cancer samples.

## Training on cancer Hi-C data at 10-kb resolution

For evaluating InfoHiC in a higher resolution, we trained InfoHiC at a 10-kb resolution. Based on five random splits that we used for a 40-kb resolution, we trained the models using T47D Hi-C data at a 10-kb resolution, and tested on the test set 1. InfoHiC and deepC showed lower average DSCs in the 10-kb models (InfoHiC, 0.456; deepC, 0.425) than in the 40-kb models (InfoHiC, 0.668; deepC, 0.607). Hi-C read coverage might be important for model performance, and higher coverage Hi-C data might be required for a 10-kb resolution. Therefore, we tested Hi-C data of a leukemia cell line (K562) that had higher coverage than T47D for the 10-kb training. The 3DIV (Kim et al, 2021a) Hi-C read mapping reported 0.76 M and 0.42 M properly paired reads in K562 and T47D, respectively. We trained the models using K562 Hi-C data and tested on a random split (80% training, 10% validation, and 10% test set 1). The K562 10-kb model showed better performance than the T47D 10-kb model (InfoHiC, 0.507; deepC, 0.442 average DSC on test set 1), showing that higher Hi-C coverage might be effective for 10-kb training. In addition, we tested the K562 10-kb model on test set 2 with SVs, and it showed 0.730 Pearson's R on SV windows. Moreover, neo-loop annotation might benefit from the 10-kb model, and we tested if the K562 10-kb model could predict the neo-loop formation around the *RAB36* gene that was reported in K562 (Wang et al, 2021). We could assemble the interchromosomal (chromosomes 9, 13, and 22) SV contig from WGS, which was the same SV contig previously found in K562 (Wang et al, 2021). When we performed Hi-C prediction on the SV contig using the K562 40-kb model, we could find a neo-TAD event of the *RAB36* gene, while neo-loops were not detected around the *RAB36* gene. Using the K562 10-kb model, we could successfully predict neo-loops around the *RAB36* gene (Appendix Fig. S4). We also provide the K562-trained InfoHiC model at https://github.com/

dmcb-gist/InfoHiC, which can help neo-loop prediction at a 10-kb resolution.

## InfoHiC predicts neo-TADs and neo-loops from cancer cell line WGS datasets

We applied InfoHiC to five breast cancer cell lines and predicted the contig Hi-C matrices. We used the 2-Mb CSCN encoding model transferred from the 1-Mb model and trained by T47D that showed the best performance on breast cancer cell line tests. We then performed contig-specific normalization to each contig Hi-C matrix and annotated the neo-TADs and neo-loops. The number of neo-TADs ranged from 47 to 303 (Fig. 4A). More than 20% of neo-TADs resulted from multiple SVs. In those cases, a Hi-C contact in the cancer Hi-C matrix was not unique to a single TAD, and the neo-TADs were overlapped with Hi-C contacts from other TADs (the average number of overlapping TADs was four, with a cut-off of 10% overlapping contacts). An example of overlapping Hi-C contacts is shown in Appendix Fig. S2, where Hi-C contacts from Contig 2 and Contig 3 are overlapped in the SV window. These results suggest that assemblies of complex SVs and multiple contig predictions are important for neo-TAD analysis. After annotating typical-enhancer (TE) and super-enhancer (SE) hijacking events of genes in neo-TADs, we searched enhancer hijacking events of several cancer-related genes (Futreal et al, 2004; Bailey et al, 2018) in the breast cancer cell lines (Appendix Table S3). Oncogenes from the Cosmic Cancer Gene Census (Futreal et al, 2004) and the TCGA driver gene list (Bailey et al, 2018) were selected as cancer-related genes to investigate the impact of enhancer hijacking events on them. Tumor suppressor genes were excluded from the cancer-related gene list. InfoHiC could predict SE hijacking events of cancer-related genes from complex SVs such as *MYC* and *PVT1* in SKBR3, which play an oncogenic role in various cancers (Jin et al, 2019a), as well as *GNA13* in BT474, which is upregulated and associated with poor outcomes in breast cancer (Jin et al, 2019b). SE hijacking events were confirmed by Hi-C experiments using the InfoHiC validation scheme (Appendix Fig. S5).

Next, we investigated the expression levels of genes in the neo-TADs and neo-loops (Fig. 4B; and Appendix Table S4). Genes in neo-TADs or neo-loops were compared with all the other genes which did not exist in neo-TADs or neo-loops (considered them to be in reference TADs or loops). Genes in neo-TADs were overexpressed compared with those in the reference TADs (Benjamini–Hochberg (BH) adjusted $P < 0.0001$, Student's one-sided $t$ test), suggesting that neo-TAD formation might induce interactions with other TEs or change the reference chromatin state (Akdemir et al, 2020). Moreover, SE hijacking events resulted in significant overexpression of genes compared with that in the reference TADs (BH-adjusted $P < 0.0001$) and neo-TADs (BH-adjusted $P = 0.011$). This overexpression was also observed in neo-loop annotations (BH-adjusted $P < 0.0001$ in all three comparisons). Furthermore, we performed analysis of covariance (ANCOVA) by controlling for CN effects on gene expression (Appendix Fig. S6). Overexpression by neo-TADs and SE hijacking events were observed under CN covariates ($P = 0.001$, one-way ANCOVA), suggesting that neo-TADs and SE hijacking events are involved in the dysregulation of gene expression in cancer.

An example of neo-TAD formation in the *IDO1* gene was observed in the HCC1954 cell line. HCC1954 had three distinct SV

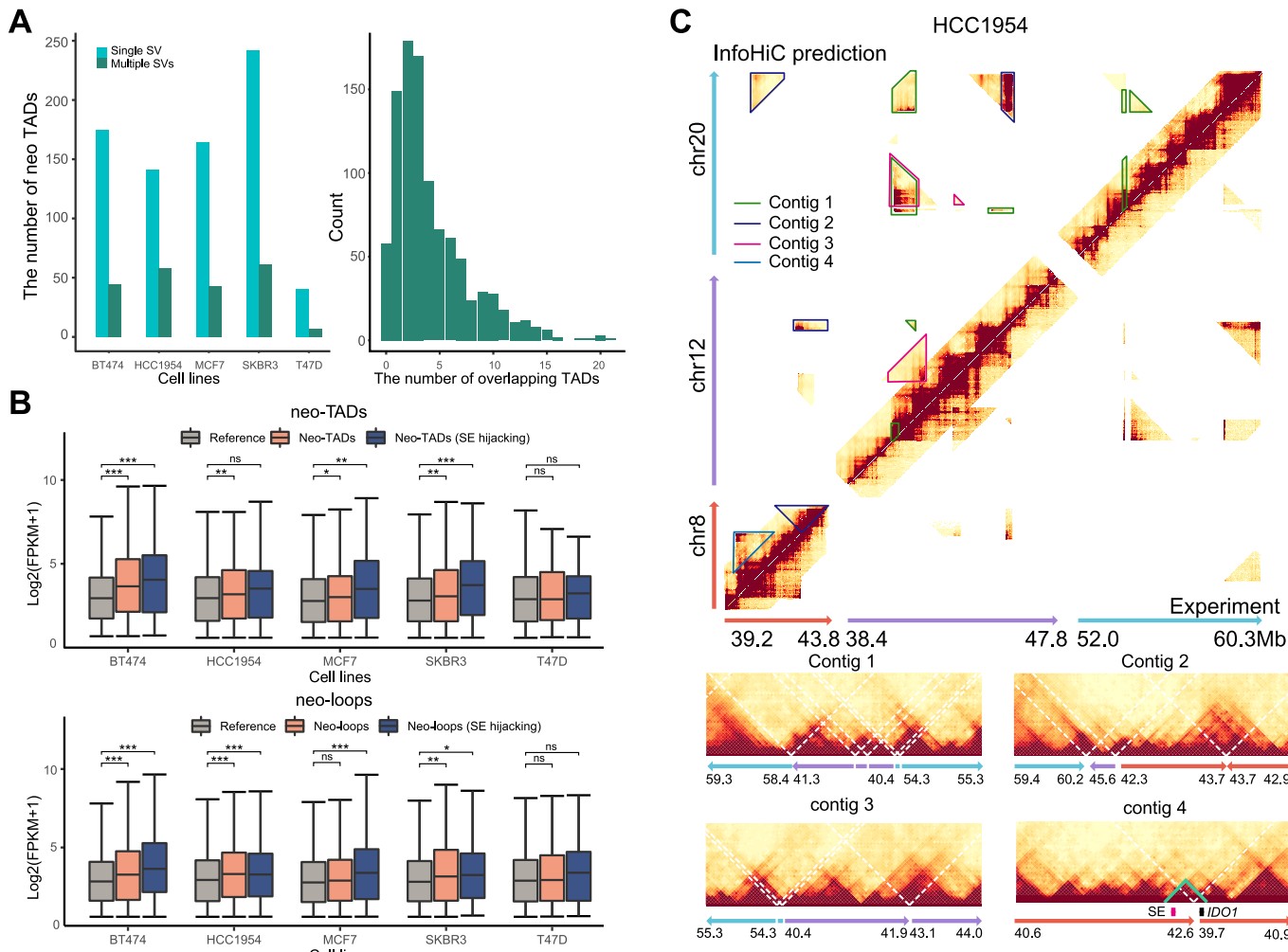

**Figure 4. InfoHiC analysis of breast cancer cell lines.**

(A) Statistics of the number of neo-TADs in each sample and a histogram of overlapping TAD counts of neo-TADs. (B) Boxplots of RNA-seq Log2(FPKM + 1) values depending on neo-TAD classes (top) and neo-loop classes (bottom) in each sample. Gene expression was measured on genes with FPKM > 0.5, excluding genes with near-zero expression values. Boxplot center lines are medians, box limits are upper and lower quartiles, and whiskers are 1.5× interquartile ranges. The *P* values from the one-sided Student's *t* test are shown in Appendix Table S4. \**P* < 0.05, \*\**P* < 0.01, and \*\*\**P* < 0.001 by the one-sided Student's *t* test versus the reference group, ns, not significant. For neo-TADs, the group *n* numbers and *P* values are *n* = 9594 (reference), *n* = 514 and *P* = 6.43e-15 (neo-TAD), and *n* = 100 and *P* = 3.79e-05 (SE hijacking) in BT474; *n* = 10,401, *n* = 315 and *P* = 5.80e-03, *n* = 75 and *P* = 0.120 in HCC1954; *n* = 13,333, *n* = 435 and *P* = 0.029, *n* = 104 and *p* = 1.23e-03 in MCF7; *n* = 12,752, *n* = 425 and *P* = 1.22e-03, *n* = 192 and *P* = 2.65e-07 in SKBR3; *n* = 9691, *n* = 46 and *P* = 0.306, *n* = 45 and *P* = 0.428 in T47D. For neo-loops, the group *n* numbers and *P* values are *n* = 9537 (reference), *n* = 608 and *P* = 1.21e-06 (neo-loop), and *n* = 441 and *P* = 1.49e-14 (SE hijacking) in BT474; *n* = 9944, *n* = 524 and *P* = 8.72e-05, *n* = 323 and *P* = 6.06e-04 in HCC1954; *n* = 12,889, *n* = 556 and *P* = 0.391, *n* = 427 and *P* = 8.30e-08 in MCF7; *n* = 12,985, *n* = 274 and *P* = 1.79e-03, *n* = 104 and *P* = 0.025 in SKBR3; *n* = 9524, *n* = 186 and *P* = 0.279, *n* = 72 and *P* = 0.114 in T47D. (C) InfoHiC prediction of chromosomes 8, 12, and 20 in the HCC1954 cell line. Hi-C contacts from different genomic contigs (contig 1, 2, 3, and 4) were annotated in the upper diagonal matrix of the total Hi-C (InfoHiC prediction). Lower diagonal matrix represents the Hi-C experiment for HCC1954. Contig Hi-C matrices of each contig are shown below with reference coordinates at the mega-base scale. Dotted white lines represent SV breakpoints. Contig 4 has a neo-TAD (green), where the hijacked SE (crimson) and the *IDO1* gene (black) interact with each other.

clusters, one of which was found among chromosomes 8, 12, and 20 (Fig. 4C). Moreover, SV clusters resulted in neo-TADs from different contigs. A tandem duplication was found near *IDO1*, which resulted in an SE hijacking event (contig 4 in Fig. 4C). Interaction between *IDO1* and the SE was absent in the other breast cell lines, indicating a neo-interaction. The *IDO1* gene showed the highest fragments per kilobase of transcripts per million mapped reads (FPKM) level among breast cancer cell lines from the Cancer Cell Line Encyclopedia (CCLE) (Appendix Fig. S7), whereas integer CNs of the genes did not significantly affect gene expression

(−0.002 Pearson's *R*). The correlation between *IDO1* expression and the intracellular level of an immunosuppressive metabolite, kynurenine, was reported in a recent CCLE report (Li et al, 2019). The HCC1954 cell line showed the second-highest kynurenine levels among the breast cancer cell lines. Collectively, these results showed that InfoHiC can provide evidence of the 3D genome context associated with *IDO1* overexpression and high kynurenine levels.

In addition, we investigated whether deepC-SV could predict InfoHiC results of SE hijacking events in neo-TADs. The number of

SE-hijacked genes predicted by InfoHiC ranged from 77 to 562 in the five breast cancer cell lines (total $n = 1567$). Because the SE-hijacked genes could interact with multiple SEs, we additionally counted all pairs of genes and SEs, which ranged from 257 to 2867 (total $n = 7879$). DeepC-SV could predict 85.2% (1335/1567) of SE-hijacked genes and 63.0% (4966/7879) of gene-SE pairs. Thirty-seven percent of gene-SE pairs were not detected by deepC-SV, even though deepC-SV predicted 1.6x more (total $n = 12,886$) gene-SE pairs than InfoHiC (Appendix Fig. 8). This could be because that InfoHiC predicted a Hi-C matrix on each SV contig, where multiple SVs could be clustered and generate a neo-TAD together (a neo-TAD from Contig 3 in Appendix Fig. S2), while deepC-SV predicted a Hi-C matrix for each SV (two neo-TADs from SV 1 and SV 2 in Appendix Fig. S2). Differences between deepC-SV and InfoHiC were shown in Appendix Fig. S2, where deepC-SV produced false positives (a–f and b–f) and false negatives (ab–hi) against the Hi-C experiment. Accordingly, experimental Hi-C intensities of deepC-SV gene-SE pairs could be lower than those predicted by InfoHiC. To investigate which gene-SE pairs were more accurate, we compared experimental Hi-C values of predicted gene-SE pairs (deepC-SV-specific, InfoHiC-specific, and shared events) in each cell line. Indeed, deepC-SV-specific events showed lower intensities in cancer Hi-C experiments across five cell lines compared to InfoHiC-specific events (Appendix Fig. S8). In BT474 and SKBR3, InfoHiC-specific events showed the highest Hi-C intensities. In HCC1954, MCF7, and T47D, shared events showed the highest Hi-C intensities, and InfoHiC-specific events showed the second-highest intensities. deepC-SV-specific events showed the lowest Hi-C intensities across five cell lines. Overall, the result indicated that events predicted by InfoHiC were observed in Hi-C experiments with higher signals than deepC-SV predicted events.

Next, we investigated cancer-related genes in gene-SE pairs. Among 60,234 genes annotated in the Ensembl database (Howe et al, 2021), DeepC-SV and InfoHiC predicted that 2764 and 1567 genes were involved in SE hijacking events, respectively. Among 382 cancer-related genes (Futreal et al, 2004; Bailey et al, 2018), DeepC-SV and InfoHiC discovered 25 ($P = 0.049$, one-sided Fisher's exact test) and 17 ($P = 0.024$, one-sided Fisher's exact test) cancer-related genes, respectively. Enrichment analysis with Fisher's exact test showed that InfoHiC ($P = 0.024$) had a lower $P$ value than DeepC-SV ($P = 0.049$), indicating cancer-related genes were more enriched by InfoHiC. Then, we investigated the Hi-C intensities of each gene-SE pair in the cancer Hi-C and noncancerous Hi-C experiments (HMEC and MCF10A). Fourteen genes were shared between deepC-SV and InfoHiC. Cancer-related genes exclusively found by InfoHiC ($n = 3$) showed an average 4.3-fold change in cancer cell line Hi-C data, while those found by deepC-SV ($n = 11$) showed an average 2.6-fold change, indicating that InfoHiC discovered cancer-related genes with higher Hi-C intensity evidence than deepC-SV. Furthermore, we observed that several gene-SE pairs found by deepC-SV have extremely low intensities (<5 Hi-C read counts) in cancer Hi-C experiments (Appendix Table S5), suggesting that deepC-SV discovered false-positive gene-SE pairs.

Among 17 cancer-related genes predicted by InfoHiC, deepC-SV could predict SE hijacking events of 14 cancer-related genes (82.4%, 14/17). However, deepC-SV could not find any SE involved with three cancer-related genes (17.6%, 3/17): NFATC2 and SALL4 in MCF7, TERT in HCC1954. These genes were involved with multiple SVs. In addition, deepC-SV could not find some of the gene-SE pairs: NT5C2 in T47D, and GNAS and BCL6 in SKBR3.

Rearrangements around *TERT* were previously reported to be involved with Hi-C contact map changes in HCC1954 (Akdemir et al, 2020). *SALL4* (Chen et al, 2020) and *GNAS* (Jin et al, 2019b) were known to promote breast cancer progression, and previously reported to be involved with SE hijacking events (Xu et al, 2022; Wang et al, 2021). When we measured Hi-C intensities of these gene-SE pairs (Appendix Table S6), the experiments showed higher intensities of the gene-SE pairs, compared to noncancerous cell lines, up to a 74X fold change (*GNAS* in SKBR3). For several genes (*TERT* in HCC1954, *SALL4* in MCF7, and *NT5C2* in T47D), the noncancerous cell lines showed zero Hi-C intensity. The results demonstrated that InfoHiC could predict SE hijacking events not found by deepC-SV, which could be verified in Hi-C experiments.

## Noncoding effect of SVs discovered from BRCA WGS datasets

Subsequently, using the 2-Mb CSCN encoding model trained by T47D, we applied InfoHiC to patients with BRCA ($n = 90$) in TCGA to investigate the noncoding effects of SVs. Neo-TADs were annotated on contig Hi-C prediction and were found with high-frequency peaks in chromosomes 1, 8, 11, 17, and 20 (bottom plots in Fig. 5A). Recurrent neo-TAD genes (found in ≥10% neo-TAD samples) were enriched in the KEGG breast cancer pathway (BH-adjusted $P = 0.014$), including breast cancer genes such as *FGF19*, *ERBB2*, *CCND1*, *MYC*, and *RPS6KB1*. Genes in neo-TADs were overexpressed under CN covariates ($P < 0.0001$, one-way ANCOVA), demonstrating the noncoding effects of SVs on gene expression (Fig. 5B; Appendix Table S7).

We also observed that neo-TAD formation frequently accompanied CNAs in breast cancer. To identify recurrent neo-TAD genes associated with overexpression control for CNAs, we performed a linear regression between gene CNs and expression in reference TAD samples without neo-TADs. Based on the regression line, we measured Cook's distance (an indicator of outlier influence on the regression line) (Altman and Krzywinski, 2016) for the neo-TAD samples. We counted the number of neo-TAD samples with a positive residual and a Cook's distance greater than $4/n$ ($n = 90$), which indicated overexpression compared with that in the reference TAD samples. Genes with such neo-TADs were found to have peaks in the 8p11, 11q13, and 17q11 regions (top plots in Fig. 5A). We selected genes with the following attributes: (1) more than 50% of neo-TAD samples showing a Cook's distance greater than $4/n$ and (2) with SE hijacking samples ≥4 as recurrent neo-TAD genes involved in overexpression (Appendix Table S8), which included known driver genes (Futreal et al, 2004) such as *ERBB2*, *CCND1*, *LASP1*, *CLTC*, and *CDK12*. We further investigated the survival outcomes of patients with recurrent neo-TAD genes across all BRCA data using RNA-seq from TCGA ($n = 1098$). A threshold for overexpression samples was defined as the minimum FPKM value over the Cook's distance cut-off for each gene (Appendix Table S8). We found three recurrent neo-TAD genes (*LASP1*, *CLTC*, and *MYO1D*) related to poor prognosis. *LASP1* overexpression is reportedly correlated with poor prognosis in breast cancers; however, the cause of *LASP1* over-expression has remained unclear (it does not result from CNAs) (Frietsch et al, 2010). Here, we discovered neo-TAD formation of *LASP1* associated with overexpression and poor prognosis (BH-adjusted $P = 0.019$, log-rank test) (Appendix Fig. S9).

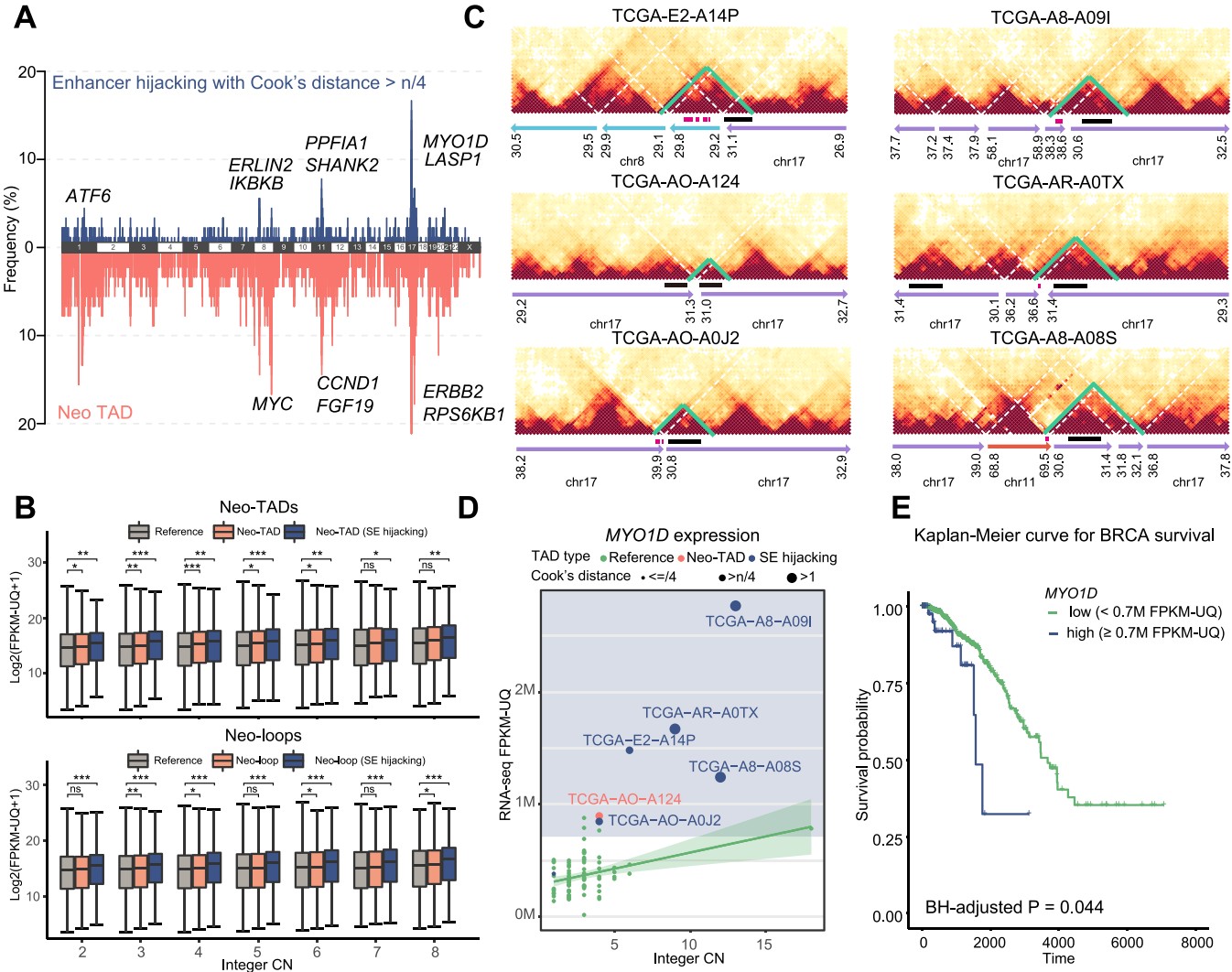

**Figure 5. InfoHiC analysis of patients with TCGA BRCA.**

(A) Frequencies of neo-TADs (apricot) and enhancer hijacking events (blue) associated with overexpression across chromosomes. Each peak represents the neo-TAD or enhancer hijacking frequency of each gene. The representative peaks were annotated using gene symbols. (B) Boxplots of RNA-seq Log2(FPKM-UQ + 1) values depending on neo-TAD classes (top) and neo-loop classes (bottom) in patients with BRCA according to integer CNs. Boxplot center lines are medians, box limits are upper and lower quartiles, and whiskers are 1.5× interquartile ranges. The P values from the one-sided Student's t test are shown in Appendix Table S7. *P < 0.05, **P < 0.01, and ***P < 0.001 by the one-sided Student's t test versus the reference group, ns, not significant. For neo-TADs, the group n numbers and P values are n = 900,669 (reference), n = 4363 and P = 0.025 (neo-TAD), and n = 1434 and P = 3.29e-03 (SE hijacking) in copy number 2; n = 835,576, n = 7364 and P = 4.94e-03, and n = 2480 and P = 8.73e-05 in copy number 3; n = 746,955, n = 9816 and P = 1.41e-05, and n = 3023 and P = 4.66e-03 in copy number 4; n = 332,513, n = 7775 and P = 0.046, and n = 3093 and P = 2.51e-05 in copy number 5; n = 111,171, n = 5535 and P = 0.046, and n = 1904 and P = 9.08e-03 in copy number 6; n = 52,380, n = 3136 and P = 0.145, and n = 1332 and P = 0.015 in copy number 7; n = 17,593, n = 1839 and P = 0.058, and n = 1012 and P = 1.66e-03 in copy number 8. For neo-loops, the group n numbers and P values are n = 860,538, n = 10,908 and P = 0.037, and n = 5742 and P = 5.01e-06 in copy number 2; n = 852,035, n = 15,608 and P = 4.48e-03, and n = 9072 and P = 1.12e-08 in copy number 3; n = 756,233, n = 20,563 and P = 0.017, and n = 11,898 and P = 2.20e-16 in copy number 4; n = 326,213, n = 15,003 and P = 0.113, and n = 8856 and P = 1.80e-06 in copy number 5; n = 104,949, n = 8818 and P = 0.037, and n = 6434 and P = 5.63e-05 in copy number 6; n = 48,319, n = 4837 and P = 0.395, and n = 3692 and P = 1.12e-05 in copy number 7; n = 15,357, n = 2648 and P = 0.050, and n = 2439 and P = 1.51e-05 in copy number 8. (C) InfoHiC prediction of contig Hi-C matrices of patients with BRCA. Neo-TADs are shown in green, with hijacked SEs (crimson) and the MYO1D gene (black). (D) Gene expression of MYO1D of patients with BRCA (dot) using WGS data (n = 90). A linear regression line between FPKM-UQ values and integer CNs is shown with the corresponding confidence interval range (green). Neo-TAD (apricot) and SE hijacking samples (blue) are shown in different circle sizes according to Cook's distance. Cut-off region of the FPKM-UQ value for overexpression is shown in the background (blue). (E) Kaplan–Meier curve for BRCA survival with RNA-seq data (n = 1098). P value was calculated using the log-rank test and adjusted using the BH procedure across neo-TAD overexpression genes.

The most recurrent neo-TAD formation associated with poor survival was found in the *MYO1D* gene (Fig. 5C). *MYO1D* is in a repressed TAD, indicating that its transcriptional activity is low in the reference state (Akdemir et al, 2020). Seven patients exhibited

neo-TAD formation of *MYO1D*. We found SE hijacking events with neo-TAD formation in six patients, five of whom showed overexpression of the *MYO1D* gene (Fig. 5D). One patient (TCGA-AO-A124) did not have an SE in the neo-TAD but also showed

overexpression of the *MYO1D* gene, as tandem duplication may have a position effect on gene expression (Loehlin and Carroll, 2016). Furthermore, overexpression of *MYO1D* was a prognostic factor for survival in TCGA BRCA patients (BH-adjusted $P = 0.044$, log-rank test) (Fig. 5E). Overexpression of *MYO1D* has previously been associated with breast cancer cell motility and viability (Ko et al, 2019). Collectively, these results suggest that InfoHiC can detect 3D genome changes associated with overexpression and poor prognosis.

In addition, we investigated whether the models trained by epithelial cell lines (HMEC and MCF10A) can reproduce the BRCA prediction results from the T47D-trained model. First, we measured the correlation between predicted Hi-C matrices from the epithelial and T47D cell line. The epithelial cell line model showed 0.928 (HMEC vs T47D) and 0.960 (MCF10A vs T47D) average Pearson's R on raw Hi-C prediction. The average DSCs were 0.829 (HMEC vs T47D) and 0.810 (MCF10A vs T47D) on raw Hi-C prediction, and after contig-specific normalization, the DSCs increased into 0.920 (HMEC vs T47D) and 0.921 (MCF10A vs T47D). Next, based on contig normalized Hi-C matrices, we performed neo-TAD prediction and measured reproducibility to determine how many enhancer hijacking events predicted by the T47D model could be reproduced by the epithelial cell line model. To quantify enhancer hijacking events, we counted all gene-SE pairs existing in the predicted neo-TADs from BRCA datasets. The total numbers of gene-SE pairs predicted by the T47D model from BRCA datasets were 44,365 in the 0 to 1-Mb range and 28,941 in the 0 to 0.5-Mb range. In the 0 to 1-Mb range, the HMEC and MCF10A models predicted 73.9% ($n = 32,782$) and 73.8% ($n = 32,760$) of 41,820 gene-SE pairs, and in the 0 to 0.5-Mb range, they predicted 83.4% ($n = 24,128$) and 83.2% ($n = 24,069$), respectively. While the DSC between Hi-C matrices from the epithelial and cancer models was greater than 0.9, the reproducibility of gene-SE pair prediction was decreased compared with that of DSC, because gene-SE pair prediction depended on neo-TAD annotation, which in turn depended on predicted Hi-C matrices. Gene-SE pair prediction was also sensitive to the gene-SE pair distance, and the epithelial models could reproduce a higher fraction of gene-SE pair prediction results from the T47D model in the 0.5-Mb range than those in in the 1-Mb range.

## Application of InfoHiC to pediatric patients with medulloblastoma

To demonstrate the analytical potential of InfoHiC in discovering noncoding driver SVs, we applied InfoHiC to five pediatric patients with medulloblastoma in our cohort. We used the 1-Mb CSCN decoding model trained by the neuro-progenitor cell line (NPC). After we performed WGS and RNA-seq analysis on data from these patients, the presence of driver mutations in coding regions were examined using cancer-related gene lists (Futreal et al, 2004; Northcott et al, 2017; Bailey et al, 2018) ("Methods and protocols"). Driver mutations were not found in three patients (PD2104, 2107, and 2109) by single nucleotide variants (SNVs), indels, and CNA calls (Appendix Tables S9 and S10). *MYC* amplification, which is a prevalent driver of group 3 medulloblastomas (Northcott et al, 2014), and LOH deletion of a tumor suppressor, *TSC1*, was found in the PD2105 and PD2110 patients, respectively. Further analysis of SVs using InfoGenomeR revealed derivative chromosomes in patients with medulloblastoma;

however, the impacts of SVs were unknown as most SVs occurred in noncoding regions and were not relevant to CN changes or fusion events of driver genes (Appendix Figs. S10–13). Next, we investigated SVs that might affect the 3D genome organization. Notably, we found complex SVs in the noncoding region in the driver mutation-negative case (PD2107), which included a subtype-specific SE of group 4 medulloblastomas near the *PRDM6* gene, the overexpression of which has been suggested as a driver of medulloblastoma (Northcott et al, 2017) (Fig. 6A). The patient also showed overexpression of *PRDM6* with a 90-fold change in FPKM compared to that in our cohort of brain tumors, suggesting that complex SVs may induce *PRDM6* overexpression (Fig. 6B). In addition, we found recurrent SVs near the *GFI1* family oncogenes; a deletion and tandem duplication near the *GFI1B* gene in the *TSC1*-deleted and *MYC*-amplified cases (PD2110 and PD2105, respectively), and reciprocal translocation near the *GFI1* gene in the driver mutation-negative case (PD2109). The patients showed overexpression of *GFI1* family oncogenes with more than 100-fold changes in FPKM (Fig. 6B), suggesting that these SVs could be medulloblastoma drivers associated with oncogenic overexpression.

Using InfoHiC, we discovered neo-TAD formation derived from complex SVs and an SE hijacking event in *PRDM6* (Fig. 6C). We then annotated the CTCF motif directions in CTCF ChIP-seq peak regions of TAD boundaries using PWMScan (Ambrosini et al, 2018) and checked the pairwise interactions of the CTCF ChIA-PET data (ENCODE Project Consortium, 2012), which supported the hypothesis that the reference TADs were arranged in forward–reverse orientations. Complex SVs included an inversion spanning from the SE to the CTCF motifs, which resulted in the SE hijacking event of the TAD containing *PRDM6* (Fig. 6C). This inversion maintained the forward–reverse CTCF arrangements, supporting neo-TAD formation with the SE (Fig. 6D). Complex SVs in the noncoding region near *PRDM6* were also reported in group 4 medulloblastomas in a previous study (Northcott et al, 2017), whereas neo-TAD formation by complex SVs remains unclear. Our results demonstrated that InfoHiC can validate neo-TAD formation by complex SVs and specify the noncoding driver SVs from the driver mutation-negative case (PD2107). Furthermore, InfoHiC revealed SE hijacking events of neo-TADs of *GFI1* oncogene families induced by various types of SVs (PD2105, 2109, and 2110) (Appendix Fig. S14; Fig. 6D). A deletion occurred across TAD boundaries causing TAD fusion (PD2110), a tandem duplication spanned an SE and the *GFI1B* gene causing a neo-TAD with the hijacked SE (PD2105), and a reciprocal t(1;5) translocation caused interchromosomal neo-TADs with distant SEs (PD2109), demonstrating that neo-TADs were involved with the activation of *GFI1* oncogene families.

## InfoHiC prediction can overcome the limitation of existing Hi-C analytic methods

While InfoHiC is a Hi-C prediction method for cancer samples without Hi-C data, it still has advantages over analytic methods that were developed to detect neo-TADs or neo-loops from existing cancer Hi-C data (Xu et al, 2022; Wang et al, 2021). First, the existing methods are restricted to large SVs (>1 Mb) or translocations. However, SVs in cancer are characterized by local clusters within the 1-Mb range (Lee and Lee, 2021). Local SV clusters around *PRDM6* found in the patient with medulloblastoma are one

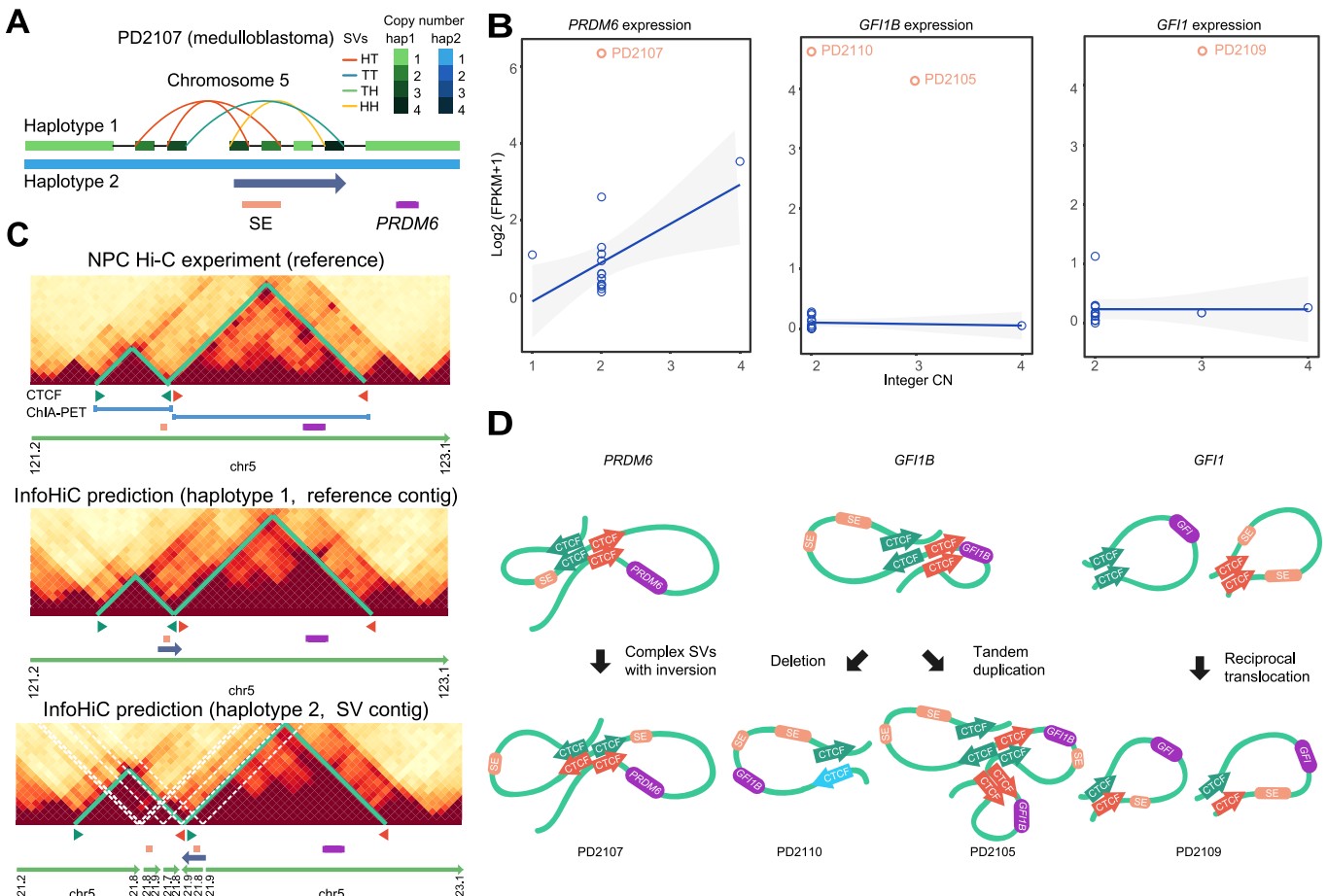

**Figure 6. Super-enhancer (SE) hijacking events of *PRDM6*, *GFI1B*, and *GFI1* found in patients with medulloblastoma.**

(A) A haplotype graph of chromosome 5 of the patient with medulloblastoma (PD2107). SVs were annotated with head-to-tail (HT), tail-to-head (TH), tail-to-tail (TT), and head-to-head (HH) orientations. (B) FPKM values of the *PRDM6*, *GFI1B*, and *GFI1* transcripts versus integer CNs measured in the brain tumor cohort. Linear regression lines are shown with the corresponding confidence interval ranges (blue), and the FPKM values of patients (PD2105, PD2107, PD2109, and PD2110) with SE hijacking events are shown above (apricot). (C) Hi-C experiment of neuro-progenitor cells (NPCs) and InfoHiC prediction of the 3D genome of patient PD2107. Reference TADs and neo-TADs are annotated (green) with CTCF motifs (arrowheads) near the boundaries, and *PRDM6* (purple) and SE (apricot) are shown below. The pairwise interactions of the CTCF ChIA-PET data are indicated by blue lines. (D) 3D genome models of the SE hijacking events of *PRDM6*, *GFI1B*, and *GFI1* derived from various types of SVs.

such case (Fig. 6). InfoHiC can be used to predict Hi-C data from all ranges of SVs, including local clusters in complex rearrangements. Second, to obtain contig Hi-C matrices, the existing methods use one-to-one mapping between Hi-C reads and an SV contig; Hi-C reads in the total Hi-C matrix are mapped into a single SV contig. However, the total Hi-C matrix in cancer has one-to-many mapping to multiple SV contigs; Hi-C reads in the total Hi-C matrix are the summation of Hi-C reads from multiple SV contigs. Because Hi-C reads from multiple SV contigs are mapped into a single SV contig in the existing methods, neo-TADs or neo-loops may not be clearly observed in the SV contig. In contrast, InfoHiC assumes one-to-many mapping; it predicts Hi-C reads from each SV contig first, and then predicts the total Hi-C matrix by merging Hi-C read counts from multiple SV contigs. The observed Hi-C matrix is used for validating the predicted total Hi-C matrix obtained from multiple SV contigs. With these advantages, InfoHiC can provide more chances to discover neo-TADs and neo-loops from SVs.

InfoHiC is a Hi-C prediction method, which is different from other neo-loop or neo-TAD finding methods that require existing Hi-C sequencing data (Wang et al, 2021). InfoHiC predicts Hi-C data for samples that have no Hi-C data. Nonetheless, to show the strength of InfoHiC for discovering neo-TADs and neo-loops, we compared our neo-TAD and neo-loop prediction results with previous NeoLoopFinder (a neo-loop finding tool) results on five breast cancer cell lines (Wang et al, 2021). First, we counted the total number of SVs used for neo-loop or neo-TAD findings. NeoLoopFinder only considered large SVs or translocations, and found a total of 479 SVs from the five cell lines. In contrast, InfoHiC found a total of 972 SVs. NeoLoopFinder reported a total of 524 neo-loop-involved genes and InfoHiC found 2325 neo-TAD-involved genes and 3738 neo-loop-involved genes across five cell lines. InfoHiC could find 73.3% of neo-loop-involved genes found by NeoLoopFinder in neo-TADs or neo-loops. In contrast, NeoLoopFinder could find 7% of neo-loop-involved genes found by InfoHiC. For example, we found the SE hijacking event of a

cancer-related gene, *GNA13* in BT474 (Appendix Fig. S5), but NeoLoopFinder could not detect it. InfoHiC and NeoLoopFinder used a different Hi-C resolution (40 kb in InfoHiC and 10 kb in NeoLoopFinder), and the 40-kb resolution may call more genes in a broader range than the 10-kb resolution. In addition, we observed that 55.3% of neo-loop-involved genes not found by NeoLoopFinder were associated with SVs <1 Mb, suggesting that small SVs discovered from WGS might reflect the difference between SE hijacking event findings. For a fairer comparison, we ran NeoLoopFinder (v0.4.3-r2) on 10 kb Hi-C data, and then fed 972 SVs including small SVs that we used for InfoHiC into NeoLoopFinder as inputs. As a result, NeoLoopFinder discovered 1,377 neo-loop-involved genes across the five cell lines. Overall, InfoHiC found 68% of neo-loop-involved genes from NeoLoopFinder, while NeoLoopFinder found 25% of neo-loop-involved genes from InfoHiC (Appendix Fig. S15). The remaining difference may be because even though we fed 972 SVs into NeoLoopFinder as inputs, NeoLoopFinder filtered out 51% of them (*n* = 492) in the SV assembly process. NeoLoopFinder measures distance decay of Hi-C contacts in the SV contig, and then filters out SVs if they are not consistent with the global distance decay (Wang et al, 2021). Sixty-two point eight percent of SVs filtered out in the assembly were <1 Mb, suggesting that Hi-C contacts from the small SVs might not be consistent with the global decay.

To further compare NeoLoopFinder and InfoHiC results, we investigated the intensities of NeoLoopFinder-specific, InfoHiC-specific, and shared events in cancer Hi-C experiments (Appendix Fig. S15). For each neo-loop-involved gene, we measured Hi-C intensities of gene-SE pairs. Even though high intensities do not necessarily represent neo-loop events, they can represent gene-SE interactions observable in cancer Hi-C experiments. HMEC and MCF10A were used as controls. In five cancer cell line Hi-C data (Appendix Fig. S15), shared events showed the highest intensities and NeoLoopFinder-specific events were the second highest in MCF7, SKBR3, and T47D. InfoHiC-specific events showed the lowest intensities in these cell lines, suggesting that they may contain false-positive predictions. However, in HCC1954 and BT474, InfoHiC-specific events were observed in higher intensities than NeoLoopFinder-specific events. The result suggested that NeoLoopFinder might not detect true events from HCC1954 and BT474 Hi-C data even though they had high intensities. Compared to analytic methods including NeoLoopFinder which require confident Hi-C observation from the corresponding SV, Hi-C prediction methods do not require Hi-C observation from SVs, and can discover neo-loops using the model trained on genomic regions with Hi-C observation. As a prediction method, InfoHiC could enable more sensitive neo-loop findings from SVs, including small or complex SVs where confident Hi-C observations are not available. InfoHiC-specific events predicted from HCC1954 and BT474 might be such cases. Moreover, we highlight that for patient-specific SVs that have not been observed in cell lines, the prediction method enables us to discover neo-loop from them as we showed in medulloblastoma cases.

## Discussion

In summary, we developed InfoHiC for cancer Hi-C prediction using a sequence-based model that enables the identification of neo-TAD and neo-loop formation from rearranged genomes. It can be trained by high-resolution Hi-C data of cancer cell lines, and can predict Hi-C matrix of other cancer samples without Hi-C data. The predicted Hi-C matrix on SV contigs can be validated using InfoHiC. Currently, there is no normalization method suitable for the cancer Hi-C matrix dealing with SV-derived contacts from multiple contigs. Therefore, instead of applying normalization methods to cancer Hi-C data to remove CNAs and SVs as biases, InfoHiC uses raw Hi-C data for training on the regions without SVs and performs contig-specific normalization after contig Hi-C prediction for SVs. Thus, InfoHiC provides normalized contig Hi-C matrices for SVs, which can be analyzed using other analytical tools developed for Hi-C data.

Compared with reference-based prediction models such as deepC (Schwessinger et al, 2020), Akita (Fudenberg et al, 2020), and Orca (Zhou, 2022), InfoHiC enables the use of cancer Hi-C data for training and validation, which takes advantage of the accumulated Hi-C datasets of cancer cell lines (Kim et al, 2021a). We showed that the model trained on the breast cancer cell line (T47D) predicted Hi-C data of other breast cancer cell lines well, which could be validated by the Hi-C data (Figs. 3 and 4). While the current study used Hi-C datasets of breast cancers, InfoHiC can be used to discover neo-TADs and neo-loops using Hi-C datasets of other cancer types (Kim et al, 2021a). In addition, applications of reference-based models (Schwessinger et al, 2020; Fudenberg et al, 2020; Zhou, 2022) are currently limited to simple deletions, duplications, or inversions; however, we showed that InfoHiC could predict Hi-C data from complex SVs, which are essential for neo-TAD and neo-loop findings in cancers. Moreover, in contrast to the reference-based models, InfoHiC utilizes WGS for cancer Hi-C prediction; (1) it performs CSCN encoding of SNPs and CNAs for each contig, (2) predicts Hi-C read counts based on the SV contig assembly, and (3) performs the contig-to-total Hi-C conversion from multiple SV contigs. InfoHiC trains the prediction model using a loss function newly defined for the total Hi-C matrix of cancer Hi-C data, and provides trained models for cancer Hi-C prediction from WGS (https://github.com/dmcb-gist/InfoHiC). With these availabilities, InfoHiC is applicable to the accumulated WGS datasets to uncover the impacts of SVs in cancer patients.

In the article, we showed comparison results between InfoHiC and deepC. When we adapted available reference-based CNN models for 3D cancer genome prediction, we also benchmarked Akita (Fudenberg et al, 2020). When we trained Akita using T47D Hi-C data, and tested on test set 1 of T47D, it showed a poor Pearson correlation of less than 0.3. The main difference between Akita and deepC is that deepC uses 1D CNN layers followed by fully connected layers with different weights per each genomic distance ("Methods and protocols"). In contrast, Akita uses 2D CNN layers, and there is no fully connected layer like in deepC, which can model Hi-C intensities depending on the genomic distance. Instead, in Akita, genomic distances are encoded in the 2D map before applying the 2D CNN layers (Fudenberg et al, 2020). Our initial benchmark showed that deepC could be successfully trained on cancer Hi-C data, and therefore, we adapted the deepC architecture for InfoHiC instead of Akita.

We showed that InfoHiC could predict cancer Hi-C matrices with a DSC >0.6 in the 1-Mb range. However, predicting long-range interactions is still challenging. When we tested using the breast cancer cell lines, the DSCs in the 1-Mb to 2-Mb range

decreased considerably (Figs. 2 and 3). The previously reported TAD sizes were about 560 kb, 880 kb, and 1480 kb for the 25th, median, and 75th quantile, respectively (Dixon et al, 2012). In the breast cancer cell lines that we analyzed, the quantile sizes of neo-TADs were similar with the previous report; 600 kb, 880 kb, and 1280 kb for the 25th, median, and 75th quantile, respectively. As more than 25% of neo-TADs were over the 1-Mb range, the current model might have to be scaled up to the mega-base range. Recently, Orca (Zhou, 2022) presented a hierarchical CNN model for the mega-base scale Hi-C prediction in noncancerous cell lines. The hierarchical model may be helpful to increase cancer Hi-C prediction performance in the 2-Mb range.

In the application study of pediatric patients with medulloblastoma, we showed that various types of SVs caused neo-TAD formation of the same *GFI1* families. In addition, the impact of complex SVs was uncovered by predicting Hi-C interaction from the SV contig. Previous approaches of SV analysis in noncoding regions include annotating juxtaposition events between genes and regulatory elements (Northcott et al, 2014) or boundary-affecting SVs from known TAD boundaries (Akdemir et al, 2020). However, SV analysis requires Hi-C evidence as chromatin interaction between DNA elements and TAD boundaries changes according to rearranged sequence contexts. InfoHiC annotates juxtaposition events (SE hijacking) and neo-TAD boundaries based on Hi-C prediction, providing more accurate results for analyzing noncoding SVs. Furthermore, the impact of complex SVs could not be investigated by previous approaches of enhancer-juxtaposing (Northcott et al, 2014) or boundary-affecting SV annotation, and InfoHiC provides Hi-C evidence for neo-TAD boundaries derived from complex SVs. Based on InfoHiC prediction, targeted therapy may be possible for noncoding driver SVs. For the example of medulloblastoma (PD2109) where *GFI1* was overexpressed by the reciprocal translocation t(1;5), inhibitors of *LSD1*, which is a cofactor of the *GFI1* transcription factor, can be used for targeted treatment. *LSD1* inhibitors such as GSK-LSD1 and ORY-1001 were shown to be effective for *GFI1*-activated medulloblastoma (Lee et al, 2019).

As an integrative work from 1D genome reconstruction to 3D genome prediction, we demonstrated that SVs have noncoding effects on the overexpression of cancer-related genes, which are associated with poor prognosis in BRCAs. We expect that the application of InfoHiC to other cancer types might improve the chances of identifying cancer drivers altered by noncoding SVs, especially for cancer types without driver SNVs and CNA events in coding regions. Furthermore, future research may identify cancer drivers in patients without common coding driver mutations, which can lead to personalized medication based on patient-specific SVs in the 3D genome context.

# Methods

## Reagents and tools table

| Reagent/resource | Reference or source | Identifier or catalog number |
| --- | --- | --- |
| **Software** | | |
| InfoHiC | This article https://github.com/DMCB-GIST/InfoHiC | |

| Reagent/resource | Reference or source | Identifier or catalog number |
| --- | --- | --- |
| InfoGenomeR | Lee and Lee (2021) https://github.com/dmcblab/InfoGenomeR | |
| DeepC | Schwessinger et al (2020) https://github.com/Hughes-Genome-Group/deepC | |
| NeoLoopFinder | Wang et al (2021) https://github.com/XiaoTaoWang/NeoLoopFinder | |
| BWA-MEM | Li (2013) https://bio-bwa.sourceforge.net | |
| 3DIV pipeline | Kim et al (2021a) https://github.com/kaistcbfg/3divv2 | |
| SpectralTAD | Cresswell et al (2020) https://github.com/dozmorovlab/SpectralTAD | |
| Peakachu | Salameh et al (2020) https://github.com/tariks/peakachu | |
| CovNorm | Kim and Jung (2021b) https://github.com/kaistcbfg/covNormRpkg | |
| Mutect | Cibulskis et al (2013) https://github.com/broadinstitute/mutect | |
| Platypus | Rimmer et al (2014) https://rahmanteamdevelopment.github.io/Platypus | |
| DELLY2 | Rausch et al (2012) https://github.com/dellytools/delly | |
| Manta | Chen et al (2016) https://github.com/Illumina/manta | |
| NovoBreak | Chong et al (2017) https://github.com/czc/nb_distribution | |
| ANNOVAR | Wang et al (2010) https://annovar.openbioinformatics.org/en/latest/ | |
| SAMtools | Li (2011) https://www.htslib.org/ | |
| BCFtools | Li (2011) https://www.htslib.org/ | |
| STAR | Dobin et al (2013) https://github.com/alexdobin/STAR | |
| Cufflinks | Trapnell et al (2010) https://github.com/cole-trapnell-lab/cufflinks | |
| PWMScan | Ambrosini et al (2018) https://sourceforge.net/projects/pwmscan/ | |
| **Other** | | |
| Maxwell RSC simply RNA Tissue Kit | Promega | AS1340 |
| Maxwell RSC Genomic DNA Kit | Promega | AS1880 |
| TruSeq Nano DNA Kit | Illumina | 20015965 |
| SureSelect$^{XT}$ RNA Direct Library Kit | Agilent | G7564A |
| Illumina Nova Seq 6000 | Illumina | |

## Methods and protocols

### *Reconstruction of genomic contigs*

Genomic contigs were obtained using InfoGenomeR from WGS data, which were used to construct a breakpoint graph to model the connectivity among genomic segments using SVs, CNAs, and SNP information and derived genomic contigs from Eulerian paths according to a minimum entropy approach (Lee and Lee, 2021). BWA-MEM (Li, 2013) was used for WGS read mapping to the GRCh37 reference. We used DELLY2 (Rausch et al, 2012), Manta

(Chen et al, 2016), and novoBreak (Chong et al, 2017) for the initial SV detection from Illumina paired-end data. Then, SV calls pre-detected from other WGS platforms (see "Data preprocessing" section) were merged into the initial SV detection. We used BIC-seq2 (Xi et al, 2016) for CNA detection and ABSOLUTE (Carter et al, 2012) for integer-CN detection (with cancer purity and ploidy estimation). SAMtools and BCFtools (Li, 2011) were used for SNP detection, and the haplotype-cluster model of BEAGLE (Browning and Browning, 2007) from the 1000 Genomes Project was used for haplotype phasing in InfoGenomeR.

### CSCN encoding

The genomic contigs were derived from SVs in haplotypes, and had different CN and SNP compositions from one another. Their sequences were represented by CSCN encoding, which was modified from one-hot encoding to represent the CNs and SNPs in a contig matrix. The column of the contig matrix represents each base index $j$ of the contig, whereas the row index $i$ represents the base composition, i.e., A, C, G, and T. One-hot encoding represents a zero or one binary for the base, and CSCN encoding multiplies the binary by the CN of the contig. We denote the genome $G$ as a set of genomic contigs $g$, with length $l$ and CSCN encoding function as $H$. The CN of the genomic contig was $\mu(g)$.

$$H(g) = (h_{ij}) \in \mathbb{Z}^{4 \times l}$$
$$h_{1j} = \mu(g) \text{ if } g_j \text{ is A, else 0.}$$
$$h_{2j} = \mu(g) \text{ if } g_j \text{ is C, else 0.}$$
$$h_{3j} = \mu(g) \text{ if } g_j \text{ is G, else 0.}$$
$$h_{4j} = \mu(g) \text{ if } g_j \text{ is T, else 0.}$$

### Hi-C data processing

We used the 3DIV pipeline for Hi-C read mapping, which previously established the first large-scale resource for cancer Hi-C (Kim et al, 2021a). Reads were mapped to the GRCh37 reference genome. The chimeric and self-ligated reads were filtered using the 3DIV pipeline. The contact matrix was obtained at a resolution of 40 kb. For the K562 cell line, the contact matrix was obtained at a resolution of 10 kb in addition to 40 kb. In InfoHiC, we used the raw contact matrix without implicit normalization for training and validation to preserve the Hi-C signals from genomic rearrangements. For comparison, we applied covNorm (Kim and Jung, 2021b), a normalization method included in the 3DIV pipeline, to the raw contact matrix for implicit normalization.

### Prediction of the total Hi-C matrix

We denote the convolutional neural network as CNN, which is composed of $\mathrm{CNN}_{\mathrm{deepSEA}}$ to extract chromatin features from DNA sequences (Zhou and Troyanskaya, 2015), a dilated convolution neural network $\mathrm{CNN}_{\mathrm{dilated}}$ to model a wide range of contexts for contact prediction (Schwessinger et al, 2020), and a fully connected layer $FC_d$ to predict the Hi-C contact for a genome distance $d$. The target value of the Hi-C interaction is $c(x, y)$, which represents the number of Hi-C reads mapped into the $x$- and $y$-coordinate bins in the Hi-C matrix. Here, we denote $c(x, y)$ as $c$. The reference version of deepC (Schwessinger et al, 2020) predicts $c$ using a single reference contig centered on $c$. In a reference model, the target $c$ is in the reference coordinate. We denote one-hot encoding based on the reference sequence as $R$.

$$c = \mathrm{FC}_d(\mathrm{CNN}(R(g)))$$

In contrast to the reference model, we predict $c$ as the sum of Hi-C contacts of multiple contigs, $g_1$, $g_2$, ..., $g_n$ with different genomic distances $d_1$, $d_2$, ... $d_n$ centered on target values $c_1$, $c_2$, ..., $c_n$ in the contig coordinates. When using different contigs rather than a reference sequence, we require a reference coordinate mapping function to change the contig coordinate prediction, which is denoted by ref.

$$c_i = \mathrm{FC}_{d_i}(\mathrm{CNN}(H(g_i)))$$
$$c = \sum_{i=1}^{n} \mathrm{ref}(c_i)$$

The ref $(c_i)$ function indicates the mapping from $c_i(x', y')$ on the $x'$ and $y'$ contig into $c_i(x, y)$ on the $x$ and $y$ reference coordinates. It returns $c_i$ if two 40-kb genomic bins ($x'$ and $y'$) targeting $c_i$ in the contig coordinate are maximally matched with two genomic bins ($x$ and $y$) targeting $c$ in the reference coordinate; otherwise, it returns 0. When Hi-C reads were mapped at a certain resolution, a genomic bin in the contig coordinates contained multiple portions of several bins in the reference coordinate. For example, a 40-kb genomic bin in the contig coordinate could contain 30 kb of bin 1 and 10 kb of bin 2 in the reference coordinate. In that case, we selected a maximum match (bin 1) to maintain the sharpness of the Hi-C data, which enables accurate TAD annotation.

For training multiple contigs with SV, there are issues compared to training the reference sequence. The reference prediction such as deepC (Schwessinger et al, 2020) uses a single one-hot encoding matrix to predict a vector of targets $c=(c^1, c^2, ... c^k)$ using a vector of genomic distances $d =(d^1, d^2, ... d^k)$ (a zigzag pole), simultaneously, rather than targeting a single scalar of $c^k$. However, for multiple contigs with SVs, prediction of the target vector is complicated. First, the genomic distances could differ when targeting $c^k$, the reference coordinate mapping should be included in the training. Third, tens of contigs could exist, and the graphics processing unit (GPU) memory would not be available for them. Thus, to simplify the training procedure for multiple contigs, we used genomic contigs in reference regions without SVs for training (breakpoint removal). The following are true for the reference regions: (1) genomic distances are the same for two haplotypes; (2) the ref function is not required because it is in the reference coordinate itself; (3) only two haplotype contigs are required to enable proper GPU memory usage. In addition, for the reference regions, contig-specific copy numbers are the same with haplotype-specific copy numbers as they have neither CNA nor SV breaks. For example, if a haplotype has three copies in the training window of the reference region, there is one genomic contig encoded by the haplotype copy number ($n = 3$). Simply, we target a vector of target $c=(c^1, c^2, ..., c^k)$ using two haplotype contigs, $g_1$ and $g_2$, with the genomic distance $d^1$, $d^2$, ..., $d^k$. Here, $c^j$ is the sum of the haplotype target values for each ($c_1^j$ and $c_2^j$).

$$c_1^j = \mathrm{FC}_{d_j}(\mathrm{CNN}(H(g_1)))$$
$$c_2^j = \mathrm{FC}_{d_j}(\mathrm{CNN}(H(g_2)))$$
$$c^j = c_1^j + c_2^j$$

Given the two haplotype contigs for training, we denote $\hat{c} = (\hat{c}^1, \hat{c}^2, ..., \hat{c}^k)$ as a zigzag pole prediction (Schwessinger et al, 2020) and minimized the mean square error (MSE) loss between $c$ and $\hat{c}$.

$$\text{Minimize } \frac{1}{k}\sum_{j=1}^{k}\left(c^j - \hat{c}^j\right)^2$$

### CSCN decoding

CSCN decoding was used for a comparison study with CSCN encoding when InfoHiC was trained by the cancer cell line (T47D). In addition, when InfoHiC was trained by the noncancerous cell lines (HMEC and MCF10A), we used CSCN decoding for the InfoHiC prediction model because CSCN encoding could not be trained by these cell lines. In CSCN decoding, haplotype contigs, $g_1$, $g_2$, ..., $g_n$ are encoded by the one-hot encodings of haplotype bases, $H_{\text{one-hot}}(g_1)$, $H_{\text{one-hot}}(g_2)$, ..., $H_{\text{one-hot}}(g_n)$, and prediction outputs are multiplied by the CN of each genomic contig.

$$c_i = \mu(g_i)\text{FC}_{d_i}\left(\text{CNN}\left(H_{\text{one-hot}}(g_i)\right)\right)$$
$$c = \sum_{i=1}^{n}\text{ref}(c_i)$$

### CNN architecture and training

We used a deepC architecture (Schwessinger et al, 2020) to predict Hi-C read counts per genome contig. DeepC was previously adapted from DeepSEA (Zhou and Troyanskaya, 2015) that employs hundreds of convolution filters to extract chromatin features. Specifically, five convolution layers (300, 600, 600, 900, and 900 hidden units) of kernel widths (8, 8, 8, 4, and 4 for each convolution layer) followed by ReLU activation, max-pooling widths (4, 5, 5, 5, and 2 for each convolution layer), and a dropout rate of 0.2 were used to extract the chromatin features. The layers were pretrained using chromatin profiling data (936 chromatin features). Then, 10 dilated gated convolutional layers (100 hidden units) with 3-width dilation schemes (rates 1, 2, 4, 8, 16, 32, 64, 128, 256, and 1 for each convolution layer) were used for long-range modeling, and the final fully connected layer output the contact value for each distance from the genomic contig. The Hi-C read counts from genomic contigs are summed into Hi-C read counts for the total Hi-C matrix.

The MSE loss between predicted and observed Hi-C read counts in the total Hi-C matrix was used for model training. We trained the model using the ADAM optimizer (learning rate = 0.001 and epsilon = 0.1), and applied the L2 normalization (regularization strength = 0.001). The model was trained on an NVIDIA Tesla V100 32GB GPU. The batch size was one, according to the GPU memory limit. The maximum training epoch was 10.

### Transfer learning and data augmentation

We used two transfer learning steps in this study. First, we used the pretrained weights of $\text{CNN}_{\text{deepSEA}}$ from deepC, which was trained using input 1-kb DNA sequences targeting 936 chromatin features. We then transferred the pretrained weights of $\text{CNN}_{\text{deepSEA}}$ and trained the convolutional neural network $\text{CNN} = \text{CNN}_{\text{dilated}} \circ \text{CNN}_{\text{deepSEA}}$ for the total Hi-C matrix prediction using 1-Mb + 40-kb genomic windows at a resolution of 40 kb, targeting a vector of 25 contact values, $c = (c_1, c_2, ...c_{25})$. Then, we transferred the trained weights from 1-Mb + 40-kb genomic windows to the 2-Mb prediction model,

targeting a vector of 49 contact values, $c = (c_1, c_2, ...c_{49})$. We excluded self-interaction (zero distance interaction in a single 40-kb bin) because raw Hi-C data have high values in self-interaction that prevented the learning of long-range interactions.

We then used augmentation processes including a 20-kb shift and reverse complement training. For the 20-kb shift, we mapped the Hi-C reads to +20-kb genomic bins. In detail, Hi-C reads were mapped to the 1-kb to 40-kb coordinate in the first mapping process, and Hi-C reads were mapped to the 21-kb to 60-kb coordinate in the second mapping, producing a doubled training data set. We trained the model using the forward strand for the original Hi-C and the reverse strand for the flipped Hi-C. For testing, we averaged the predictions of the forward and reverse strands.

For training at a resolution of 10 kb, the convolution neural network network $\text{CNN} = \text{CNN}_{\text{dilated}} \circ \text{CNN}_{\text{deepSEA}}$ was trained on 1-Mb + 20-kb genomic windows, targeting a vector of 101 contact values, $c = (c_1, c_2, ...c_{101})$. The self-interaction in a single 10-kb bin was excluded for training. The augmentation processes were performed using a 10-kb shift, after mapping the Hi-C reads to +10-kb genomic bins. Reverse complementary training was performed at a resolution of 10 kb, and predicted vectors from the forward and reverse strands were averaged into the output vector.

### Training using noncancerous cell lines

We compared InfoHiC, deepC, and deepC-SV by training using noncancerous breast epithelial cell lines (HMEC and MCF10A), where performances of deepC and deepC-SV were measured before and after implicit normalization of raw Hi-C data (Appendix Table S2). For training using HMEC and MCF10A, InfoHiC used the reference sequence for training raw Hi-C data and performed Hi-C prediction on SV contigs using CSCN decoding, as CSCN encoding could not be trained by noncancerous cell lines, where most genomic regions had a haplotype copy number one. DeepC used the reference sequence for training as well, and performed Hi-C prediction using the one-hot encoding model. For comparison of prediction performance in SV regions, deepC-SV performed Hi-C prediction on the DNA sequence derived from each SV call of InfoGenomeR.

### Contig-specific normalization

Normalization methods commonly used for the Hi-C matrix, such as the iterative correction and eigenvector decomposition (ICE) (Imakaev et al, 2012), consider CNAs and SVs as biases. ICE normalization was employed to implicitly normalize the GC content bias, mappability bias, and other experimental noise together with CNA and SV biases by introducing biases $b_i$ and $b_j$ for each $i$th and $j$th bin. Here, $c(i, j)$ is an observation of the total Hi-C contact, $t(i, j)$ is the normalized Hi-C contact, and $N$ is the number of bins.

$$c(i, j) = b_i b_j t(i, j)$$
$$\sum_{i=1, |i-j|>1}^{N} t(i, j) = 1$$

When using the total Hi-C observation $c(i, j)$ for cancer Hi-C, it can result in inaccurate production of a normalized matrix for the reference sequence. In addition, Hi-C contacts from CNAs and SVs

should not be removed to discover 3D genome organization for rearranged contigs. Previously, we obtained $\hat{c}$ for $n$ assembled contigs.

$$c(i,j) = \sum_{k=1}^{n} \hat{c}_k$$

Here, we applied ICE normalization to $\hat{c}_k$, which is an assembled Hi-C contact free from CNA and SV biases. Here, $i'$ and $j'$ are indices in contig coordinates for each $k$ contig. The Iced Python module was used for contig-specific ICE normalization.

$$\hat{c}_k(i',j') = b_{ki'}b_{kj'}\hat{t}_k(i',j')$$

$$\sum_{i'=1,|i'-j'|>1}^{N'} \hat{t}_k(i',j') = 1$$

### InfoHiC models used for Hi-C prediction

We used different InfoHiC models depending on each cancer type. For breast cancer cell lines (T47D, BT474, MCF7, HCC1954, and SKBR3), we used the 2-Mb CSCN encoding model transferred from the 1-Mb CSCN encoding model, which was trained by T47D Hi-C data. For BRCA datasets, we used the 2-Mb CSCN encoding model as well, which was used for breast cancer cell lines. For patients with medulloblastoma, we trained the 1-Mb CSCN decoding model using Hi-C data of the NPC cell line, and then we fine-tuned the model for each medulloblastoma driver gene (*GFI1*, *GFI1B*, and *PRDM6*); the 1-Mb CSCN decoding model was trained with an additional round of ten iterations with the 4-Mb window centered on each gene. The fine-tuned models showed better neo-TAD annotation results on each gene than the NPC cell line model without fine-tuning (Appendix Fig. S16), and they were used for medulloblastoma sample analysis: *PDRM6* for PD2107, *GFI1B* for PD2105 and PD2110, and *GFI1* for PD2109, respectively.

### Neo-TAD and neo-loop annotation

We annotated the TADs and loops on the normalized contig Hi-C matrix $\hat{T}_k = (\hat{t}_k(i',j'))$ using spectralTAD (Cresswell et al, 2020) and Peakachu (Salameh et al, 2020), respectively. We hierarchically obtained primary to tertiary TADs using spectral clustering (Cresswell et al, 2020). Neo-loops were annotated using the CNN-based method Peakachu (Salameh et al, 2020) on the contig Hi-C obtained by InfoHiC, which is free of CNA and SV biases and does not require any further steps of CN normalization or pseudo-assembly of SV-derived contacts (Wang et al, 2021). A TE or SE hijacking event of a gene in the neo-TAD was defined as follows: (1) the enhancer and the gene did not co-exist in the reference TAD prediction (with a ± 40-kb offset to adjust errors of TAD boundaries) and (2) interaction between the enhancer and the gene was predicted to exist in the SV window. TEs and SEs were downloaded from SEdb (Jiang et al, 2019). We used tissue-specific annotations depending on cancer types, which are summarized in Appendix Table S11.

### Sample preparation

For brain tumors, macro-dissection was performed on tumor areas with a tumor cell content of more than 90% stained with hematoxylin-eosin. Genomic DNA and total RNA were extracted

from the freshly frozen tissues and peripheral blood of patients using the Maxwell RSC Genomic DNA and simplyRNA Tissue Kit (Promega, Madison, WI, USA), respectively. The WGS library was prepared using the TruSeq Nano DNA Kit (Illumina, San Diego, CA, USA) and sequenced using the Illumina NovaSeq6000 platform. The RNA-seq library was prepared using the SureSelectXT RNA Direct Library Kit (Agilent Technologies, Santa Clara, CA, USA) and sequenced using the Illumina NovaSeq6000 platform. The study on brain tumors was approved by the Institutional Review Board of Seoul National University Hospital (IRB No:1905–108-1035). The experiments conformed to the principles set out in the WMA Declaration of Helsinki and the Department of Health and Human Services Belmont Report.

### Data processing

For breast cancer cell lines, WGS fastq files of the T47D, MCF7, SKBR3, and K562 cell lines were downloaded from the SRA: T47D (PRJNA380394), MCF7 (PRJNA486532), SKBR3 (PRJNA476239), and K562 (PRJNA380394). Paired-end WGS reads from the T47D, MCF7, SKBR3, and K562 cell lines were mapped to the human reference genome (GRCh37) using BWA-MEM (Li, 2013) with default parameters (version 0.7.15). WGS bam files of the HCC1954 and BT474 cell lines were downloaded from TCGA in the GDC data portal and CCLE in Google Cloud Storage, respectively. For patients with breast cancer from TCGA, WGS bam files were downloaded from dbGaP (accession code phs000178.v11.p8). For brain tumors, paired-end WGS reads (150 bp) were mapped GRCh37 using BWA-MEM (Li, 2013) with default parameters (version 0.7.15).

From WGS bam files, somatic SVs were detected by DELLY2 (Rausch et al, 2012), Manta (Chen et al, 2016), and novoBreak (Chong et al, 2017). We merged SV calls from WGS with those from long reads for HCC1954 and SKBR3. SV calls of HCC1954 from linked-read sequencing data (10X Genomics) were downloaded from https://cf.10xgenomics.com/samples/genome/HCC1954T_WGS_210/HCC1954T_WGS_210_large_svs.vcf.gz. SV calls for SKBR3 from long-read sequencing data (PacBio) (Nattestad et al, 2018) were downloaded from http://labshare.cshl.edu/shares/schatzlab/www-data/skbr3/reads_lr_skbr3.fa_ngmlr-0.2.3_mapped.bam.sniffles1kb_auto_l8_s5_noalt.vcf.gz. Somatic SNVs and indels were detected using Mutect (Cibulskis et al, 2013) and Platypus (Rimmer et al, 2014) respectively. Variants were annotated using ANNOVAR (Wang et al, 2010).

Hi-C fastq files of breast cancer cell lines were downloaded from the sequencing read archive (SRA); MCF7, BT474, and SKBR3 Hi-C data (accession codes PRJNA430222); HCC1954 (accession code PRJNA479882); and T47D (PRJNA438511), respectively. Hi-C fastq files of the K562 and NPC cell line were downloaded from the SRA (accession code PRJNA268125 and PRJNA798046, respectively). Hi-C reads were mapped to GRCh37 using the 3DIV pipeline (Kim et al, 2021a). CTCF ChIP-seq data of NPC cells and ChIA-PET data of a neuroblastoma cell line were downloaded from ENCODE (accession code ENCSR125NBL and ENCSR514HBO, respectively).

For breast cancer cell lines, RNA-seq gene expression data were downloaded from the Cancer Cell Line Encyclopedia (https://sites.broadinstitute.org/ccle/datasets). For patients with breast cancer from TCGA, RNA-seq gene expression data were

downloaded from the GDC data portal (https://gdc.cancer.gov/access-data/gdc-data-portal). For brain tumors, paired-end RNA-seq reads (101 bp) were mapped to GRCh37 using STAR (Dobin et al, 2013), and gene expression was quantified using Cufflinks (Trapnell et al, 2010).

## Data availability

The datasets and computer code produced in this study are available in the following databases: InfoHiC: GitHub (https://github.com/dmcb-gist/InfoHiC); WGS and RNA-seq fastq files and variant calls (SNVs, indels, CNAs, and SVs) from pediatric patients with medulloblastoma: European Genome-phenome Archive EGAD00001009507; Gene expression profiles from pediatric patients with medulloblastoma: Zenodo (https://doi.org/10.5281/zenodo.10544039).

The source data of this paper are collected in the following database record: biostudies:S-SCDT-10_1038-S44320-024-00065-2.

## Peer review information

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

## Acknowledgements

This work was supported by an Institute of Information & Communications Technology Planning & Evaluation (IITP) grant funded by the Korean government (MSIT) (No. 2019-0-00567, Development of Intelligent SW Systems for Uncovering Genetic Variation and Developing Personalized Medicine for Cancer Patients with Unknown Molecular Genetic Mechanisms, No.2019-0-01842, Artificial Intelligence Graduate School Program (GIST)). The brain tumor biospecimens and data used in this study were provided by the Human Biobank of Seoul National University Hospital, a member of Korea Biobank Network (KBN4_A03), and Seoul National University Hospital Cancer Tissue Bank. All samples derived from the Biobanks of SNUH were obtained with informed consent under institutional review board-approved protocols.

## Author contributions

**Yeonghun Lee**: Conceptualization; Software; Validation; Visualization; Methodology; Writing—original draft. **Sung-Hye Park**: Resources; Data curation; Investigation; Writing—review and editing. **Hyunju Lee**: Conceptualization; Supervision; Funding acquisition; Investigation; Methodology; Project administration; Writing—review and editing.

Source data underlying figure panels in this paper may have individual authorship assigned. Where available, figure panel/source data authorship is listed in the following database record: biostudies:S-SCDT-10_1038-S44320-024-00065-2.

## Disclosure and competing interests statement

The authors declare no competing interests.

