## [Peer Review File · Molecular Systems Biology]

Prediction of the 3D cancer genome from whole genome sequencing using InfoHiC

Yeonghun Lee, Sung-Hye Park, and Hyunju Lee

Corresponding author(s): Hyunju Lee (hyunjulee@gist.ac.kr)

Review Timeline:

Submission Date:	22nd Jul 23
Editorial Decision:	15th Sep 23
Revision Received:	30th Jan 24
Editorial Decision:	11th Mar 24
Revision Received:	28th Jun 24
Editorial Decision:	18th Jul 24
Revision Received:	3rd Sep 24
Accepted:	9th Sep 24

Editor: Poonam Bheda

Transaction Report:

15th Sep 2023

Manuscript Number: MSB-2023-11897

Title: Prediction of the 3D cancer genome from whole genome sequencing using InfoHiC

Author: Yeonghun Lee

Sung-Hye Park

Hyunju Lee

Dear Dr. Lee,

Thank you again for submitting your work to Molecular Systems Biology. We have now heard back from the three reviewers who agreed to evaluate your study. As you will see below, the reviewers appreciate that the presented approach addresses a relevant problem. However, they raise a series of concerns, which we would ask you to address in a major revision.

I think that the recommendations of the reviewers are rather clear and I therefore do not see the need to repeat the comments listed below. One of the more fundamental points raised by Reviewer 2 is concerns the documentation and installation of the package. In particular:

- InfoHiC needs to be provided as a package that can be easily installed and used by future users and can be combined with existing workflows. Improved documentation including a guide on how to use the method would help make the InfoHiC easily accessible.

All other issues raised would need to be satisfactorily addressed. Please let me know in case you would like to discuss in further detail any of the issues raised, I would be happy to schedule a call.

We require:

4) A .docx formatted letter INCLUDING the reviewers' reports and your detailed point-by-point responses to their comments. As part of the EMBO Press transparent editorial process, the point-by-point response is part of the Review Process File (RPF), which will be published alongside your paper.

5) A complete author checklist, which you can download from our author guidelines (<https://www.embopress.org/page/journal/17574684/authorguide#submissionofrevisions>). Please insert information in the checklist that is also reflected in the manuscript. The completed author checklist will also be part of the RPF.

6) Please note that all corresponding authors are required to supply an ORCID ID for their name upon submission of a revised manuscript.

7) It is mandatory to include a 'Data Availability' section after the Materials and Methods. Before submitting your revision, primary datasets produced in this study need to be deposited in an appropriate public database, and the accession numbers and database listed under 'Data Availability'. Please remember to provide a reviewer password if the datasets are not yet public (see <https://www.embopress.org/page/journal/17574684/authorguide#dataavailability>).

In case you have no data that requires deposition in a public database, please state so in this section. Note that the Data Availability Section is restricted to new primary data that are part of this study. This study includes no data deposited in external repositories.

8) For data quantification: please specify the name of the statistical test used to generate error bars and P values, the number

(n) of independent experiments (specify technical or biological replicates) underlying each data point and the test used to calculate p-values in each figure legend. The figure legends should contain a basic description of n, P and the test applied. Graphs must include a description of the bars and the error bars (s.d., s.e.m.). Please provide exact p values.

<https://www.embopress.org/page/journal/17574684/authorguide#expandedview>

11) For more information: There is space at the end of each article to list relevant web links for further consultation by our readers. Could you identify some relevant ones and provide such information as well? Some examples are patient associations, relevant databases, OMIM/proteins/genes links, author's websites, etc...

12) Author contributions: CRediT has replaced the traditional author contributions section because it offers a systematic machine readable author contributions format that allows for more effective research assessment. Please remove the Authors Contributions from the manuscript and use the free text boxes beneath each contributing author's name in our system to add specific details on the author's contribution. More information is available in our guide to authors.

13) Disclosure statement and competing interests: We updated our journal's competing interests policy in January 2022 and request authors to consider both actual and perceived competing interests. Please review the policy <https://www.embopress.org/competing-interests> and update your competing interests if necessary.

14) Every published paper now includes a 'Synopsis' to further enhance discoverability. Synopses are displayed on the journal webpage and are freely accessible to all readers. They include a short stand first (maximum of 300 characters, including space) as well as 2-5 one-sentences bullet points that summarizes the paper. Please write the bullet points to summarize the key NEW findings. They should be designed to be complementary to the abstract - i.e. not repeat the same text. We encourage inclusion of key acronyms and quantitative information (maximum of 30 words / bullet point). Please use the passive voice. Please attach these in a separate file or send them by email, we will incorporate them accordingly.

Share synopsis text and image, as well as eTOC:

Please note that these would be the final versions and changes during proofing are usually not allowed

15) As part of the EMBO Publications transparent editorial process initiative (see our Editorial at <http://embomolmed.embopress.org/content/2/9/329>), Molecular Systems Biology Medicine will publish online a Review Process File (RPF) to accompany accepted manuscripts.

In the event of acceptance, this file will be published in conjunction with your paper and will include the anonymous referee reports, your point-by-point response and all pertinent correspondence relating to the manuscript. Let us know whether you agree with the publication of the RPF and as here, if you want to remove or not any figures from it prior to publication.

Molecular Systems Biology has a "scooping protection" policy, whereby similar findings that are published by others during review or revision are not a criterion for rejection. Should you decide to submit a revised version, I do ask that you get in touch after three months if you have not completed it, to update us on the status.

I look forward to receiving your revised manuscript.

Yours sincerely,

Poonam Bheda, PhD
Scientific Editor
Molecular Systems Biology

Reviewer #1:

In this manuscript, the authors develop a tool InfoHiC, specifically designed for 3D genomic contact prediction in cancers. Different from current universal prediction tools, the InfoHiC uses the SV information and haplotype copy number and predicts contig and total HiC matrices. By comparing the performances of InfoHiC and DeepC, the authors showed that InfoHiC has better prediction accuracy in breast cancer cell lines. InfoHiC also predicts neoTAD and super-enhancer hijacking events, those are associated with aberrant gene activation and prognosis in breast cancer and medulloblastoma. This tool is potentially an effective complement to 3D genome prediction in cancer studies.

Major points:

1. The robustness and availability are critical for the InfoHiC application in different types/subtypes of cancer. The authors show that the training model from cancer cell line T47D performs similarly but better than the model trained from normal epithelial cells HMEC and MCF10A. Therefore, when training data from cancer is unavailable, users can use model trained from normal tissue instead. This is a critical point. The authors should also show how many neo-TAD and SE hijacking events predicted in Figure 5 can also be reproduced using models from HMEC and MCF10A.
2. The authors show that InfoHiC outperforms DeepC in training datasets. It is also good to show if this is the same in external datasets and patient samples. For example, the authors should also show if DeepC can also predict the experiment-validated neo-TAD and SE hijacking events in Fig. S2. Similarly, it would be interesting to show if the MYO1D overexpression-associated SE hijacking events in Figure 4 can be identified by DeepC.
3. In Fig. 2&3, the authors clearly show that the prediction accuracy drops quickly when the distance increases. It should be at least discussed the relationship of prediction accuracy of neo-TAD and neo-loops against the size of TAD distance.

Minor points:

1. Supplementary Fig. 1 is not cited in the manuscript.
2. The title of axis in Fig. 2 are not clearly labeled.
3. The p-values or statistical significance should be presented in Figure 4b and Figure 5b.

Reviewer #2:

In this manuscript, Lee et al. introduce InfoHiC, a novel method for predicting Hi-C contacts using whole genome sequencing data in cancer. The approach comprises two main steps: 1) Identifying and assembling cancer-specific genomic contigs using their previously published method InfoGenomeR. 2) Employing the well-established CNN-based framework DeepC (Schwessinger et al. Nature Methods. 2020), which was initially designed for predicting Hi-C contacts from DNA sequences in normal cells, to predict Hi-C contacts specific to each contig identified in the first step. The authors demonstrate the utility of InfoHiC by applying it to datasets from patients with breast cancer and medulloblastoma. The results underscore its potential in uncovering the impact of non-coding SVs on the regulation of oncogene transcription. While the core concept is intriguing, the unique contributions of this work require further clarification. Specific concerns are detailed below.

Major concerns:

- 1) The authors need to clarify the unique contributions of this work. At present, it seems InfoHiC is just a pipeline that combines InfoGenomeR and DeepC, with minor alterations in encoding. Additionally, a pertinent concern arises from the observation that there is no significant difference between deepC-SV and InfoHiC in supplementary figure 1.
- 2) In the original DeepC paper, a particular procedure is employed to normalize the Hi-C data by genomic distance before model training. However, it seems that the authors of this paper trained both the InfoHiC and DeepC models using raw Hi-C signals. Could the authors clarify if any adaptations or modifications have been made in this aspect?
- 3) Regarding to the HSCN encoding, there is a discrepancy between the scheme depicted in figure 1a and the descriptions provided in the Method section. According to the Method section, each value in the contig matrix represents the copy number of the corresponding DNA base (A/T/C/G). However, in figure 1a, contig 2 is elongated due to a duplication event, resulting in its extension beyond the lengths of contig 1 and contig 3. If such an elongation (or shrinkage) occurs for every contig experiencing copy number variations, it follows that each column in the encoding should ideally reflect a copy number of 1, rather than the original copy number value estimated from whole-genome sequencing.

- 4) The authors benchmarked multiple variants of InfoHiC models in figures 2-3 and supplementary figure 1, and showed that the best performance is achieved by the 2-Mb model with the 1-Mb model transferred. However, there remains ambiguity regarding the specific model employed for the subsequent analyses. Specifically, for patients with breast cancer, it is not clear whether the model employed was trained on T47D Hi-C data or noncancerous cell lines. Similarly, in the case of patients with medulloblastoma, details regarding the training data for the applied model are needed.
- 5) All the training and predictions performed in this work were limited to a relatively low resolution of 40kb. While this resolution is appropriate for predicting TADs/neo-TADs, it may not be optimal for accurately predicting neo-loops or enhancer hijacking. How about the performance of InfoHiC at higher resolution such as 10kb or 5kb?
- 6) The method by which super enhancers and typical enhancers were defined is not mentioned.
- 7) In the introduction, the authors state that "Third, the existing methods cannot utilize cancer Hi-C data; 1) even though Hi-C data of numerous cancer types have been accumulated, they cannot be used for training and 2) Hi-C prediction from SVs cannot be validated using cancer Hi-C data, preventing researchers from verifying neo-TAD predictions". This appears to be inaccurate. If this is the case, it raises questions about how the authors conducted their own comparison of methods using cancer Hi-C data. The second point is even more confusing, as it is feasible to compare the predictions against the observed Hi-C contacts by visually juxtaposing contact maps around SVs. This practice is exemplified in Figure 4 from Tan J et al. (Nat Biotechnol 2023, <https://doi.org/10.1038/s41587-022-01612-8>).
- 8) As a method paper, it is essential for the authors to provide detailed guidelines that ensure successful installation and execution of the software. The current documentation solely offers a list of required packages for the installation, but my attempt to install those packages failed on my machine. This issue arises from a possible conflict where tensorflow=1.14 necessitates Python 3.7, while the neoloop package mandates Python version 3.8 or higher.

Minor concerns:

- 1) There is a discrepancy between figure 1a and its corresponding figure caption. The caption specifies "haplotype 1 in blue and haplotype 2 in green", however, the haplotype 1 is actually drawn in green, and the haplotype 2 is in blue, based on the figure.
- 2) In Figure 2, the x-axis labels are missing.
- 3) Based on my comprehension, the distance-stratified correlation (DSC) consists of a series of correlation values, each corresponding to a specific genomic distance. I'm interested in understanding how the DSC values in the second paragraph of the section "InfoHiC outperforms the reference-based CNN model" were computed. Are these values simply the averages taken across all genomic distances?
- 4) On page 6, I was confused about the 1-Mb and 2-Mb models mentioned in the final paragraph. This puzzlement arises from the fact that both training and validation were conducted using 4-Mb reference genomic windows, as indicated in the preceding paragraph on the same page.
- 5) It would be beneficial and easier for readers to understand by adding panel labels to supplementary figures, just as the authors did for the main figures.
- 6) I was confused by one of the criteria used for TE/SE hijacking annotation within a neo-TAD "an enhancer did not exist in the reference TAD prediction" (on page 25). Did the authors mean that the enhancer and the gene should not co-exist in the reference TAD prediction?

Reviewer #3:

The authors have proposed a computational framework called InfoHiC, which starts from whole-genome sequencing (WGS) data and can predict the chromatin contact maps of cancer samples using convolutional neural networks. They demonstrated that InfoHiC can produce chromatin contact maps similar to real Hi-C experiments and can correct aberrant chromatin conformation caused by structural variations. As a result, it can identify newly formed topologically associated domains and chromatin loops in cancer samples. They provided some examples of enhancer-hijacking events to illustrate the correlation between the formation of new chromatin structures and the abnormal expression of key genes in cancer. We believe that this work has the potential to advance our understanding of the three-dimensional genome in cancer research.

Major issues:

1. The InfoHiC model incorporates a pre-trained model from deepC and further extends it by adding some dilated convolutional layers. The authors also employ various strategies to retrain the entire model with both cancer Hi-C data and WGS data. In their method comparisons, they mainly compare InfoHiC with deepC. However, the authors did not provide a detailed description of how they utilized deepC in their method comparisons. It remains unclear whether they directly employed the pre-trained deepC model or retrained deepC with the cancer data. Besides, the main models of InfoHiC and deepC are very similar. There are also other methods that can predict 3D genome folding from DNA sequence, such as Akita as mentioned. The authors should compare InfoHiC with these methods.
2. The authors demonstrated that InfoHiC can generate chromatin contact maps similar to those from cancer Hi-C experiments using whole-genome sequencing (WGS) data and can also correct the maps in SV regions. They also mentioned another method, NeoLoopFinder, which can identify chromatin interactions induced by SVs from Hi-C contact maps and reconstruct local Hi-C maps around breakpoints. NeoLoopFinder can also be used to discover new loops and TADs and identify enhancer-

hijacking events. After all, correcting Hi-C maps affected by SVs to their correct versions seems to be a very crucial aspect of cancer three-dimensional genomics research. InfoHiC and NeoLoopFinder approach this issue from different perspectives. The authors may further discuss the similarities and differences between these two methods, as well as their respective advantages. The authors should compare the performance of these two methods in reconstructing accurate Hi-C maps, discovering new loops, and identifying enhancer hijacking events.

Minor issues:

1. I am confused about the statement on page 9, "In those cases, a Hi-C contact in the cancer Hi-C matrix was not unique to a single TAD, and neo-TADs were overlapped with Hi-C contacts from other TADs". We think a Hi-C contact, which is a pixel in the Hi-C map, either falls within one TAD or doesn't. How could it possibly overlap with multiple TADs? Thus, I can't understand the meaning of Fig 4.a. The authors may give more explanations.
2. The authors compare the expression of genes in neo-TADs or neo-loops with genes in reference TADs and loops in Fig4 and Fig5. However, they didn't provide a detailed description of how they defined the reference TADs or loops.
3. The authors seem to produce the WGS data and RNA-seq data of brain tumors on page 25 "Sample preparation and data preprocessing". However, they didn't provide the accession number of these data in "Availability of data and materials" section.
4. In the legend of Supplementary Fig. 1. The statement "according to the mega-base scale distance range (0-2 Mb)", We think it should be 0-1Mb.
5. The authors should polish the writing further.

Answers to Reviewer #1:

In this manuscript, the authors develop a tool InfoHiC, specifically designed for 3D genomic contact prediction in cancers. Different from current universal prediction tools, the InfoHiC uses the SV information and haplotype copy number and predicts contig and total HiC matrices. By comparing the performances of InfoHiC and DeepC, the authors showed that InfoHiC has better prediction accuracy in breast cancer cell lines. InfoHiC also predicts neoTAD and super-enhancer hijacking events, those are associated with aberrant gene activation and prognosis in breast cancer and medulloblastoma. This tool is potentially an effective complement to 3D genome prediction in cancer studies.

Major points:

1. The robustness and availability are critical for the InfoHiC application in different types/subtypes of cancer. The authors show that the training model from cancer cell line T47D performs similarly but better than the model trained from normal epithelial cells HMEC and MCF10A. Therefore, when training data from cancer is unavailable, users can use model trained from normal tissue instead. This is a critical point. The authors should also show how many neo-TAD and SE hijacking events predicted in Figure 5 can also be reproduced using models from HMEC and MCF10A.

Answer) Thank you for the comment. When Hi-C data of a certain type of cancer is not available, InfoHiC can be trained using Hi-C data from the normal tissue and be used for Hi-C prediction on cancer samples. We showed the performance of InfoHiC trained by the normal tissues (HMEC and MCF10A) at the training on noncancerous cell lines section in the manuscript. In this revision, to show that InfoHiC trained by the normal tissues can be used for the neo-TAD and SE hijacking event prediction, we added the results in the main manuscript as follows.

Page 9-10) In addition, we investigated whether the models trained by epithelial cell lines (HMEC and MCF10A) can reproduce the BRCA prediction results from the T47D-trained model. First, we measured the correlation between predicted Hi-C matrices from the epithelial and T47D cell line. The epithelial cell line model showed 0.928 (HMEC vs T47D) and 0.960 (MCF10A vs T47D) average Pearson's R on raw Hi-C prediction. The average DSCs were 0.829 (HMEC vs T47D) and 0.810 (MCF10A vs T47D) on raw Hi-C prediction, and after contig-specific normalization, the DSCs increased into 0.920 (HMEC vs T47D) and 0.921 (MCF10A vs T47D). Next, based on contig normalized Hi-C matrices, we performed neo-TAD prediction and measured reproducibility to determine how many enhancer hijacking events predicted by the T47D model could be reproduced by the epithelial cell line model. To quantify enhancer hijacking events, we counted all gene-SE pairs existing in the predicted neo-TADs from BRCA datasets. The total numbers of gene-SE pairs predicted by the T47D model from BRCA datasets were 44,365 in the 0 to 1-Mb range and 28,941 in the 0 to 0.5-Mb range. In the 0 to 1-Mb range, the HMEC and MCF10A models predicted 73.9% ($n =$

32,782) and 73.8% (n = 32,760) of 41,820 gene-SE pairs, and in the 0 to 0.5-Mb range, they predicted 83.4% (n = 24,128) and 83.2% (n = 24,069), respectively. While the DSC between Hi-C matrices from the epithelial and cancer models was greater than 0.9, reproducibility of gene-SE pair prediction was decreased compared with that of DSC, because gene-SE pair prediction depended on neo-TAD annotation, which in turn depended on predicted Hi-C matrices. Gene-SE pair prediction was also sensitive to the gene-SE pair distance, and the epithelial models could reproduce a higher fraction of gene-SE pair prediction results from the T47D model in the 0.5-Mb range than those in the 1-Mb range.

2. The authors show that InfoHiC outperforms DeepC in training datasets. It is also good to show if this is the same in external datasets and patient samples. For example, the authors should also show if DeepC can also predict the experiment-validated neo-TAD and SE hijacking events in Fig. S2. Similarly, it would be interesting to show if the MYO1D overexpression-associated SE hijacking events in Figure 4 can be identified by DeepC.

Answer) Thank you for the comment. We showed that InfoHiC showed the better correlation with the experimental truths than deepC at the external test on breast cancer cell lines section in the manuscript. According to the reviewer's comment, when we performed Hi-C prediction using deepC-SV, we found that deepC-SV could also predict the SE hijacking events of MYC, PVT1, and GNA13 shown in Fig. S2 (changed as Fig. S5 in the revised manuscript). In addition, deepCSV could predict the MYO1D SE hijacking events in Figure 4 (changed as Figure 5 in the revised manuscript). However, in this revision, we further investigated whether InfoHiC could predict neo-TAD and SE hijacking events better than deepC-SV by comparing all SE hijacking events found in InfoHiC and deepC-SV. We added the results in the main manuscript as follows.

Page 8)

In addition, we investigated whether deepC-SV could predict InfoHiC results of SE hijacking events. The number of SE-hijacked genes predicted by InfoHiC ranged from 77 to 562 in the five breast cancer cell lines (total n = 1,567). Because the SE-hijacked genes could interact with multiple SEs, we additionally counted all pairs of genes and SEs, which ranged from 257 to 2,867 (total n = 7,879). DeepC-SV could predict 85.2% (1,335 / 1,567) of SE-hijacked genes and 63.0% (4,966 / 7,879) of gene-SE pairs. Among 17 cancer-related genes involved with SE hijacking events predicted by InfoHiC (Appendix Table S3), deepC-SV could predict SE hijacking events of 14 cancer-related genes (82.4%, 14/17). However, deepC-SV could not find any SE involved with three cancer-related genes (17.6%, 3/17): *NFATC2* and *SALL4* in MCF7, *TERT* in HCC1954. In addition, deepC-SV could not find some of gene-SE pairs: *NT5C2* in T47D, and *GNAS* and *BCL6* in SKBR3. Rearrangements around *TERT* were previously reported to be involved with Hi-C contact map changes in HCC1954 (Akdemir et al. 2020). *SALL4* (T. Chen et al. 2020) and *GNAS* (X. Jin et al. 2019) were known to

promote breast cancer progression, and previously reported to be involved with SE hijacking events (Xu et al. 2022; Wang et al. 2021). To further verify these gene-SE pair findings predicted by InfoHiC, we investigated the Hi-C intensities of each gene-SE pair in the cancer Hi-C and noncancerous Hi-C experiments (HMEC and MCF10A) (Appendix Table S5). The cancer Hi-C experiments showed higher Hi-C intensities of the gene-SE pairs, compared to noncancerous cell lines, up to 74X fold change (*GNAS* in SKBR3). For several genes (*TERT* in HCC1954, *SALL4* in MCF7, and *NT5C2* in T47D), the noncancerous cell lines showed the zero Hi-C intensity. The results demonstrated that InfoHiC could predict SE hijacking events not found by deepC-SV, which could be verified in Hi-C experiments.

3. In Fig. 2&3, the authors clearly show that the prediction accuracy drops quickly when the distance increases. It should be at least discussed the relationship of prediction accuracy of neo-TAD and neo-loops against the size of TAD distance.

Answer) Thank you for the comment. The prediction accuracy of the long-range interactions may be important to predict neo-TADs over the 1-Mb range. We added discussion on our results with respect to the genomic distance in the main manuscript as follows.

Page 13) We showed that InfoHiC could predict cancer Hi-C matrices with a DSC > 0.6 in the 1-Mb range. However, predicting long-range interactions is still challenging. When we tested using the breast cancer cell lines, the DSCs in the 1-Mb to 2-Mb range decreased considerably (Figs. 2 and 3). The previously reported TAD sizes were about 560 kb, 880 kb, and 1,480 kb for the 25th, median, and 75th quantile, respectively (Dixon et al. 2012). In the breast cancer cell lines that we analyzed, the quantile sizes of neo-TADs were similar with the previous report; 600 kb, 880 kb, and 1,280 kb for the 25th, median, and 75th quantile, respectively. As more than 25% of neo-TADs were over the 1-Mb range, the current model might have to be scaled up to the mega-base range. Recently, Orca (Zhou 2022) presented a hierarchical CNN model for the mega-base scale Hi-C prediction in noncancerous cell lines. The hierarchical model may be helpful to increase cancer Hi-C prediction performance in the 2-Mb range.

Minor points:

1. Supplementary Fig. 1 is not cited in the manuscript.

Answer) Thank you for the comment. Supplementary Fig. 1 was changed as Appendix Fig. 3 in the revised manuscript. We added the citation for Appendix Fig. 3 in the main manuscript as follows.

Page 6) we tested the models on the test set of four breast cancer cell lines (Appendix Fig. S3).

2. The title of axis in Fig. 2 are not clearly labeled.

Answer) Thank you for the comment. We added the title of axis in Fig. 2.

3. The p-values or statistical significance should be presented in Figure 4b and Figure 5b.

Answer) Thank you for the comment. We presented the statistical significance using the notations (*, **, ***, or not significant) in Figure 4b and Figure 5b. In addition, we added Appendix Tables 4 and 6 to show the p-values that we obtained. We added the description at the figure legends in the main manuscript as follows.

Fig. 4 legend at page 22-23) The p-values from the one-sided Student's t-test are shown in Appendix Table S4. *, $p < 0.05$, **, $p < 0.01$, and ***, $p < 0.001$ by the one-sided Student's t-test versus the reference group, ns, not significant.

Fig. 5 legend at page 23) The p-values from the one-sided Student's t-test are shown in Appendix Table S6. *, $p < 0.05$, **, $p < 0.01$, and ***, $p < 0.001$ by the one-sided Student's t-test versus the reference group, ns, not significant.

Answers to Reviewer #2:

In this manuscript, Lee et al. introduce InfoHiC, a novel method for predicting Hi-C contacts using whole genome sequencing data in cancer. The approach comprises two main steps: 1) Identifying and assembling cancer-specific genomic contigs using their previously published method InfoGenomeR. 2) Employing the well-established CNN-based framework DeepC (Schwessinger et al. Nature Methods. 2020), which was initially designed for predicting Hi-C contacts from DNA sequences in normal cells, to predict Hi-C contacts specific to each contig identified in the first step. The authors demonstrate the utility of InfoHiC by applying it to datasets from patients with breast cancer and medulloblastoma. The results underscore its potential in uncovering the impact of non-coding SVs on the regulation of oncogene transcription. While the core concept is intriguing, the unique contributions of this work require further clarification. Specific concerns are detailed below.

Major concerns:

1) The authors need to clarify the unique contributions of this work. At present, it seems InfoHiC is just a pipeline that combines InfoGenomeR and DeepC, with minor alterations in

encoding. Additionally, a pertinent concern arises from the observation that there is no significant difference between deepC-SV and InfoHiC in supplementary figure 1.

Answer) Thank you for the comment. InfoHiC enables Hi-C prediction from WGS of cancer patients. The previous methods (deepC, Akita, and Orca) does not start from WGS, and they are not readily applicable to cancer patients, where numerous genomic variants exist and can change Hi-C contacts in a genome-wide scale. Our work requires considerable efforts including integration with WGS variant calling, SV contig generation, and re-defining the model for cancer Hi-C prediction. The re-defined model includes the original concepts such as contig-to-total Hi-C conversion from multiple contigs, and the total Hi-C loss function that is required for cancer Hi-C training. We showed that our implementation can predict Hi-C data from WGS and is applicable to cancer patients as we shown in the main manuscript. We believe that our effort can contribute to cancer research fields significantly. To clarify our contribution and the difference between InfoHiC and deepC, we added a comparison section in the manuscript along with Appendix Figs. S1 and S2 as follows.

Page 4)

Comparison with the reference-based CNN model

We compared InfoHiC with the reference-based CNN model, deepC (Schwessinger et al. 2020). The main feature of InfoHiC against deepC is that genomic variants are included in the prediction model by utilizing WGS. In the training phase, we used the same transfer learning approach with deepC, where the first five CNN layers were pre-trained to predict chromatin features (Zhou and Troyanskaya 2015). The layers were initialized with the same weights between InfoHiC and DeepC. Then, we used cancer Hi-C data to train the entire CNN models of deepC and InfoHiC with dilated CNN layers added to the first five layers. DeepC used the reference sequence for training, and it applied several normalization steps to raw Hi-C data, which includes 1) the implicit Hi-C normalization (Imakaev et al. 2012), 2) percentile normalization (Schwessinger et al. 2020), and 3) mean filter smoothing (Schwessinger et al. 2020). Because the normalization steps assumed that Hi-C contacts were from the reference sequence where no genomic variation exists, it is not readily applicable to cancer Hi-C data where genomic variations change Hi-C contact maps. Specifically, percentile normalization assigned fixed values (one to ten integers) depending on the percentile of the Hi-C intensity for each genomic distance. However, cancer Hi-C intensities depend on large genomic variants (CNAs and SVs); Hi-C contacts have different intensities even in the same genomic distance. Therefore, we removed the normalization steps, and trained deepC and InfoHiC using raw Hi-C data. InfoHiC assumed that the Hi-C contact could be derived from the haplotypes containing different SNPs and CNAs, and used haplotype sequences that were inferred from WGS for training.

In the testing phase, deepC was not available for SV-derived Hi-C prediction, specifically for large SVs or translocations commonly existing in cancer genomes (Lee and Lee 2021). Thus, to compare

with InfoHiC, we assumed a common scenario of the deepC usage for SV-derived Hi-C prediction, named deepC-SV, where users would manipulate the reference sequence using variant calls from an SV caller, and then perform Hi-C prediction on the SV-derived sequence. Because the SV caller detects an SV in a local window, the sequence used for deepC prediction contained one SV. In contrast, InfoHiC used SV contigs reconstructed from InfoGenomeR, which could contain multiple SVs, and then performed Hi-C prediction on SV contigs. SV contigs could contain different SNPs and CNAs, which were encoded in the matrices of SV contig sequences. In addition, InfoHiC assumed that the Hi-C contacts can be derived from multiple SV contigs, and summarized prediction results from the multiple SV contigs in the output with the total Hi-C matrix. An illustration of the differences among deepC, deepC-SV, and InfoHiC for cancer Hi-C prediction is shown in Appendix Figs. S1 and S2).

Appendix Fig. 1: Comparison between InfoHiC and deepC on reference windows InfoHiC and deepC use different strategies for Hi-C prediction on reference windows. **a** The Hi-C experiment truth is shown on the reference window of the test set 2. **b** InfoHiC utilizes WGS and reconstruct SV contigs, resulting in three SV contigs (Contig 1 to 3). Contig 1 is the same with the reference (red), Contig 2 has the fg deletion (blue), and Contig 3 has the fg and cd deletion together (green). **c** In contrast, deepC uses the reference sequence for the reference window prediction. No genomic variation is encoded in deepC. InfoHiC validates Hi-C prediction results on the total Hi-C coordinate, where Hi-C intensities from SV contigs are summed together. Note that even in the reference window, CNAs and SVs can change Hi-C intensities.

Appendix Fig. 2: Comparison between InfoHiC and deepC-SV on SV windows InfoHiC and deepC-SV use different strategies for Hi-C prediction on SV windows. **a** The Hi-C experiment truth is shown on the SV window of the test set 2. **b** InfoHiC utilizes WGS and reconstruct SV contigs, resulting in three SV contigs (Contig 1 to 3). Contig 1 is the same with the reference (red), Contig 2 has the fg deletion (blue), and Contig 3 has the fg and cd deletion together (green). **c** In contrast, deepC-SV uses SV calls from an SV caller, and performs prediction on each SV call (the fg deletion and cd deletion, respectively). InfoHiC validates Hi-C prediction results on the total Hi-C coordinate, where Hi-C intensities from SV contigs are summed together. Three SV windows can be validated (ab-ef, de-hi, and ab-hi windows). Note that Hi-C contacts (the e-h contact and e-i contact) from Contig 2 (the de-hi TAD) and Contig 3 (the ab-e-hi TAD) are overlapped in the de-hi SV window. The ab-hi window is not verifiable in the deepC-SV.

2) In the original DeepC paper, a particular procedure is employed to normalize the Hi-C data by genomic distance before model training. However, it seems that the authors of this paper trained both the InfoHiC and DeepC models using raw Hi-C signals. Could the authors clarify if any adaptations or modifications have been made in this aspect?

Answer) Thank you for the comment. We added the description of the normalization process we modified in the manuscript as follows.

Page 4) DeepC used the reference sequence for training, and it applied several normalization steps to raw Hi-C data, which includes 1) the implicit Hi-C normalization (Imakaev et al. 2012), 2) percentile normalization (Schwessinger et al. 2020), and 3) mean filter smoothing (Schwessinger et al. 2020). Because the normalization steps assumed that Hi-C contacts were from the reference sequence where no genomic variation exists, it is not readily applicable to cancer Hi-C data where genomic variations change Hi-C contact maps. Specifically, percentile normalization assigned fixed values (one to ten integers) depending on the percentile of the Hi-C intensity for each genomic distance. However, cancer Hi-C intensities depend on large genomic variants (CNAs and SVs); Hi-C contacts have different intensities even in the same genomic distance. Therefore, we removed the normalization steps, and trained deepC and InfoHiC using raw Hi-C data. InfoHiC assumed that the Hi-C contact could be derived from the haplotypes containing different SNPs and CNAs, and used haplotype sequences that were inferred from WGS for training.

3) Regarding to the HSCN encoding, there is a discrepancy between the scheme depicted in figure 1a and the descriptions provided in the Method section. According to the Method section, each value in the contig matrix represents the copy number of the corresponding DNA base (A/T/C/G). However, in figure 1a, contig 2 is elongated due to a duplication event, resulting in its extension beyond the lengths of contig 1 and contig 3. If such an elongation (or shrinkage) occurs for every contig experiencing copy number variations, it follows that each column in the encoding should ideally reflect a copy number of 1, rather than the original copy number value estimated from whole-genome sequencing.

Answer) Thank you for the comment. We changed the haplotype-specific copy number (HSCN) term into contig-specific copy number (CSCN). If the elongated contig had multiple copies more than 1, each element in the encoded contig-specific matrix had the corresponding copy number value. For training, because we performed training on the reference regions without SVs, no elongation exists on training windows, thus contig-specific copy numbers are the same with the haplotype-specific copy numbers. We added more description for the contig-specific copy number encoding in the manuscript as follows.

Page 16) In addition, for the reference regions, contig-specific copy numbers are the same with

haplotype-specific copy numbers as they have neither CNA nor SV breaks. For example, if a haplotype has three copies in the training window of the reference region, there is one genomic contig encoded by the haplotype copy number ($n = 3$).

Page 22) Note that the contigs were from two copies of $\text{der}(8)\text{t}(8;14)$, and the matrices were encoded by two copies, instead of the one-hot encoding. Prediction was performed on the matrices where each base element has a contig-specific copy number value.

4) The authors benchmarked multiple variants of InfoHiC models in figures 2-3 and supplementary figure 1, and showed that the best performance is achieved by the 2-Mb model with the 1-Mb model transferred. However, there remains ambiguity regarding the specific model employed for the subsequent analyses. Specifically, for patients with breast cancer, it is not clear whether the model employed was trained on T47D Hi-C data or noncancerous cell lines. Similarly, in the case of patients with medulloblastoma, details regarding the training data for the applied model are needed.

Answer) Thank you for the comment. We specified the InfoHiC model used for the subsequent analyses, and described the details in the method section in the manuscript as follows.

Page 7) We used the 2-Mb CSCN encoding model transferred from the 1-Mb model and trained by T47D that showed the best performance on breast cancer cell line tests.

Page 8) Subsequently, using the 2-Mb CSCN encoding model trained by T47D, we applied InfoHiC to patients with BRCA ($n = 90$) in TCGA to investigate the non-coding effects of SVs.

Page 10) We used the 1-Mb CSCN decoding model trained by the neuro-progenitor cell line (NPC).

Page 19)

InfoHiC models used for Hi-C prediction

We used different InfoHiC models depending on each cancer type. For breast cancer cell lines (T47D, BT474, MCF7, HCC1954, and SKBR3), we used the 2-Mb CSCN encoding model transferred from the 1-Mb CSCN encoding model, which was trained by T47D Hi-C data. For BRCA datasets, we used the 2-Mb CSCN encoding model as well, which was used for breast cancer cell lines. For patients with medulloblastoma, we trained the 1-Mb CSCN decoding model using Hi-C data of the NPC cell line, and then we fine-tuned the model for each medulloblastoma driver gene (*GFI1*, *GFI1B*, and *PRDM6*); the 1-Mb CSCN decoding model was trained with an additional round of ten iterations with the 4-Mb window centered on each gene. The fine-tuned models showed better neo-TAD annotation

results on each gene than the NPC cell line model without fine-tuning (Appendix Fig. S14), and they were used for medulloblastoma sample analysis: *PDRM6* for PD2107, *GFI1B* for PD2105 and PD2110, and *GFI1B* for PD2109, respectively.

5) All the training and predictions performed in this work were limited to a relatively low resolution of 40kb. While this resolution is appropriate for predicting TADs/neo-TADs, it may not be optimal for accurately predicting neo-loops or enhancer hijacking. How about the performance of InfoHiC at higher resolution such as 10kb or 5kb?

Answer) Thank you for the comment. For training at higher resolution such as 10kb, higher coverage Hi-C data may be required. We obtained high-coverage K562 Hi-C data, and added the results in the manuscript as follows.

Page 6-7)

Training on cancer Hi-C data at 10-kb resolution

For evaluating InfoHiC in a higher resolution, we trained InfoHiC at a 10-kb resolution. Based on five random splits that we used for a 40-kb resolution, we trained the models using T47D Hi-C data at a 10-kb resolution, and tested on the test set 1. InfoHiC and DeepC showed lower average DSCs in the 10-kb models (InfoHiC, 0.456; DeepC, 0.425) than in the 40-kb models (InfoHiC, 0.668; DeepC, 0.607). Hi-C read coverage might be important for model performance, and higher coverage Hi-C data might be required for a 10-kb resolution. Therefore, we tested Hi-C data of a leukemia cell line (K562) that had higher coverage than T47D for the 10-kb training. The 3DIV (Kim et al. 2021) Hi-C read mapping reported 0.76M and 0.42M properly paired reads in K562 and T47D, respectively. We trained the models using K562 Hi-C data and tested on a random split (80% training, 10% validation, and 10% test set 1). The K562 10-kb model showed better performance than the T47D 10-kb model (InfoHiC, 0.507; DeepC, 0.442 average DSC on test set 1), showing that higher Hi-C coverage might be effective for 10-kb training. In addition, we tested the K562 10-kb model on test set 2 with SVs, and it showed 0.730 Pearson's R on SV windows. Moreover, neo-loop annotation might benefit from the 10-kb model, and we tested if the K562 10-kb model could predict the neo-loop formation around the *RAB36* gene that was reported in K562 (X. Wang et al. 2021). We could assemble the inter-chromosomal (chromosomes 9, 13, and 22) SV contig from WGS, which was the same SV contig previously found in K562 (X. Wang et al. 2021). When we performed Hi-C prediction on the SV contig using the K562 40-kb model, we could find a neo-TAD event of the *RAB36* gene, while neo-loops were not detected around the *RAB36* gene. Using the K562 10-kb model, we could successfully predict neo-loops around the *RAB36* gene (Appendix Fig. S4). We also provide the K562-trained InfoHiC model at <https://github.com/dmcb-gist/InfoHiC>, which can help neo-loop prediction at a 10-kb resolution.

6) The method by which super enhancers and typical enhancers were defined is not mentioned.

Answer) Thank you for the comment. We described the details in the manuscript as follows.

Page 20) TEs and SEs were downloaded from SEdb (Jiang et al. 2019). We used tissue-specific annotations depending on cancer types, which are summarized in Appendix Table S10.

7) In the introduction, the authors state that "Third, the existing methods cannot utilize cancer Hi-C data; 1) even though Hi-C data of numerous cancer types have been accumulated, they cannot be used for training and 2) Hi-C prediction from SVs cannot be validated using cancer Hi-C data, preventing researchers from verifying neo-TAD predictions". This appears to be inaccurate. If this is the case, it raises questions about how the authors conducted their own comparison of methods using cancer Hi-C data. The second point is even more confusing, as it is feasible to compare the predictions against the observed Hi-C contacts by visually juxtaposing contact maps around SVs. This practice is exemplified in Figure 4 from Tan J et al. (Nat Biotechnol 2023, <https://doi.org/10.1038/s41587-022-01612-8>).

Answer) Thank you for the comment. The existing methods ignores CNA- and SV-derived Hi-C contacts, thus may not be optimal for training on cancer Hi-C data. In addition, cancer Hi-C validation methods were not defined in the existing methods. The juxtaposing method can be used for validation possibly, but it is outside of their prediction workflows, and is restricted to a specific SV contig not on a genome-wide scale. On the other hand, InfoHiC performs validation on the predicted total Hi-C matrix in the workflow, and it is not restricted to a specific SV contig. To make it clear, we revised the paragraph in the manuscript as follows.

Page 2) Third, the existing methods used the reference sequence for training, which ignores CNA- and SV-derived Hi-C contacts existing in cancer Hi-C data. Cancer genomes have genomic variations that can change Hi-C contact maps on a genome-wide scale (Akdemir et al. 2020). To leverage accumulated cancer Hi-C data (Kim et al. 2021) for variant-derived 3D genome prediction, genomic variations existing in cancer genomes should be included in the prediction model. In addition, because there is no validation method for cancer Hi-C prediction, users should manipulate the observed Hi-C contact map (Tan et al. 2023; X. Wang et al. 2021) to validate prediction results outside of their prediction workflows. Because of such limitations, the 3D cancer genome has been poorly investigated by the existing prediction methods.

8) As a method paper, it is essential for the authors to provide detailed guidelines that ensure successful installation and execution of the software. The current documentation solely offers a list of required packages for the installation, but my attempt to install those packages failed on my machine. This issue arises from a possible conflict where tensorflow=1.14 necessitates Python 3.7, while the neoloop package mandates Python version 3.8 or higher.

Answer) Thank you for the comment. InfoHiC is also available for tensorflow 2.5 and python 3.8. We updated requirements in the GitHub repository to resolve the conflict with the neoloop package.

Minor concerns:

1) There is a discrepancy between figure 1a and its corresponding figure caption. The caption specifies "haplotype 1 in blue and haplotype 2 in green", however, the haplotype 1 is actually drawn in green, and the haplotype 2 is in blue, based on the figure.

Answer) Thank you for the comment. We edited Figure 1, and updated the legend as follows.

Page 22) Copy numbers of each haplotype (haplotype 1 in green and haplotype 2 in blue) and the number of Hi-C reads (red) are represented by color saturation.

2) In Figure 2, the x-axis labels are missing.

Answer) Thank you for the comment. We added labels of the x-axis in Figure 2.

3) Based on my comprehension, the distance-stratified correlation (DSC) consists of a series of correlation values, each corresponding to a specific genomic distance. I'm interested in understanding how the DSC values in the second paragraph of the section "InfoHiC outperforms the reference-based CNN model" were computed. Are these values simply the averages taken across all genomic distances?

Answer) Thank you for the comment. We added the description of the DSC computed as follows.

Page 5) The DSC was computed as an average of correlations across all genomic distances, where each correlation value was measured in a specific genomic distance.

4) On page 6, I was confused about the 1-Mb and 2-Mb models mentioned in the final paragraph. This puzzlement arises from the fact that both training and validation were conducted using 4-Mb reference genomic windows, as indicated in the preceding paragraph on the same page.

Answer) Thank you for the comment. We clarified naming in the manuscript as follows.

Page 5) The 4-Mb window contained 100 (4 Mb / 40 kb) output vectors at a 40-kb resolution. For each output vector, the 1-Mb or 2-Mb genomic sequence centered on the output vector was used for prediction. We named the 1-Mb or 2-Mb model to indicate the 1-Mb or 2-Mb genomic sequence was used for prediction of each output vector, respectively.

5) It would be beneficial and easier for readers to understand by adding panel labels to supplementary figures, just as the authors did for the main figures.

Answer) Thank you for the comment. We added panel labels to supplementary figures. Note that supplementary figures are renamed as Appendix figures.

6) I was confused by one of the criteria used for TE/SE hijacking annotation within a neo-TAD "an enhancer did not exist in the reference TAD prediction" (on page 25). Did the authors mean that the enhancer and the gene should not co-exist in the reference TAD prediction?

Answer) Thank you for the comment. We clarified the sentence.

Page 20) the enhancer and the gene did not co-exist in the reference TAD prediction

Answers to Reviewer #3:

The authors have proposed a computational framework called InfoHiC, which starts from whole-genome sequencing (WGS) data and can predict the chromatin contact maps of cancer samples using convolutional neural networks. They demonstrated that InfoHiC can produce chromatin contact maps similar to real Hi-C experiments and can correct aberrant chromatin conformation caused by structural variations. As a result, it can identify newly formed topologically associated domains and chromatin loops in cancer samples. They provided some examples of enhancer-hijacking events to illustrate the correlation between the formation of new chromatin structures and the abnormal expression of key genes in cancer. We believe that this work has the potential to advance our understanding of the three-dimensional genome in cancer research.

Major issues:

1. The InfoHiC model incorporates a pre-trained model from deepC and further extends it by adding some dilated convolutional layers. The authors also employ various strategies to retrain the entire model with both cancer Hi-C data and WGS data. In their method comparisons, they mainly compare InfoHiC with deepC. However, the authors did not provide a detailed description of how they utilized deepC in their method comparisons. It remains

unclear whether they directly employed the pre-trained deepC model or retrained deepC with the cancer data. Besides, the main models of InfoHiC and deepC are very similar. There are also other methods that can predict 3D genome folding from DNA sequence, such as Akita as mentioned. The authors should compare InfoHiC with these methods.

Answer 1) Thank you for the comment. We described the method detail of InfoHiC at the method section in the original manuscript. However, we agreed that there should be the detailed description before showing the comparison results in the manuscript. To address the first comment, we added the detailed description of InfoHiC when starting the result section, and attached the illustrations (Appendix Figs. S1 and S2) to show differences between deepC and InfoHiC in the revised manuscript as follows.

Page 4)

Comparison with the reference-based CNN model

We compared InfoHiC with the reference-based CNN model, deepC (Schwessinger et al. 2020). The main feature of InfoHiC against deepC is that genomic variants are included in the prediction model by utilizing WGS. In the training phase, we used the same transfer learning approach with deepC, where the first five CNN layers were pre-trained to predict chromatin features (Zhou and Troyanskaya 2015). The layers were initialized with the same weights between InfoHiC and DeepC. Then, we used cancer Hi-C data to train the entire CNN models of deepC and InfoHiC with dilated CNN layers added to the first five layers. DeepC used the reference sequence for training, and it applied several normalization steps to raw Hi-C data, which includes 1) the implicit Hi-C normalization (Imakaev et al. 2012), 2) percentile normalization (Schwessinger et al. 2020), and 3) mean filter smoothing (Schwessinger et al. 2020). Because the normalization steps assumed that Hi-C contacts were from the reference sequence where no genomic variation exists, it is not readily applicable to cancer Hi-C data where genomic variations change Hi-C contact maps. Specifically, percentile normalization assigned fixed values (one to ten integers) depending on the percentile of the Hi-C intensity for each genomic distance. However, cancer Hi-C intensities depend on large genomic variants (CNAs and SVs); Hi-C contacts have different intensities even in the same genomic distance. Therefore, we removed the normalization steps, and trained deepC and InfoHiC using raw Hi-C data. InfoHiC assumed that the Hi-C contact could be derived from the haplotypes containing different SNPs and CNAs, and used haplotype sequences that were inferred from WGS for training.

In the testing phase, deepC was not available for SV-derived Hi-C prediction, specifically for large SVs or translocations commonly existing in cancer genomes (Lee and Lee 2021). Thus, to compare with InfoHiC, we assumed a common scenario of the deepC usage for SV-derived Hi-C prediction, named deepC-SV, where users would manipulate the reference sequence using variant calls from an SV caller, and then perform Hi-C prediction on the SV-derived sequence. Because the SV caller

detects an SV in a local window, the sequence used for deepC prediction contained one SV. In contrast, InfoHiC used SV contigs reconstructed from InfoGenomeR, which could contain multiple SVs, and then performed Hi-C prediction on SV contigs. SV contigs could contain different SNPs and CNAs, which were encoded in the matrices of SV contig sequences. In addition, InfoHiC assumed that the Hi-C contacts can be derived from multiple SV contigs, and summarized prediction results from the multiple SV contigs in the output with the total Hi-C matrix. An illustration of the differences among deepC, deepC-SV, and InfoHiC for cancer Hi-C prediction is shown in Appendix Fig. S1 and S2).

Appendix Fig. 1: Comparison between InfoHiC and deepC on reference windows InfoHiC and deepC use different strategies for Hi-C prediction on reference windows. **a** The Hi-C experiment truth is shown on the reference window of the test set 2. **b** InfoHiC utilizes WGS and reconstruct SV contigs, resulting in three SV contigs (Contig 1 to 3). Contig 1 is the same with the reference (red), Contig 2 has the fg deletion (blue), and Contig 3 has the fg and cd deletion together (green). **c** In contrast, deepC uses the reference sequence for the reference window prediction. No genomic variation is encoded in deepC. InfoHiC validates Hi-C prediction results on the total Hi-C coordinate, where Hi-C intensities from SV contigs are summed together. Note that even in the reference window, CNAs and SVs can change Hi-C intensities.

Appendix Fig. 2: Comparison between InfoHiC and deepC-SV on SV windows InfoHiC and deepC-SV use different strategies for Hi-C prediction on SV windows. **a** The Hi-C experiment truth is shown on the SV window of the test set 2. **b** InfoHiC utilizes WGS and reconstruct SV contigs, resulting in three SV contigs (Contig 1 to 3). Contig 1 is the same with the reference (red), Contig 2 has the fg deletion (blue), and Contig 3 has the fg and cd deletion together (green). **c** In contrast, deepC-SV uses SV calls from an SV caller, and performs prediction on each SV call (the fg deletion and cd deletion, respectively). InfoHiC validates Hi-C prediction results on the total Hi-C coordinate, where Hi-C intensities from SV contigs are summed together. Three SV windows can be validated (ab-ef, de-hi, and ab-hi windows). Note that Hi-C contacts (the e-h contact and e-i contact) from Contig 2 (the de-hi TAD) and Contig 3 (the ab-e-hi TAD) are overlapped in the de-hi SV window. The ab-hi window is not verifiable in the deepC-SV.

Answer 2) When we started the method development, we also benchmarked Akita. However, Akita showed a poor performance on cancer Hi-C prediction, thus we selected deepC as a baseline model instead of Akita. We added discussion on Akita in the manuscript as follows.

Page 13) In the article, we showed comparison results between InfoHiC and deepC. When we adapted available reference-based CNN models for 3D cancer genome prediction, we also benchmarked Akita (Fudenberg, Kelley, and Pollard 2020). When we trained Akita using T47D Hi-C data, and tested on test set 1 of T47D, it showed a poor Pearson correlation of less than 0.3. The main difference between Akita and deepC is that deepC uses 1D CNN layers followed by fully connected layers with different weights per each genomic distance (Methods). In contrast, Akita uses 2D CNN layers, and there is no fully connected layer like in deepC, which can model Hi-C intensities depending on the genomic distance. Instead, in Akita, genomic distances are encoded in the 2D map before applying the 2D CNN layers (Fudenberg, Kelley, and Pollard 2020). Our initial benchmark showed that deepC could be successfully trained on cancer Hi-C data, and therefore, we adapted the deepC architecture for InfoHiC instead of Akita.

2. The authors demonstrated that InfoHiC can generate chromatin contact maps similar to those from cancer Hi-C experiments using whole-genome sequencing (WGS) data and can also correct the maps in SV regions. They also mentioned another method, NeoLoopFinder, which can identify chromatin interactions induced by SVs from Hi-C contact maps and reconstruct local Hi-C maps around breakpoints. NeoLoopFinder can also be used to discover new loops and TADs and identify enhancer-hijacking events. After all, correcting Hi-C maps affected by SVs to their correct versions seems to be a very crucial aspect of cancer three-dimensional genomics research. InfoHiC and NeoLoopFinder approach this issue from different perspectives. The authors may further discuss the similarities and differences between these two methods, as well as their respective advantages. The authors should compare the performance of these two methods in reconstructing accurate Hi-C maps, discovering new loops, and identifying enhancer hijacking events.

Answer) Thank you for the comment. NeoLoopFinder is a Hi-C analytic method to discover neo-loops when Hi-C data exist. InfoHiC is a Hi-C prediction method and predicts Hi-C data for samples that have no Hi-C data. To clarify the difference, we added the description in the manuscript. In addition, even though their purposes are different, it may be worth to compare neo-loops found by NeoLoopFinder with the prediction results by InfoHiC. We added the comparison result in the manuscript as follows.

Page 12) InfoHiC is a Hi-C prediction method, which is different from other neo-loop or neo-TAD

finding methods that require existing Hi-C sequencing data (Wang et al. 2021). InfoHiC predicts Hi-C data for samples that have no Hi-C data. Nonetheless, to show the strength of InfoHiC for discovering neo-TADs and neo-loops, we compared our neo-TAD and neo-loop prediction results with previous NeoLoopFinder (a neo-loop finding tool) results on five breast cancer cell lines (Wang et al. 2021). First, we counted the total number of SVs used for neo-loop or neo-TAD findings. NeoLoopFinder only considered large SVs or translocations, and found a total of 479 SVs from the five cell lines. In contrast, InfoHiC found a total of 1,944 SVs. NeoLoopFinder reported a total of 524 neo-loop-involved genes and InfoHiC found 2,325 neo-TAD-involved genes and 3,738 neo-loop-involved genes. InfoHiC could find 73.3% of neo-loop-involved genes found by NeoLoopFinder in neo-TADs or neo-loops. In contrast, NeoLoopFinder could only find 7% of neo-loop-involved genes found by InfoHiC. For example, we found the SE hijacking event of a cancer-related gene, *GNA13* in BT474 (Appendix Fig. S5), but NeoLoopFinder could not detect it. InfoHiC and NeoLoopFinder used a different Hi-C resolution (40 kb in InfoHiC and 10 kb in NeoLoopFinder), and the 40-kb resolution may call more genes in a broader range than the 10-kb resolution. However, we observed that 55.3% of neo-loop-involved genes not found by NeoLoopFinder were associated with SVs < 1 Mb, suggesting that small SVs were important for neo-loop findings and InfoHiC enabled the discovery of neo-loops from them.

Minor issues:

1. I am confused about the statement on page 9, "In those cases, a Hi-C contact in the cancer Hi-C matrix was not unique to a single TAD, and neo-TADs were overlapped with Hi-C contacts from other TADs". We think a Hi-C contact, which is a pixel in the Hi-C map, either falls within one TAD or doesn't. How could it possibly overlap with multiple TADs? Thus, I can't understand the meaning of Fig 4.a. The authors may give more explanations.

Answer) Thank you for the comment. To help readers understand the Hi-C contact overlap, we added an example of overlapping Hi-C contacts with an illustration (Appendix Figure 2) in the manuscript as follows.

Page 7) An example of overlapping Hi-C contacts is shown in Appendix Fig. S2, where Hi-C contacts from Contig 2 and Contig 3 are overlapped in the SV window.

Page 3 in the Appendix Figures) Note that Hi-C contacts (the e-h contact and e-i contact) from Contig 2 (the de-hi TAD) and Contig 3 (the ab-e-hi TAD) are overlapped in the de-hi SV window.

2. The authors compare the expression of genes in neo-TADs or neo-loops with genes in reference TADs and loops in Fig4 and Fig5. However, they didn't provide a detailed description of how they defined the reference TADs or loops.

Answer) Thank you for the comment. We added more description on reference TADs or loops.

Page 7) Genes in neo-TADs or neo-loops were compared with all the other genes which did not exist in neo-TADs or neo-loops (considered them to be in reference TADs or loops).

3. The authors seem to produce the WGS data and RNA-seq data of brain tumors on page 25 "Sample preparation and data preprocessing". However, they didn't provide the accession number of these data in "Availability of data and materials" section.

Answer) Thank you for the comment. We uploaded variant call files from WGS and gene expression profiles from RNA-seq. Raw WGS and RNA-seq fastqs are available upon request.

Page 21) Variant calls (SNVs, indels, CNAs, and SVs) from WGS and gene expression profiles from RNA-seq of pediatric patients with medulloblastoma are available from the European Genome-phenome Archive (accession code EGAD00001009507) and Zenodo (<https://doi.org/10.5281/zenodo.10544039>), respectively. Raw WGS and RNA-seq fastq files are available upon request.

4. In the legend of Supplementary Fig. 1. The statement "according to the mega-base scale distance range (0-2 Mb)", We think it should be 0-1Mb.

Answer) Thank you for the comment. It should be 0-1Mb. Note that in the revision, Supplementary Fig. 1 was changed as Appendix Fig. 3. We edited as follows.

Page 4 in Appendix Figures) Distance-stratified correlations (line) for reference windows are shown (top) according to the mega-base scale distance range (0-1 Mb)

5. The authors should polish the writing further.

Answer) Thank you for the comment. We polished the manuscript again with an English editor.

11th Mar 2024

Manuscript Number: MSB-2023-11897R

Title: Prediction of the 3D cancer genome from whole genome sequencing using InfoHiC

Dear Dr. Lee,

Thank you again for submitting your work to Molecular Systems Biology. We have now heard back from two of the original reviewers who had evaluated your study. As you will see below, Reviewer 2 continues to have concerns on issues that were already brought up in the previous round of review, particularly on the documentation and usability of InfoHiC, which is important to make InfoHiC easily accessible for future users.

In addition, in order to hopefully speed up the review process, I had asked Reviewer 2 in a cross-commenting session to check whether the concerns of Reviewers 1 and 3 had been fully addressed, as neither reviewer had submitted a report. Reviewer 2 offered their additional concerns on the benchmarking, which I have included below. While Reviewer 3 has subsequently submitted a review that is supportive of your manuscript, given the importance of benchmarking in a manuscript with such a strong methodological focus, we would highly encourage you to perform additional benchmarking analyses to highlight the utility and advantage of InfoHiC as suggested by Reviewer 2. Please ensure that you include responses to concerns in your point-by-point response. In case you would like to discuss in further detail any of the comments, I would be happy to schedule a call.

Once again, please ensure that the revised manuscript follows the formatting requirements as mentioned in the previous decision letter and reiterated here that we require:

1) A .docx formatted version of the manuscript text (including legends for main figures, EV figures and tables). Please make sure that the changes are highlighted to be clearly visible. Alternatively you may choose to submit your manuscript as a LaTeX file.

4) A .docx formatted letter INCLUDING the reviewers' reports and your detailed point-by-point responses to their comments. As part of the EMBO Press transparent editorial process, the point-by-point response is part of the Peer Review File (PRF), which will be published alongside your paper.

5) A complete author checklist, which you can download from our author guidelines (<https://www.embopress.org/page/journal/17574684/authorguide#submissionofrevisions>). Please insert information in the checklist that is also reflected in the manuscript. The completed author checklist will also be part of the PRF.

6) Please note that all corresponding authors are required to supply an ORCID ID for their name upon submission of a revised manuscript.

7) It is mandatory to include a 'Data Availability' section after the Materials and Methods. Before submitting your revision, primary datasets produced in this study need to be deposited in an appropriate public database, and the accession numbers and database listed under 'Data Availability'. Please remember to provide a reviewer password if the datasets are not yet public (see <https://www.embopress.org/page/journal/17574684/authorguide#dataavailability>).

In case you have no data that requires deposition in a public database, please state so in this section. Note that the Data Availability Section is restricted to new primary data that are part of this study. This study includes no data deposited in external repositories.

8) For data quantification: please specify the name of the statistical test used to generate error bars and P values, the number (n) of independent experiments (specify technical or biological replicates) underlying each data point and the test used to calculate p-values in each figure legend. The figure legends should contain a basic description of n, P and the test applied. Graphs must include a description of the bars and the error bars (s.d., s.e.m.). Please provide exact p values.

9) Our journal encourages inclusion of *data citations in the reference list* to directly cite datasets that were re-used and obtained from public databases. Data citations in the article text are distinct from normal bibliographical citations and should directly link to the database records from which the data can be accessed. In the main text, data citations are formatted as follows: "Data ref: Smith et al, 2001" or "Data ref: NCBI Sequence Read Archive PRJNA342805, 2017". In the Reference list,

data citations must be labeled with "[DATASET]". A data reference must provide the database name, accession number/identifiers and a resolvable link to the landing page from which the data can be accessed at the end of the reference. Further instructions are available at .

<https://www.embopress.org/page/journal/17574684/authorguide#expandedview>

11) For more information: There is space at the end of each article to list relevant web links for further consultation by our readers. Could you identify some relevant ones and provide such information as well? Some examples are patient associations, relevant databases, OMIM/proteins/genes links, author's websites, etc...

12) Author contributions: CRediT has replaced the traditional author contributions section because it offers a systematic machine readable author contributions format that allows for more effective research assessment. Please remove the Authors Contributions from the manuscript and use the free text boxes beneath each contributing author's name in our system to add specific details on the author's contribution. More information is available in our guide to authors.

13) Disclosure statement and competing interests: We updated our journal's competing interests policy in January 2022 and request authors to consider both actual and perceived competing interests. Please review the policy <https://www.embopress.org/competing-interests> and update your competing interests if necessary.

14) Every published paper now includes a 'Synopsis' to further enhance discoverability. Synopses are displayed on the journal webpage and are freely accessible to all readers. They include a short stand first (maximum of 300 characters, including space) as well as 2-5 one-sentences bullet points that summarizes the paper. Please write the bullet points to summarize the key NEW findings. They should be designed to be complementary to the abstract - i.e. not repeat the same text. We encourage inclusion of key acronyms and quantitative information (maximum of 30 words / bullet point). Please use the passive voice. Please attach these in a separate file or send them by email, we will incorporate them accordingly.

Please also suggest a striking image or visual abstract to illustrate your article as a PNG file 550 px wide x 300-600 px high. Share synopsis text and image, as well as eTOC:

Please note that these would be the final versions and changes during proofing are usually not allowed

15) As part of the EMBO Publications transparent editorial process initiative (see our policy here:

https://www.embopress.org/transparent-process#Review_Process), Molecular Systems Biology will publish online a Peer Review File (PRF) to accompany accepted manuscripts.

In the event of acceptance, this file will be published in conjunction with your paper and will include the anonymous referee reports, your point-by-point response and all pertinent correspondence relating to the manuscript. Let us know whether you agree with the publication of the PRF and as here, if you want to remove or not any figures from it prior to publication.

Please note that the Authors checklist will be published at the end of the PRF.

Molecular Systems Biology has a "scooping protection" policy, whereby similar findings that are published by others during review or revision are not a criterion for rejection. Should you decide to submit a revised version, I do ask that you get in touch after three months if you have not completed it, to update us on the status.

I look forward to receiving your revised manuscript.

Yours sincerely,

Poonam Bheda, PhD
Scientific Editor
Molecular Systems Biology

Reviewer #2:

The authors have done a great job in addressing most of my previous comments. However, my concern regarding software installation and usage still persists.

Here are specific suggestions for improvement:

- 1) Instead of simply listing requirements, provide users with specific and tested commands for installing all the prerequisites, including the InfoHiC package itself.
- 2) Develop a comprehensive, step-by-step tutorial that elucidates how to predict config-specific Hi-C maps from WGS. The tutorial should start with WGS FASTQ and the reference genome files, guiding users through the procedure to generate Hi-C matrices and neo-loop/enhancer-hijacking annotations for each contig.
- 3) Add a detailed tutorial explaining how to train the model from scratch using WGS and Hi-C data. This is crucial for users who want to apply InfoHiC to cancer types where pretrained models are not available.

While acknowledging the authors' efforts in addressing my previous comments, I want to stress the importance of facilitating users in installing and running the pipelines with their own data. As this is a method paper, I cannot recommend the paper to be published in Molecular Systems Biology unless the concerns above are satisfactorily addressed.

Reviewer #3:

The authors have addressed my major concerns.

Reviewer #2 cross-commenting:

After carefully examining the authors' responses to the benchmarking concerns raised by Reviewers 1 and 3, I believe the following critical points need further attention:

1. A more comprehensive comparison between DeepC and InfoHiC is necessary (regarding to the response to the second point from Reviewer 1). A reasonable strategy would involve initially creating a Venn diagram to provide a global view of the percentage of DeepC-specific, InfoHiC-specific, and common SE-hijacking events. Following this, it is essential to demonstrate whether the InfoHiC-specific events are more enriched with cancer-specific genes and are more likely to be validated by experimental data compared with DeepC-specific events. Currently, the authors primarily focus on InfoHiC-specific events, and their analysis is not sufficient to establish that InfoHiC outperforms DeepC in external datasets.
 2. The comparison between InfoHiC and NeoLoopFinder is not fair, and this should be corrected (regarding to the response to the second point from Reviewer 3). In their comparison, NeoLoopFinder and InfoHiC used different set of SVs as input. The results reported by Wang et al. 2021 are limited to large SVs and inter-chromosomal translocations because the goal of that paper was to use Hi-C as the only input to identify enhancer hijacking events. At that time, Hi-C could only detect SVs greater than 1Mb. However, NeoLoopFinder itself supports SVs detected by other technologies, including WGS, as input. To ensure a fair comparison, the authors should rerun NeoLoopFinder with the same 1,944 SVs used by InfoHiC as input. Otherwise, the results may reflect the difference between WGS and Hi-C in detecting SVs rather than the difference between NeoLoopFinder and InfoHiC.
-

Answer to Reviewer #2:

The authors have done a great job in addressing most of my previous comments. However, my concern regarding software installation and usage still persists.

Here are specific suggestions for improvement:

1) Instead of simply listing requirements, provide users with specific and tested commands for installing all the prerequisites, including the InfoHiC package itself.

Answer) Thank you for the comment. We understand that complexities of our workflows including installation may prevent users from using it easily. Thus, we extensively refactored our workflows with the effort equivalent to writing one more application paper, and managed them using Snakemake (<https://snakemake.readthedocs.io/en/stable/>). All the prerequisites can be installed using conda environments. We provide commands for installing them as below (<https://github.com/DMCB-GIST/InfoHiC>)

```
snakemake --core all --use-conda hic_mapping_env
snakemake --core all --use-conda InfoHiC_env
```

InfoHiC had a dependency to InfoGenomeR, which might be hard to install as well. Thus, we also refactored InfoGenomeR workflow using Snakemake, and it can be easily installed as below in the same machine. (<https://github.com/dmclab/InfoGenomeR>).

```
snakemake --core all --use-conda InfoGenomeR_env
```

2) Develop a comprehensive, step-by-step tutorial that elucidates how to predict config-specific Hi-C maps from WGS. The tutorial should start with WGS FASTQ and the reference genome files, guiding users through the procedure to generate Hi-C matrices and neo-loop/enhancer-hijacking annotations for each contig.

Answer) Thank you for the comment. We provide tutorials starting from WGS FASTQ, and explain output files including neo-loop/enhancer annotations for each contig. The details are explained in <https://github.com/DMCB-GIST/InfoHiC>

Step 1. InfoGenomeR workspace setting

```
cd $HOME

#InfoGenomeR repository
git clone https://github.com/dmclab/InfoGenomeR.git
InfoGenomeR_repo=${PWD}/InfoGenomeR
cd ${InfoGenomeR_repo}

#environment setting
snakemake --core all --use-conda InfoGenomeR_env

#dataset download
snakemake --cores all --use-conda InfoGenomeR_download

#example dataset download
snakemake --core all --use-conda InfoGenomeR_example_download

# make a workspace directory
cd ${InfoGenomeR_repo}
workspace_dir=InfoGenomeR_workspace1
mkdir -p ${workspace_dir}

# link the reference in the workspace directory
```

```
In -s ${PWD}/humandb/ref ${workspace_dir}/ref
# use low coverage examples data
In -s ${PWD}/examples/fastq ${workspace_dir}/fastq
```

Step 2. I InfoGenomeR run

Note that we provide a single command line for InfoGenomeR run.

```
# Run the InfoGenomeR workflow. Select either somatic or total mode.
snakemake --core all --use-conda ${workspace_dir}/InfoGenomeR_output --config mode=somatic
min_ploidy=2.5 max_ploidy=3.5

# This is InfoGenomeR output
InfoGenomeR_output=`readlink -f ${workspace_dir}/InfoGenomeR_output`
```

Step 3. InfoHiC workspace setting

```
# go to the InfoHiC repo directory
cd ${InfoHiC_repo}

# make a workspace directory
workspace_dir=InfoHiC_workspace1
mkdir -p ${workspace_dir}

# Link the reference and the InfoHiC model in the workspace
In -s ${PWD}/humandb/ref ${workspace_dir}/ref
In -s ${PWD}/models/breast_model ${workspace_dir}/model

# take the InfoGenomeR output
cp -r ${InfoGenomeR_output} ${workspace_dir}/InfoGenomeR_output
```

Step 4. InfoHiC run

Note that we provide a single command line for InfoHiC run, and post processing one after InfoHiC run.

```
# set wildcards
resolution=40000
window=2000000
cancer_type=BRCA
model=CSCN_encoding
gpu=1 # use the available gpu

# run the InfoHiC workflow
snakemake --cores all --use-conda
${workspace_dir}/InfoHiC_prediction/hic_${resolution}.window_${window}.${cancer_type}.${model}.gpu${gpu}

# run the post process
snakemake --cores all --use-conda
${workspace_dir}/InfoHiC_prediction/hic_${resolution}.window_${window}.${cancer_type}.${model}.gpu${gpu}.
post_process
```

Outputs

Outputs are provided per contig from each chromosome set.

```
InfoHiC_workspace1/InfoHiC_prediction/hic_40000.window_2000000.BRCA.CSCN_encoding.gpu1.post_proc
ess
└─ euler.8.14
```

- the post_process output folder contains chromosome sets where InfoHiC performed SV Hi-C prediction.
 - euler.8.14 indicates chromosomes 8 and 14
- Each euler directory contains SV contigs
 - contig2.58095296.66230780 indicates contig index, contig coordinate start, and contig coordinate end in a prediction window.
- Each contig folder contains predicted Hi-C information
 - contigs.tsv
 - full contig information of the contig
 - contig index, breakpoint graph node index1, breakpoint graph node index2, ref chrom, ref pos, ref end, cumulative length
 - contigs_ref_coor.tsv
 - the current SV prediction window in the contig
 - ref chrom, ref pos, ref end, ref orientation, contig index, contig pos, contig end, contig orientation
 - ens_gene.bed
 - Ensembl gene location in the config
 - super_enhancer.bed
 - SE location in the config
 - typical_enhancer.bed
 - TE location in the contig
 - hic_iced_loop.bed
 - neo-loop location in the iced contig
 - hic_iced.matrix
 - prediction results with post iced normalization
 - hic_iced_tad.bed
 - noe-TAD location in the iced contig. The fourth column indicates a TAD level from spectralTAD.
 - hic_iced.pdf
 - Visualization of contig Hi-C matrix, with neo-TAD, neo-loop, ens_gene, super_enhancer, and typical_enhancer annotations

3) Add a detailed tutorial explaining how to train the model from scratch using WGS and Hi-C data. This is crucial for users who want to apply InfoHiC to cancer types where pretrained models are not available.

Answer) Thank you for the comment. We added a section explaining how to train InfoHiC from scratch using WGS and Hi-C data in <https://github.com/DMCB-GIST/InfoHiC>.

InfoHiC training

- The training step can be skipped because we provide InfoHiC trained models.
- It takes two weeks to train on 1 Mb windows (~10 epoches).
- On 2 Mb windows, it takes a month.
- Take the one-day example on 1 Mb windows using the subset of T47D Hi-C reads.

Inputs

- InfoGenomeR output from WGS (<https://github.com/dmcb/InfoGenomeR>)
- Hi-C reads

Workflow

```
# go to the InfoHiC base directory
cd InfoHiC

# make the root output directory
root_dir=InfoHiC_training_output
mkdir -p ${root_dir}

# link the reference and the seed file in the root directory
ln -s ${PWD}/humandb/ref ${root_dir}/ref
ln -s ${PWD}/models/seed_file ${root_dir}/seed_file

# Place Hi-C reads and the InfoGenomeR output in the root directory
cp -r ${PWD}/examples/T47D_hic_reads_subset ${root_dir}/hic_reads
cp -r examples/T47D_chromosomes_8_14/InfoGenomeR_output ${root_dir}/InfoGenomeR_output

# set wildcards
resolution=40000

# Run Hi-C read mapping
snakemake --cores all --use-conda ${root_dir}/hic_${resolution}/3div

# set wildcards
window=1040000
split_idx=1
split_rate=0.1 # 0.1 for valid and test, and 0.8 for train

# Run Hi-C data post process and split
snakemake --cores all --use-conda
${root_dir}/hic_${resolution}/window ${window}.split${split_idx}_rate${split_rate}
```

```

# set wildcards
cancer_type=BRCA
mode=CSCN_encoding
gpu=1
epoch=1

# InfoHiC training
snakemake --cores all --use-conda
${root_dir}/InfoHiC_training.${cancer_type}/hic_${resolution}.window_${window}.split${split_idx}_rate${split_rate}/${mode}/gpu${gpu}.epoch${epoch}

```

Then, go to Export an InfoHiC model

Export an InfoHiC model

After training, select a model and export it

```

# List the models, and select a best_checkpoint saved at the last step
model_dir=${root_dir}/InfoHiC_training.${cancer_type}/hic_${resolution}.window_${window}.split${split_idx}_rate${split_rate}/${mode}/gpu${gpu}.epoch${epoch}
ls -l ${model_dir}/training_run_data_40kb_ascn_ACGT_rc_shift

# This is an example of best_checkpoint
model=${model_dir}/training_run_data_40kb_ascn_ACGT_rc_shift/best_checkpoint-51198

# copy the model into the model_dir base with the best_checkpoint prefix
cp -r $model.data-00000-of-00001 ${model_dir}/best_checkpoint.data-00000-of-00001
cp -r $model.index ${model_dir}/best_checkpoint.index
cp -r $model.meta ${model_dir}/best_checkpoint.meta

# export it as a contig model finally
snakemake --cores all --use-conda ${model_dir}.model

```

Outputs

Make a link of the model output in an InfoHiC workspace before running InfoHiC.

```

InfoHiC_training_output/InfoHiC_training.BRCA/hic_40000.window_1040000.split1_rate0.1/CSCN_encoding/gpu1.epoch1.model
├── contig_model.data-00000-of-00001
├── contig_model.index
└── contig_model.meta

```

While acknowledging the authors' efforts in addressing my previous comments, I want to stress the importance of facilitating users in installing and running the pipelines with their own data. As this is a method paper, I cannot recommend the paper to be published in Molecular Systems Biology unless the concerns above are satisfactorily addressed.

Answer) Thank you for the comment. We agree that it should be easy to install and use. We refactored our workflows extensively during revision periods and believe that this effort can address the concerns.

Answer to Reviewer #2 cross-commenting:

After carefully examining the authors' responses to the benchmarking concerns raised by Reviewers 1 and 3, I believe the following critical points need further attention:

1. A more comprehensive comparison between DeepC and InfoHiC is necessary (regarding to the response to the second point from Reviewer 1). A reasonable strategy would involve initially creating a Venn diagram to provide a global view of the percentage of DeepC-specific, InfoHiC-specific, and common SE-hijacking events. Following this, it is essential to demonstrate whether the InfoHiC-specific events are more enriched with cancer-specific genes and are more likely to be validated by experimental data compared with DeepC-specific events. Currently, the authors primarily focus on InfoHiC-specific events, and their analysis is not sufficient to establish that InfoHiC outperforms DeepC in external datasets.

Answer) Thank you for the comment. We added a Venn diagram for each cancer cell line. Differences between deepC-SV and InfoHiC can be due to the fact that InfoHiC predicted a Hi-C matrix on each SV contig, where multiple SVs could be clustered and generate a neo-TAD together (a neo-TAD from Contig 3 in Appendix Fig. S2), while deepC-SV predicted a Hi-C matrix for each SV (two neo-TADs from SV 1 and SV 2 in Appendix Fig. S2). For example, in Appendix Fig. S2, deepC-SV produced false positives (a-f and b-f) and false negatives (ab-hi) against the Hi-C experiment. To show that InfoHiC outperforms DeepC in external datasets, we compared Hi-C intensities from deepC-specific, InfoHiC-specific, and common SE-hijacking events as follows.

Page 8-9) In addition, we investigated whether deepC-SV could predict InfoHiC results of SE hijacking events in neo-TADs. The number of SE-hijacked genes predicted by InfoHiC ranged from 77 to 562 in the five breast cancer cell lines (total $n = 1,567$). Because the SE-hijacked genes could interact with multiple SEs, we additionally counted all pairs of genes and SEs, which ranged from 257 to 2,867 (total $n = 7,879$). DeepC-SV could predict 85.2% ($1,335 / 1,567$) of SE-hijacked genes and 63.0% ($4,966 / 7,879$) of gene-SE pairs. Thirty-seven percent of gene-SE pairs were not detected by deepC-SV, even though deepC-SV predicted 1.6x more (total $n = 12,886$) gene-SE pairs than InfoHiC (Appendix Fig. 8). This could be because that InfoHiC predicted a Hi-C matrix on each SV contig, where multiple SVs could be clustered and generate a neo-TAD together (a neo-TAD from Contig 3 in Appendix Fig. S2), while deepC-SV predicted a Hi-C matrix for each SV (two neo-TADs from SV 1 and SV 2 in Appendix Fig. S2). Differences between deepC-SV and InfoHiC were shown in Appendix Fig. S2, where deepC-SV produced false positives (a-f and b-f) and false negatives (ab-hi) against the Hi-C experiment. Accordingly, experimental Hi-C intensities of deepC-SV gene-SE pairs could be lower than those predicted by InfoHiC. To investigate which gene-SE pairs were more accurate, we compared experimental Hi-C values of predicted gene-SE pairs (deepC-SV specific, InfoHiC-specific, and shared events) in each cell line. Indeed, deepC-SV specific events showed lower intensities in cancer Hi-C experiments across five cell lines compared to InfoHiC-specific events (Appendix Fig. S8). In BT474 and SKBR3, InfoHiC-specific events showed the highest Hi-C intensities. In HCC1954, MCF7, and T47D, shared events showed the highest Hi-C intensities, and InfoHiC-specific events showed the second highest intensities. deepC-SV specific events showed the lowest Hi-C intensities across five cell lines. Overall, the result indicated that events predicted by InfoHiC were observed in Hi-C experiments with higher signals than deepC-SV predicted events.

Next, we investigated cancer-related genes in gene-SE pairs. Among 17 cancer-related genes involved with SE hijacking events predicted by InfoHiC (Appendix Table S3), deepC-SV could predict SE hijacking events of 14 cancer-related genes (82.4%, 14/17). However, deepC-SV could not find any SE involved with three cancer-related genes (17.6%, 3/17): *NFATC2* and *SALL4* in MCF7, *TERT* in HCC1954. These genes were involved with multiple SVs. In addition, deepC-SV could not find some of the gene-SE pairs: *NT5C2* in T47D, and *GNAS* and *BCL6* in SKBR3. Rearrangements around *TERT* were previously reported to be involved with Hi-C contact map changes in HCC1954 (Akdemir et al. 2020). *SALL4* (T. Chen et al. 2020) and *GNAS* (X. Jin et al. 2019) were known to promote breast cancer progression, and previously reported to be involved with SE hijacking events (Xu et al. 2022; Wang et al. 2021). To further verify these gene-SE pair findings predicted by InfoHiC, we investigated the Hi-C intensities of each gene-SE pair in the cancer Hi-C and noncancerous Hi-C experiments (HMEC and MCF10A) (Appendix Table S5). The cancer Hi-C experiments showed higher Hi-C intensities of the gene-SE pairs, compared to noncancerous cell lines, up to a 74X fold change (*GNAS* in SKBR3). For

several genes (*TERT* in HCC1954, *SALL4* in MCF7, and *NT5C2* in T47D), the noncancerous cell lines showed zero Hi-C intensity. The results demonstrated that InfoHiC could predict SE hijacking events not found by deepC-SV, which could be verified in Hi-C experiments.

Appendix page 9-10)

Appendix Fig. 8: Comparison of gene-SE pairs in predicted neo-TADs between deepC-SV and InfoHiC.

A Venn diagram (left) of predicted gene-SE pairs and Hi-C intensities in the Hi-C experiment (right) for each cancer cell line. Cancer cell lines include **a** BT474, **b** HCC1954, **c** MCF7, **d** SKBR3, and **e** T47D. The Venn diagram (left) shows portions of deepC-SV specific (cyan), InfoHiC specific (orange), and shared events between them (brown). The box plot (right) shows Hi-C intensities observed in the cancer Hi-C experiment. Hi-C intensities of predicted gene-SE pairs (either by deepC-SV or InfoHiC, or shared) in HMEC and MCF10A experiments were measured as controls. The intensities were normalized as read counts per million Hi-C reads. Box plot colors represent events predicted by deepC-SV (blue), InfoHiC (orange), and shared events between them (brown). Boxplot center lines are medians, box limits are upper and lower quartiles, and dots are outliers.

2. The comparison between InfoHiC and NeoLoopFinder is not fair, and this should be corrected (regarding to the response to the second point from Reviewer 3). In their comparison, NeoLoopFinder and InfoHiC used different set of SVs as input. The results reported by Wang et al. 2021 are limited to large SVs and inter-chromosomal translocations because the goal of that paper was to use Hi-C as the only input to identify enhancer hijacking events. At that time, Hi-C could only detect SVs greater than 1Mb. However, NeoLoopFinder itself supports SVs detected by other technologies, including WGS, as input. To ensure a fair comparison, the authors should rerun NeoLoopFinder with the same 1,944 SVs used by InfoHiC as input. Otherwise, the results may reflect the difference between WGS and Hi-C in detecting SVs rather than the difference between NeoLoopFinder and InfoHiC.

Answer) Thank you for the comment. Before starting comparison, we had to correct the number of SVs. They were double counted as 1,944 breakpoints (two breakpoint per SV) in the previous manuscript. In the revised manuscript, it was corrected as follows.

Page 13)

In contrast, InfoHiC found a total of 1,944 SVs → In contrast, InfoHiC found a total of 972 SVs

Answer) For a fairer comparison, we fed 972 SVs into NeoLoopFinder as inputs, and added Venn diagrams. In addition, we added Hi-C intensity plots of NeoLoopFinder-specific, InfoHiC-specific, and shared events to show that InfoHiC prediction results were observed in high intensities in cancer Hi-C experiments. Moreover, the comparison needs to include broader discussion on analytic and prediction methods, and we added it into the last part of the comparison paragraph.

Page 13-14)

For a fairer comparison, we ran NeoLoopFinder (v0.4.3-r2) on 10 kb Hi-C data, and then fed 972 SVs including small SVs that we used for InfoHiC into NeoLoopFinder as inputs. As a result, NeoLoopFinder discovered 1,377 neo-loop-involved genes across the five cell lines. Overall, InfoHiC found 68% of neo-loop-involved genes from NeoLoopFinder, while NeoLoopFinder found 25% of neo-loop-involved genes from InfoHiC (Appendix Fig. 15). The remaining difference may be because even though we fed 972 SVs into NeoLoopFinder as inputs, NeoLoopFinder filtered out 51% of them (n=492) in the SV assembly process. NeoLoopFinder measures distance decay of Hi-C contacts in the SV contig, and then filters out SVs if they are not consistent with the global distance decay (Wang et al. 2021). Sixty-two point eight percent of SVs filtered out in the assembly were < 1 Mb, suggesting that Hi-C contacts from the small SVs might not be consistent with the global decay.

To further compare NeoLoopFinder and InfoHiC results, we investigated intensities of NeoLoopFinder-specific, InfoHiC-specific, and shared events in cancer Hi-C experiments (Appendix Fig. 15). For each neo-loop-involved gene, we measured Hi-C intensities of gene-SE pairs. Even though high intensities do not necessarily represent neo-loop events, they can represent gene-SE interactions observable in cancer Hi-C experiments. HMEC and MCF10A were used as controls. In five cancer cell line Hi-C data (Appendix Fig. 15), shared events showed the highest intensities and NeoLoopFinder-specific events were the second highest in MCF7, SKBR3, and T47D. InfoHiC-specific events showed the lowest intensities in these cell lines, suggesting that they may contain false positive predictions. However, in HCC1954 and BT474, InfoHiC-specific events were observed in higher intensities than NeoLoopFinder-specific events. The result suggested that NeoLoopFinder might not detect true events from HCC1954 and BT474 Hi-C data even though they had high intensities. Compared to analytic methods including NeoLoopFinder which require confident Hi-C observation from the corresponding SV, Hi-C prediction methods do not require Hi-C observation from SVs, and can discover neo-loops using the model trained on genomic regions with Hi-C observation. As a prediction method, InfoHiC could enable more sensitive neo-loop findings from SVs, including small or complex SVs where confident Hi-C observations are not available. InfoHiC-specific events predicted from HCC1954 and BT474 might be such cases. Moreover, we highlight that for patient-specific SVs that have not been observed in cell lines, the prediction method enables us to discover neo-loop from them as we showed in medulloblastoma cases.

Appendix page 17-18)

BT474

MCF7

HCC1954

Appendix Fig. 15: Comparison of neo-loop-involved genes found by NeoLoopFinder and InfoHiC. A Venn diagram (left) of neo-loop-involved genes with SE hijacking events and Hi-C intensities in the Hi-C experiment (right) for each cancer cell line. Cancer cell lines include **a** BT474, **b** HCC1954, **c** MCF7, **d** SKBR3, and **e** T47D. The Venn diagram (left) shows portions of NeoLoopFinder-specific (cyan), InfoHiC-specific (orange), and shared events between them (brown). For each neo-loop-involved genes, Hi-C intensities of gene-SE pairs were measured. The box plot (right) shows Hi-C intensities of gene-SE pairs of neo-loop-involved genes observed in the cancer Hi-C experiment. Hi-C intensities of gene-SE pairs (either by NeoLoopFinder or InfoHiC, or shared) in HMEC and MCF10A experiments were measured as controls. The intensities were normalized as read counts per million Hi-C reads. Box plot colors represent events predicted by NeoLoopFinder (blue), InfoHiC (orange), and shared events between them (brown). Boxplot center lines are medians, box limits are upper and lower quartiles, and dots are outliers.

18th Jul 2024

Manuscript Number: MSB-2023-11897RR

Title: Prediction of the 3D cancer genome from whole genome sequencing using InfoHiC

Author: Yeonghun Lee

Sung-Hye Park

Hyunju Lee

Dear Dr. Lee,

Thank you for the submission of your revised manuscript to Molecular Systems Biology. I am pleased to inform you that we will be able to accept your manuscript pending the following final amendments and appropriate response to the remaining suggestions from Reviewer 2:

1) In line with the remaining comments from Reviewer 2, please include a comparison between DeepC and InfoHiC to determine the overlap of cancer-related genes identified by the two algorithms and perform a statistical test to show whether these are more enriched by InfoHiC than DeepC.

2) In the main manuscript file, please include keywords to max. 5.

3) Please combine your "Code Availability" and "Availability of data and materials" sections to "Data availability", and only include your newly generated code and sequencing data information in this section. This section needs to be formatted according to the example below:

"The computer code produced in this study are available in the following databases:

- Chip-Seq data: Gene Expression Omnibus GSE46748 (<https://www.ncbi.nlm.nih.gov/geo/query/acc.cgi?acc=GSE46748>)

- Modeling computer scripts: GitHub (<https://github.com/SysBioChalmers/GECKO/releases/tag/v1.0>)

- [data type]: [full name of the resource] [accession number/identifier] ([doi or URL or identifiers.org/DATABASE:ACCESSION])"

4) Please ensure that all newly produced sequencing datasets are now publicly released.

5) For the sequencing datasets that you have reanalyzed, where the information is currently in the "Availability of data and materials" please incorporate the information into the relevant Methods section(s). Alternatively, you may consider including *data citations in the reference list* to directly cite datasets that were re-used and obtained from public databases. Data citations in the article text are distinct from normal bibliographical citations and should directly link to the database records from which the data can be accessed. In the main text, data citations are formatted as follows: "Data ref: Smith et al, 2001" or "Data ref: NCBI Sequence Read Archive PRJNA342805, 2017". In the Reference list, data citations must be labeled with "[DATASET]". A data reference must provide the database name, accession number/identifiers and a resolvable link to the landing page from which the data can be accessed at the end of the reference. Further instructions are available at .

6) Please rename "Competing Interests" to "Disclosure and competing interests statement". We updated our journal's competing interests policy in January 2022 and request authors to consider both actual and perceived competing interests. Please review the policy <https://www.embopress.org/competing-interests> and update your competing interests if necessary.

7) Author contributions: Please remove it from the manuscript and specify author contributions in our submission system.

CRedit has replaced the traditional author contributions section because it offers a systematic machine-readable author contributions format that allows for more effective research assessment. You are encouraged to use the free text boxes beneath each contributing author's name to add specific details on the author's contribution. More information is available in our guide to authors:

<https://www.embopress.org/page/journal/17574684/authorguide#authorshipguidelines>

8) References: Please correct the reference citation in the reference list. Currently your format is to use et al after 7 authors - we would like you to increase this so that et al is only used once there are more than 10 authors on a paper. Please check "Author Guidelines" for more information.

<https://www.embopress.org/page/journal/17574684/authorguide#referencesformat>

9) In the Methods, please take care of the following:

- Studies with human research participants: Please move the "Ethics approval and consent to participate" section into the relevant Methods section. Although you have stated that the samples were obtained with informed consent in the acknowledgements, please also be sure to include this in the Methods. Please also edit the ethics statements to state that the experiments conformed to the principles set out in the WMA Declaration of Helsinki and the Department of Health and Human Services Belmont Report.

10) All materials and methods need to be described in the main text using our 'Structured Methods' format, which is required for all research articles. According to this format, the Methods section includes a Reagents and Tools Table (listing key reagents, experimental models, software and relevant equipment and including their sources and relevant identifiers) followed by a Methods and Protocols section describing the methods using a step-by-step protocol format. The aim is to facilitate adoption of the methodologies across labs. More information on how to adhere to this format as well as a downloadable template (.docx) for the Reagents and Tools Table can be found in our author guidelines:

<https://www.embopress.org/page/journal/17574684/authorguide#structuredmethods>

An example of a Method paper with Structured Methods can be found here:
<https://www.embopress.org/doi/10.15252/msb.20178071>.

11) Please place individual sections of the manuscript in the following order: Title page - Abstract & Keywords - Introduction - Results - Discussion - Methods - Data Availability - Acknowledgements - Disclosure and Competing Interests Statement - References - Figure Legends - Expanded View Figure Legends.

12) For the figures and figure legends, please take care of the following:

- Please move the Figure Legends after the References
- Please note that information related to n is missing in the legend of figures 4b; 5b.
- Please note that we require exact p-values to be reported. Currently exact p-values are not provided - please update this in the figures or respective figure legends.

13) Appendix:

- Please remove the "Appendix Information" section from the main manuscript file.
- Please upload the Appendix as a single PDF - i.e. please combine the 2 appendix files containing the figures and tables
- Please add a Table of Contents with page numbers for each figure/table
- Please rename the figures/tables as Appendix Figure S1-S14 and Appendix Table S1-S11 throughout the ms and Appendix file. Please also ensure the callouts are updated in the manuscript to match.

14) Synopsis:

- Synopsis image: Please provide a graphic that summarises the main findings of the manuscript on a glance and upload it as a high-resolution jpeg file 550 pixels wide x (250-400) pixels high.
- Synopsis text: Please provide a short standfirst (maximum of 300 characters, including space), limit the bullet points to max. 5 and upload it as a separate .doc file. Please write the bullet points to summarise the key NEW findings. They should be designed to be complementary to the abstract - i.e. not repeat the same text. We encourage inclusion of key acronyms and quantitative information (maximum of 30 words / bullet point). Please use the passive voice.
- Please check your synopsis text and image before submission with your revised manuscript. Please be aware that in the proof stage minor corrections only are allowed (e.g., typos).

15) As part of the EMBO Publications transparent editorial process initiative (see our policy here:

https://www.embopress.org/transparent-process#Review_Process), Molecular Systems Biology will publish online a Peer Review File (PRF) to accompany accepted manuscripts. This file will be published in conjunction with your paper and will include the anonymous referee reports, your point-by-point response and all pertinent correspondence relating to the manuscript. Let us know whether you agree with the publication of the PRF and as here, if you want to remove or not any figures from it prior to publication. Please note that the Authors checklist will be published at the end of the PRF.

16) Please provide a point-by-point letter INCLUDING my comments as well as the reviewer's reports and your detailed responses (as Word file).

I look forward to reading a new revised version of your manuscript as soon as possible.

Yours sincerely,

Poonam Bheda, PhD
Scientific Editor
Molecular Systems Biology

Reviewer #2:

Overall, the authors have addressed my previous concerns well, particularly regarding the software installation and usage, which I greatly appreciate.

However, there is one remaining issue regarding the comparison between DeepC and InfoHiC. Previously, I suggested it is essential to demonstrate whether the InfoHiC-specific SE-hijacking events are more enriched with cancer-specific genes compared to DeepC-specific events. However, in this revision, the authors still focus on the 17 cancer-related genes involved with SE-hijacking events predicted by InfoHiC, without mentioning the cancer-related genes detected by DeepC.

Taking the BT474 cell line in Appendix Figure 8 as an example, DeepC detected 3,395 unique gene-SE pairs. It is hard to imagine these pairs do not overlap with any unique cancer-related genes. I suggest a further analysis as follows:

1. From the gene-SE pairs specifically detected by DeepC and InfoHiC, count how many overlap with cancer-related genes and how many do not.
2. Perform a statistical test, such as Fisher's exact test, to determine if the gene-SE pairs uniquely detected by InfoHiC are more enriched with cancer-related genes than those uniquely detected by DeepC.

Answer to Editor after the initial quality check:

Thank you for your contribution to Molecular Systems Biology. Unfortunately there are some problems with your submission, which are listed below:

- Thank you for including a comparison between DeepC and InfoHiC in your manuscript. However, we would ask you to clarify/elaborate what you mean by "The tests showed that cancer-related genes were more enriched by InfoHiC than by deepC-SV." Do you mean that a 0.024 p-value is better than a 0.049 p-value, or do you mean 17 out of 1567 genes identified is better than 25 out of 2764?

Answer) Thank you for the comment. We clarified the sentence as follows.

Page 9) Enrichment analysis with Fisher's exact test showed that InfoHiC ($p = 0.024$) had a lower p value than DeepC-SV ($p = 0.049$), indicating cancer-related genes were more enriched by InfoHiC.

- Please remove the Reagent and Tools Table from the manuscript file and upload it as an individual file. For more information, please check <https://www.embopress.org/page/journal/14602075/authorguide#structuredmethods> and download the template for Reagent Table (attached for your convenience)

Answer) Thank you for the comment. We removed the Reagent and Tools Table from the manuscript file and uploaded it as an individual file, Reagents and tools table.docx.

- Thank you for updating most of the Appendix figure/table callouts. We note that the callout for Appendix figure S6 should still be corrected - 'Appendix' needs to be added (currently 'Fig. S6'); In addition, callouts for Appendix figure S15 should be corrected - 'S' needs to be added (currently 'Appendix Fig. 15').

Answer) Thank you for the comment. We edited callouts for Appendix figures as follows.

Page 8) Furthermore, we performed analysis of covariance (ANCOVA) by controlling for CN effects on gene expression (Appendix Fig. S6).

Page 13) Overall, InfoHiC found 68% of neo-loop-involved genes from NeoLoopFinder, while NeoLoopFinder found 25% of neo-loop-involved genes from InfoHiC (Appendix Fig. S15).

Page 14) To further compare NeoLoopFinder and InfoHiC results, we investigated intensities of NeoLoopFinder-specific, InfoHiC-specific, and shared events in cancer Hi-C experiments (Appendix Fig. S15).

Page 14) HMEC and MCF10A were used as controls. In five cancer cell line Hi-C data (Appendix Fig. S15), shared events showed the highest intensities and NeoLoopFinder-specific events were the second highest in MCF7, SKBR3, and T47D.

- Please upload the synopsis image in png, jpg, or tiff format, exactly 550 pixels wide and between 300-600 pixels high.

Answer) Thank you for the comment. We uploaded the synopsis image in png format which is 550 pixels wide and 350 pixels high.

- Please upload the synopsis text as a separate .docx document to your submission. Please also remove 1 word from the second bullet point, as these should be 30 words maximum.

Answer) Thank you for the comment. We uploaded Synopsis.docx file. We removed two words "as input" from the previous sentence as follows.

The second bullet point in Synopsis.docx) InfoHiC takes whole genome sequencing reads in contrast to other Hi-C prediction tools requiring pre-defined sequences, enabling Hi-C prediction from cancer genomes where users cannot define sequence inputs easily.

Please address the comments above and resubmit your manuscript using the tracking number MSB-2023-11897RRR. Please login to our online manuscript submission system using the following link. <https://msb.msubmit.net/cgi-bin/main.plex?el=A51k6CuM6A6DTNM2J6A9ftddKwiK0mvwdSRF2rq4TFMTQY>
Please do not share this URL as it will give anyone who clicks it access to your account.

Answer) Thank you for the comment. We addressed the comments above and resubmitted the manuscript.

Answer to Editor:

Thank you for the submission of your revised manuscript to Molecular Systems Biology. I am pleased to inform you that we will be able to accept your manuscript pending the following final amendments and appropriate response to the remaining suggestions from Reviewer 2:

1) In line with the remaining comments from Reviewer 2, please include a comparison between DeepC and InfoHiC to determine the overlap of cancer-related genes identified by the two algorithms and perform a statistical test to show whether these are more enriched by InfoHiC than DeepC.

Answer) Thank you for the comment. We added a paragraph to show cancer-related genes are more enriched by InfoHiC as follows.

Page 8) Oncogenes from the Cosmic Cancer Gene Census (Futreal *et al*, 2004) and the TCGA driver gene list (Bailey *et al*, 2018) were selected as cancer-related genes to investigate the impact of enhancer hijacking events on them. Tumor suppressor genes were excluded from the cancer-related gene list.

Page 9) Next, we investigated cancer-related genes in gene-SE pairs. Among 60,234 genes annotated in the Ensembl database (Howe *et al*, 2021), DeepC-SV and InfoHiC predicted that 2,764 and 1,567 genes were involved with SE hijacking events, respectively. Among 382 cancer-related genes (Futreal *et al*, 2004; Bailey *et al*, 2018), DeepC-SV and InfoHiC discovered 25 ($p = 0.049$, one-sided Fisher's exact test) and 17 ($p = 0.024$, one-sided Fisher's exact test) cancer-related genes, respectively. The tests showed that cancer-related genes were more enriched by InfoHiC than by deepC-SV. Then, we investigated the Hi-C intensities of each gene-SE pair in the cancer Hi-C and noncancerous Hi-C experiments (HMEC and MCF10A). Fourteen genes were shared between deepC-SV and InfoHiC. Cancer-related genes exclusively found by InfoHiC ($n = 3$) showed an average 4.3-fold change in cancer cell line Hi-C data, while those found by deepC-SV ($n = 11$) showed an average 2.6-fold change, indicating that InfoHiC discovered cancer-related genes with higher Hi-C intensity evidence than deepC-SV. Furthermore, we observed that several gene-SE pairs found by deepC-SV have extremely low intensities (< 5 Hi-C read counts) in cancer Hi-C experiments (Appendix Table S5), suggesting that deepC-SV discovered false positive gene-SE pairs.

Appendix page 26)

Appendix Table S5. DeepC-SV specific cancer related genes involved with SE hijacking events and their Hi-C intensities.

Gene	Super enhancer (SEdb)	Cancer cell line	Hi-C intensity corresponding cancer cell line (left)	Hi-C intensity noncancerous cell line (MCF10A)	Hi-C intensity noncancerous cell line (HMEC)
CCND1	SE_02_05600663	BT474	0	0	0
CCND1	SE_02_05700895	BT474	0	0	0
SALL4	SE_01_07200274	BT474	0	0	0
MUC4	SE_01_07200856	SKBR3	28	23	9
MUC4	SE_02_05700781	SKBR3	28	22	19
MUC4	SE_01_07200856	MCF7	30	23	9
MUC4	SE_02_05700781	MCF7	31	22	19
CNBD1	SE_02_11500036	SKBR3	0	0	0
NRAS	SE_01_04600226	MCF7	3	0	0
JAK3	SE_02_05600655	MCF7	13	2	6
JAK3	SE_02_05600875	MCF7	36	13	19
JAK3	SE_02_07900144	MCF7	28	14	6
JAK3	SE_02_40300145	MCF7	28	14	6
PPM1D	SE_01_04501060	MCF7	2	0	0
PPM1D	SE_01_07200858	MCF7	4	0	0
TERT	SE_02_40100594	T47D	3	1	0
U2AF1	SE_01_04501149	HCC1954	3	6	4
U2AF1	SE_01_07200506	HCC1954	2	4	2
U2AF1	SE_02_05600338	HCC1954	3	6	4
U2AF1	SE_02_05700934	HCC1954	3	6	4
U2AF1	SE_02_39400338	HCC1954	29	0	0
SOX2	SE_01_04500829	HCC1954	12	0	0
SOX2	SE_02_05600722	HCC1954	12	0	0
SOX2	SE_02_05700975	HCC1954	12	0	0
SOX2	SE_02_40300315	HCC1954	12	0	0
BCL6	SE_02_05701179	HCC1954	24	4	4

2) In the main manuscript file, please include keywords to max. 5.

Answer) Thank you for the comment. We added keywords as follows.

Page 2)

Keywords

Hi-C prediction; 3D genome; Structural variation; Cancer genome; Deep learning

3) Please combine your "Code Availability" and "Availability of data and materials" sections to "Data availability", and only include your newly generated code and sequencing data information in this section. This section needs to be formatted according to the example below:

"The computer code produced in this study are available in the following databases:

- Chip-Seq data: Gene Expression Omnibus GSE46748

(<https://www.ncbi.nlm.nih.gov/geo/query/acc.cgi?acc=GSE46748>)

- Modeling computer scripts: GitHub

(<https://github.com/SysBioChalmers/GECKO/releases/tag/v1.0>)

- [data type]: [full name of the resource] [accession number/identifier] ([doi or URL or identifiers.org/DATABASE:ACCESSION)]"

Answer) Thank you for the comment. We combined them to Data availability and only included newly generated code and data as follows.

Page 25)

Data availability

The datasets and computer code produced in this study are available in the following databases:

- InfoHiC: GitHub (<https://github.com/dmcb-gist/InfoHiC>)
- WGS and RNA-seq fastq files and variant calls (SNVs, indels, CNAs, and SVs) from pediatric patients with medulloblastoma: European Genome-phenome Archive EGAD00001009507 (<https://ega-archive.org/datasets/EGAD00001009507>)
- Gene expression profiles from pediatric patients with medulloblastoma: Zenodo (<https://doi.org/10.5281/zenodo.10544039>)

4) Please ensure that all newly produced sequencing datasets are now publicly released.

Thank you for the comment. We uploaded newly produced sequencing fastq reads to European Genome-phenome Archive, and they were publicly released with an accession ID EGAD00001009507 (<https://ega-archive.org/datasets/EGAD00001009507>).

Whole genome sequencing and somatic mutation profiles and whole genome transcription files of pediatric medulloblastoma

Fastq, Mutect (SNVs), Platypus (indels), and InfoGenomeR (SVs and CNAs) calls from whole genome sequencing data and fastq files of whole genome transcription data of five patients with pediatric medulloblastoma.

🔥 1 sample 📍 DAC: EGAC00001002896 🧬 Technology: Illumina NovaSeq 6000

Access Policy 1 Study **50 Files (973.1 GB)**

Metadata Request Access

This table displays only public information pertaining to the files in the dataset. If you wish to access this dataset, please submit a request. If you already have access to these data files, please consult the download documentation.

File checksums visible behind login

ID	File Type	Size	Located in	
EGAF00007253564	vcf	6.7 kB	 	
EGAF00007253565	vcf	20.9 kB	 	
EGAF00007253566	vcf	3.7 MB	 	
EGAF00007253567	vcf	622.6 kB	 	
EGAF00007253568	vcf	8.8 kB	 	
EGAF00007253569	vcf	28.3 kB	 	
EGAF00007253570	vcf	3.8 MB	 	
EGAF00007253571	vcf	565.4 kB	 	
EGAF00007253572	vcf	7.6 kB	 	
EGAF00007253573	vcf	20.3 kB	 	
EGAF00007253574	vcf	3.5 MB	 	
EGAF00007253575	vcf	531.0 kB	 	
EGAF00007253576	vcf	6.1 kB	 	
EGAF00007253577	vcf	17.9 kB	 	
EGAF00007253578	vcf	3.3 MB	 	
EGAF00007253579	vcf	563.6 kB	 	
EGAF00007253580	vcf	6.7 kB	 	
EGAF00007253581	vcf	21.8 kB	 	
EGAF00007253582	vcf	3.4 MB	 	
EGAF00007253583	vcf	550.3 kB	 	
EGAF00008544677	fastq.gz	26.4 GB		
EGAF00008544678	fastq.gz	28.4 GB		
EGAF00008544679	fastq.gz	33.1 GB		
EGAF00008556508	fastq.gz	30.0 GB		
EGAF00008556514	fastq.gz	31.9 GB		
EGAF00008556515	fastq.gz	33.6 GB		
EGAF00008556517	fastq.gz	34.4 GB		
EGAF00008556518	fastq.gz	31.9 GB		
EGAF00008556519	fastq.gz	28.6 GB		
EGAF00008556520	fastq.gz	54.0 GB		
EGAF00008556521	fastq.gz	59.7 GB		
EGAF00008556522	fastq.gz	34.6 GB		
EGAF00008556523	fastq.gz	56.5 GB		
EGAF00008556524	fastq.gz	61.6 GB		
EGAF00008556525	fastq.gz	70.2 GB		
EGAF00008556526	fastq.gz	68.2 GB		
EGAF00008556527	fastq.gz	67.9 GB		
EGAF00008556528	fastq.gz	65.8 GB		
EGAF00008556529	fastq.gz	61.1 GB		
EGAF00008556532	fastq.gz	63.4 GB		
EGAF00008570589	fastq.gz	3.0 GB		
EGAF00008570590	fastq.gz	3.1 GB		
EGAF00008570591	fastq.gz	3.1 GB		
EGAF00008570592	fastq.gz	3.3 GB		
EGAF00008570593	fastq.gz	3.1 GB		
EGAF00008570594	fastq.gz	3.2 GB		
EGAF00008570595	fastq.gz	3.1 GB		
EGAF00008570596	fastq.gz	3.3 GB		
EGAF00008570597	fastq.gz	3.1 GB		
EGAF00008570598	fastq.gz	3.3 GB		

50 Files (973.1 GB)

5) For the sequencing datasets that you have reanalyzed, where the information is currently in the "Availability of data and materials" please incorporate the information into the relevant Methods section(s). Alternatively, you may consider including *data citations in the reference list* to directly cite datasets that were re-used and obtained from public databases. Data citations in the article text are distinct from normal bibliographical citations and should directly link to the database records from which the data can be accessed. In the main text, data citations are formatted as follows: "Data ref: Smith et al, 2001" or "Data ref: NCBI Sequence Read Archive PRJNA342805, 2017". In the Reference list, data citations must be labeled with "[DATASET]". A data reference must provide the database name, accession number/identifiers and a resolvable link to the landing page from which the data can be accessed at the end of the reference. Further instructions are available at <https://www.embopress.org/page/journal/17574684/authorguide#referencesformat>.

Answer) Thank you for the comment. We moved the information of sequencing datasets that we have reanalyzed into the Methods section as follows.

Page 24)

Data processing

For breast cancer cell lines, WGS fastq files of the T47D, MCF7, SKBR3, and K562 cell lines were downloaded from the SRA: T47D (PRJNA380394), MCF7 (PRJNA486532), SKBR3 (PRJNA476239), and K562 (PRJNA380394). Paired-end WGS reads from the T47D, MCF7, SKBR3, and K562 cell lines were mapped to the human reference genome (GRCh37) using BWA-MEM (Li, 2013) with default parameters (version 0.7.15). WGS bam files of the HCC1954 and BT474 cell lines were downloaded from TCGA in the GDC data portal and CCLE in Google Cloud Storage, respectively. For patients with breast cancer from TCGA, WGS bam files were downloaded from dbGaP (accession code phs000178.v11.p8) [https://www.ncbi.nlm.nih.gov/projects/gap/cgi-bin/study.cgi?study_id=phs000178.v11.p8]. For brain tumors, paired-end WGS reads (150 bp) were mapped GRCh37 using BWA-MEM (Li, 2013) with default parameters (version 0.7.15).

From WGS bam files, somatic SVs were detected by DELLY2 (Rausch *et al*, 2012), Manta (Chen *et al*, 2016), and novoBreak (Chong *et al*, 2017). We merged SV calls from WGS with those from long reads for HCC1954 and SKBR3. SV calls of HCC1954 from linked-read sequencing data (10X Genomics) were downloaded from https://cf.10xgenomics.com/samples/genome/HCC1954T_WGS_210/HCC1954T_WGS_210_large_s_vs.vcf.gz. SV calls for SKBR3 from long-read sequencing data (PacBio) (Nattestad *et al*, 2018) were downloaded from http://labshare.cshl.edu/shares/schatzlab/www-data/skbr3/reads_lr_skbr3.fa_ngmlr-0.2.3_mapped.bam.sniffles1kb_auto_l8_s5_noalt.vcf.gz. Somatic SNVs and indels were detected using Mutect (Cibulskis *et al*, 2013) and Platypus (Rimmer *et al*, 2014) respectively. Variants were annotated using ANNOVAR (Wang *et al*, 2010).

Hi-C fastq files of breast cancer cell lines were downloaded from the sequencing read archive (SRA); MCF7, BT474, and SKBR3 Hi-C data (accession codes PRJNA430222); HCC1954 (accession code PRJNA479882); and T47D (PRJNA438511), respectively. Hi-C fastq files of the K562 and NPC cell line were downloaded from the SRA (accession code PRJNA268125 and PRJNA798046, respectively). Hi-C reads were mapped to GRCh37 using the 3DIV pipeline (Kim *et al*, 2021a). CTCF ChIP-seq data of NPC cells and ChIA-PET data of a neuroblastoma cell line were downloaded from ENCODE (accession code ENCSR125NBL and ENCSR514HBO, respectively).

For breast cancer cell lines, RNA-seq gene expression data were downloaded from the Cancer Cell Line Encyclopedia (<https://sites.broadinstitute.org/ccle/datasets>). For patients with breast cancer from TCGA, RNA-seq gene expression data were downloaded from the GDC data portal (<https://gdc.cancer.gov/access-data/gdc-data-portal>). For brain tumors, paired-end RNA-seq reads (101 bp) were mapped to GRCh37 using STAR (Dobin *et al*, 2013), and gene expression was quantified using Cufflinks (Trapnell *et al*, 2010).

6) Please rename "Competing Interests" to "Disclosure and competing interests statement". We updated our journal's competing interests policy in January 2022 and request authors to consider both actual and perceived competing interests. Please review the policy <https://www.embopress.org/competing-interests> and update your competing interests if necessary.

Answer) Thank you for the comment. We renamed Competing Interests to Disclosure and competing interests statement as follows.

Page 26)

Disclosure and competing interests statement

The authors declare that they have no competing interests.

7) Author contributions: Please remove it from the manuscript and specify author contributions in our submission system. CRediT has replaced the traditional author contributions section because it offers a systematic machine-readable author contributions format that allows for more effective research assessment. You are encouraged to use the free text boxes beneath each contributing author's name to add specific details on the author's contribution. More information is available in our guide to authors:

<https://www.embopress.org/page/journal/17574684/authorguide#authorshipguidelines>

Answer) Thank you for the comment. We removed Author contributions from the manuscript.

8) References: Please correct the reference citation in the reference list. Currently your format is to use et al after 7 authors - we would like you to increase this so that et al is only used once there are more than 10 authors on a paper. Please check "Author Guidelines" for more information.

<https://www.embopress.org/page/journal/17574684/authorguide#referencesformat>

Answer) Thank you for the comment. We reformatted References and et al is only used once there are more than 10 authors in the revised manuscript.

9) In the Methods, please take care of the following:

- Studies with human research participants: Please move the "Ethics approval and consent to participate" section into the relevant Methods section. Although you have stated that the samples were obtained with informed consent in the acknowledgements, please also be sure to include this in the Methods. Please also edit the ethics statements to state that the experiments conformed to the principles set out in the WMA Declaration of Helsinki and the Department of Health and Human Services Belmont Report.

Answer) Thank you for the comment. We move the Ethics approval and consent to participate section into the Sample preparation section in the Methods as follows.

Page 24)

Sample preparation

For brain tumors, macro-dissection was performed on tumor areas with a tumor cell content of more than 90% stained with hematoxylin-eosin. Genomic DNA and total RNA were extracted from the freshly frozen tissues and peripheral blood of patients using the Maxwell RSC Genomic DNA and simplyRNA Tissue Kit (Promega, Madison, WI, USA), respectively. The WGS library was prepared using the TruSeq Nano DNA Kit (Illumina, San Diego, CA, USA) and sequenced using the Illumina NovaSeq6000 platform. The RNA-seq library was prepared using the SureSelectXT RNA Direct Library Kit (Agilent Technologies, Santa Clara, CA, USA) and sequenced using the Illumina NovaSeq6000 platform. The study on brain tumors was approved by the Institutional Review Board of Seoul National University Hospital (IRB No:1905-108-1035). The experiments conformed to the principles set out in the WMA Declaration of Helsinki and the Department of Health and Human Services Belmont Report.

10) All materials and methods need to be described in the main text using our 'Structured Methods' format, which is required for all research articles. According to this format, the Methods section includes a Reagents and Tools Table (listing key reagents, experimental models, software and relevant equipment and including their sources and relevant identifiers) followed by a Methods and Protocols section describing the methods using a step-by-step protocol format. The aim is to facilitate adoption of the methodologies across labs. More information on how to adhere to this format as well as a downloadable template (.docx) for the Reagents and Tools Table can be found in our author guidelines: <https://www.embopress.org/page/journal/17574684/authorguide#structuredmethods>

An example of a Method paper with Structured Methods can be found here: <https://www.embopress.org/doi/10.15252/msb.20178071>.

Answer) Thank you for the comment. We added the Reagents and tools table as follows.

Pages 16-18)

Reagents and tools table

Reagent/Resource	Reference or Source	Identifier or Catalog Number
Software		
InfoHiC	This article https://github.com/DMCB-GIST/InfoHiC	
InfoGenomeR	Lee and Lee (2021) https://github.com/dmclab/InfoGenomeR	
DeepC	Schwessinger et al (2020) https://github.com/Hughes-Genome-Group/deepC	
NeoLoopFinder	Wang et al (2021) https://github.com/XiaoTaoWang/NeoLoopFinder	
BWA	Li (2023) https://bio-bwa.sourceforge.net	
3DIV pipeline	Kim et al (2021a) https://github.com/kaistcbfg/3divv2	
SpectralTAD	Cresswell et al (2020) https://github.com/dozmorovlab/SpectralTAD	
Peakachu	Salameh et al (2020) https://github.com/tariks/peakachu	
covNorm	Kim et al (2021b) https://github.com/kaistcbfg/covNormRpkg	
Mutect	Cibulskis et al (2013) https://github.com/broadinstitute/mutect	

Platypus	Rimmer et al (2014) https://rahmanteamdevelopment.github.io/Platypus	
DELLY2	Rausch et al (2013) https://github.com/dellytools/delly	
Manta	Chen et al (2016) https://github.com/Illumina/manta	
novoBreak	Chong et al (2017) https://github.com/czc/nb_distribution	
ANNOVAR	Wang et al (2010) https://annovar.openbioinformatics.org/en/latest/	
SAMtools	Li (2011) https://www.htslib.org/	
BCFtools	Li (2011) https://www.htslib.org/	
STAR	Dobin et al (2013) https://github.com/alexdobin/STAR	
Cufflinks	Trapnell et al (2010) https://github.com/cole-trapnell-lab/cufflinks	
PWMScan	Ambrosini et al (2018) https://sourceforge.net/projects/pwmscan/	
Other		
Maxwell RSC simplyRNA Tissue Kit	Promega	AS1340
Maxwell RSC Genomic DNA Kit	Promega	AS1880
TruSeq Nano DNA Kit	Illumina	20015965
SureSelect ^{XT} RNA Direct Library Kit	Agilent	G7564A
Illumina NovaSeq 6000	Illumina	

11) Please place individual sections of the manuscript in the following order: Title page - Abstract & Keywords - Introduction - Results - Discussion - Methods - Data Availability - Acknowledgements - Disclosure and Competing Interests Statement - References - Figure Legends - Expanded View Figure Legends.

Answer) Thank you for the comment. We placed individual sections in the order described above in the revised script.

**12) For the figures and figure legends, please take care of the following:
- Please move the Figure Legends after the References**

Answer) Thank you for the comment. We moved the Figure legends after the References.

- Please note that information related to n is missing in the legend of figures 4b; 5b.
- Please note that we require exact p -values to be reported. Currently exact p -values are not provided - please update this in the figures or respective figure legends.

Answer) Thank you for the comment. We moved the Figure legends after the References. We added information related to n and exact p -values to the legend of figures 4b and 5b as follows.

The figure 4 legend in page 31) For neo-TADs, the group n numbers and p values are $n = 9,594$ (reference), $n = 514$ and $p = 6.43e-15$ (neo-TAD), and $n = 100$ and $p = 3.79e-05$ (SE hijacking) in BT474; $n = 10,401$, $n = 315$ and $p = 5.80e-03$, $n = 75$ and $p = 0.120$ in HCC1954; $n = 13,333$, $n = 435$ and $p = 0.029$, $n = 104$ and $p = 1.23e-03$ in MCF7; $n = 12,752$, $n = 425$ and $p = 1.22e-03$, $n = 192$ and $p = 2.65e-07$ in SKBR3; $n = 9,691$, $n = 46$ and $p = 0.306$, $n = 45$ and $p = 0.428$ in T47D. For neo-loops, the group n numbers and p values are $n = 9,537$ (reference), $n = 608$ and $p = 1.21e-06$ (neo-loop), and $n = 441$ and $p = 1.49e-14$ (SE hijacking) in BT474; $n = 9,944$, $n = 524$ and $p = 8.72e-05$, $n = 323$ and $p = 6.06e-04$ in HCC1954; $n = 12,889$, $n = 556$ and $p = 0.391$, $n = 427$ and $p = 8.30e-08$ in MCF7; $n = 12,985$, $n = 274$ and $p = 1.79e-03$, $n = 104$ and $p = 0.025$ in SKBR3; $n = 9,524$, $n = 186$ and $p = 0.279$, $n = 72$ and $p = 0.114$ in T47D.

The figure 5 legend in page 31) For neo-TADs, the group n numbers and p values are $n = 900,669$ (reference), $n = 4,363$ and $p = 0.025$ (neo-TAD), and $n = 1,434$ and $p = 3.29e-03$ (SE hijacking) in copy number 2; $n = 835,576$, $n = 7,364$ and $p = 4.94e-03$, and $n = 2,480$ and $p = 8.73e-05$ in copy number 3; $n = 746,955$, $n = 9,816$ and $p = 1.41e-05$, and $n = 3,023$ and $p = 4.66e-03$ in copy number 4; $n = 332,513$, $n = 7,775$ and $p = 0.046$, and $n = 3,093$ and $p = 2.51e-05$ in copy number 5; $n = 111,171$, $n = 5,535$ and $p = 0.046$, and $n = 1,904$ and $p = 9.08e-03$ in copy number 6; $n = 52,380$, $n = 3,136$ and $p = 0.145$, and $n = 1,332$ and $p = 0.015$ in copy number 7; $n = 17,593$, $n = 1,839$ and $p = 0.058$, and $n = 1,012$ and $p = 1.66e-03$ in copy number 8. For neo-loops, the group n numbers and p values are $n = 860,538$, $n = 10,908$ and $p = 0.037$, and $n = 5,742$ and $p = 5.01e-06$ in copy number 2; $n = 852,035$, $n = 15,608$ and $p = 4.48e-03$, and $n = 9,072$ and $p = 1.12e-08$ in copy number 3; $n = 756,233$, $n = 20,563$ and $p = 0.017$, and $n = 11,898$ and $p = 2.20e-16$ in copy number 4; $n = 326,213$, $n = 15,003$ and $p = 0.113$, and $n = 8,856$ and $p = 1.80e-06$ in copy number 5; $n = 104,949$, $n = 8,818$ and $p = 0.037$, and $n = 8,818$ and $p = 5.63e-05$ in copy number 6; $n = 48,319$, $n = 4,837$ and $p = 0.395$, and $n = 4,837$ and $p = 1.12e-05$ in copy number 7; $n = 15,357$, $n = 2,648$ and $p = 0.050$, and $n = 2,648$ and $p = 1.51e-05$ in copy number 8.

13) Appendix:

- Please remove the "Appendix Information" section from the main manuscript file.

Answer) Thank you for the comment. We removed the Appendix Information section.

- Please upload the Appendix as a single PDF - i.e. please combine the 2 appendix files containing the figures and tables

Answer) Thank you for the comment. We combined them into a single PDF.

- Please add a Table of Contents with page numbers for each figure/table

Answer) Thank you for the comment. We added a Table of Contents with page numbers for each figure/table.

- Please rename the figures/tables as Appendix Figure S1-S14 and Appendix Table S1-S11 throughout the ms and Appendix file. Please also ensure the callouts are updated in the manuscript to match.

Answer) Thank you for the comment. We renamed the figures/tables as Appendix Figure S1-S16 and Appendix Table S1-S11 in the revised manuscript.

14) Synopsis:

- **Synopsis image:** Please provide a graphic that summarises the main findings of the manuscript on a glance and upload it as a high-resolution jpeg file 550 pixels wide x (250-400) pixels high.

- **Synopsis text:** Please provide a short standfirst (maximum of 300 characters, including space), limit the bullet points to max. 5 and upload it as a separate .doc file. Please write the bullet points to summarise the key NEW findings. They should be designed to be complementary to the abstract - i.e. not repeat the same text. We encourage inclusion of key acronyms and quantitative information (maximum of 30 words / bullet point). Please use the passive voice.

Answer) Thank you for the comment. We made a Synopsis image and text as follows.

Synopsis.pdf)

Page 1 in Synopsis.docx)

InfoHiC is developed for cancer 3D genome prediction enabling to find neo-TADs and neo-loops starting from whole genome sequencing reads.

- InfoHiC is trained from available cancer Hi-C data and predicts Hi-C matrices for patients with cancer, which eliminates the burden of high Hi-C sequencing costs.
- InfoHiC takes whole genome sequencing reads as input in contrast to other Hi-C prediction tools requiring pre-defined sequences, enabling Hi-C prediction from cancer genomes where users cannot define sequence inputs easily.
- It analyzes all structural variation types including those found in non-coding regions, and predicts the impact on cancer development in the 3D genome context.
- It can lead to personalized medication by discovering neo-TADs and neo-loops in patients where common coding driver mutations or copy number changes are not found.

15) As part of the EMBO Publications transparent editorial process initiative (see our policy here: https://www.embopress.org/transparent-process#Review_Process), Molecular Systems Biology will publish online a Peer Review File (PRF) to accompany accepted manuscripts. This file will be published in conjunction with your paper and will include the anonymous referee reports, your point-by-point response and all pertinent correspondence relating to the manuscript. Let us know whether you agree with the publication of the PRF and as here, if you

want to remove or not any figures from it prior to publication. Please note that the Authors checklist will be published at the end of the PRF.

Answer) Thank you for the comment. Yes, we agree with the publication of the PRF.

16) Please provide a point-by-point letter INCLUDING my comments as well as the reviewer's reports and your detailed responses (as Word file).

Answer) Thank you for the comment. We included our detailed responses in this Word file.

I look forward to reading a new revised version of your manuscript as soon as possible.

Answer to Reviewer #2:

Reviewer #2:

Overall, the authors have addressed my previous concerns well, particularly regarding the software installation and usage, which I greatly appreciate.

However, there is one remaining issue regarding the comparison between DeepC and InfoHiC. Previously, I suggested it is essential to demonstrate whether the InfoHiC-specific SE-hijacking events are more enriched with cancer-specific genes compared to DeepC-specific events. However, in this revision, the authors still focus on the 17 cancer-related genes involved with SE-hijacking events predicted by InfoHiC, without mentioning the cancer-related genes detected by DeepC.

Taking the BT474 cell line in Appendix Figure 8 as an example, DeepC detected 3,395 unique gene-SE pairs. It is hard to imagine these pairs do not overlap with any unique cancer-related genes. I suggest a further analysis as follows:

1. From the gene-SE pairs specifically detected by DeepC and InfoHiC, count how many overlap with cancer-related genes and how many do not.

Answer) Thank you for the comment. We added a paragraph to describe how many overlap with cancer-related genes and how many do not as follows.

Page 8) Oncogenes from the Cosmic Cancer Gene Census (Futreal *et al*, 2004) and the TCGA driver gene list (Bailey *et al*, 2018) were selected as cancer-related genes to investigate the impact of enhancer hijacking events on them. Tumor suppressor genes were excluded from the cancer-related gene list.

Page 9) Next, we investigated cancer-related genes in gene-SE pairs. Among 60,234 genes annotated in the Ensembl database (Howe *et al*, 2021), DeepC-SV and InfoHiC predicted that 2,764 and 1,567 genes were involved with SE hijacking events, respectively. Among 382 cancer-related genes (Futreal *et al*, 2004; Bailey *et al*, 2018), DeepC-SV and InfoHiC discovered 25 ($p = 0.049$, one-sided Fisher's exact test) and 17 ($p = 0.024$, one-sided Fisher's exact test) cancer-related genes, respectively. The tests showed that cancer-related genes were more enriched by InfoHiC than deepC-SV. Then, we investigated the Hi-C intensities of each gene-SE pair in the cancer Hi-C and noncancerous Hi-C experiments (HMEC and MCF10A). Fourteen genes were shared between deepC-SV and InfoHiC. Cancer-related genes exclusively found by InfoHiC ($n = 3$) showed an average 4.3X fold change in cancer cell line Hi-C data, while those found by deepC-SV ($n = 11$) showed an average 2.6X fold change, indicating that InfoHiC discovered cancer-related genes that had higher Hi-C intensity evidence than deepC-SV. Furthermore, we observed that several gene-SE pairs found by deepC-SV have extremely low intensities (< 5 Hi-C read counts) in cancer Hi-C experiments (Appendix Table S5), suggesting that deepC-SV discovered false positive gene-SE pairs.

2. Perform a statistical test, such as Fisher's exact test, to determine if the gene-SE pairs

uniquely detected by InfoHiC are more enriched with cancer-related genes than those uniquely detected by DeepC.

Answer) Thank you for the comment. We added sentences to show that InfoHiC are more enriched with cancer-related genes with a statistical test as follows.

Page 9) DeepC-SV and InfoHiC discovered 25 ($p = 0.049$, one-sided Fisher's exact test) and 17 ($p = 0.024$, one-sided Fisher's exact test) cancer-related genes, respectively. The tests showed that cancer-related genes were more enriched by InfoHiC than deepC-SV.

9th Sep 2024

Manuscript number: MSB-2023-11897RRR

Title: Prediction of the 3D cancer genome from whole genome sequencing using InfoHiC

Dear Dr. Lee,

Thank you again for sending us your revised manuscript. We are now satisfied with the modifications made and I am pleased to inform you that your paper has been accepted for publication.

Yours sincerely,

Poonam Bheda, PhD
Scientific Editor
Molecular Systems Biology
